# Statistical Consistency and Generalization of Contrastive Representation Learning

**Yuanfan Li** [1]  **Xiyuan Wei** [2]  **Tianbao Yang** [2]  **Yiming Ying** [1]

## Abstract

Contrastive representation learning (CRL) underpins many modern foundation models. Despite recent theoretical progress, existing analyses suffer from several key limitations: (i) the statistical consistency of CRL remains poorly understood; (ii) available generalization bounds deteriorate as the number of negative samples increases, contradicting the empirical benefits of large negative sets; and (iii) the retrieval performance of CRL has received limited theoretical attention. In this paper, we develop a unified statistical learning theory for CRL. For downstream tasks, we evaluate retrieval quality using an AUC-type population criterion and show that the contrastive loss is *statistically consistent* with optimal ranking. We further establish a *calibration-style inequality* that quantitatively relates excess contrastive risk to excess retrieval suboptimality. For upstream training, we study both supervised and self-supervised contrastive objectives and derive generalization bounds of order $O(1/m + 1/\sqrt{n})$ and $O(1/\sqrt{m} + 1/\sqrt{n})$, respectively, where $m$ denotes the number of negative samples and $n$ the number of anchor points. These bounds not only explain the empirical advantages of large negative sets but also reveal an explicit trade-off between $m$ and $n$. Extensive experiments on large-scale vision–language models corroborate our theoretical predictions.

## 1. Introduction

Foundation models have emerged as a prominent paradigm in artificial intelligence, achieving remarkable success across diverse domains. A wide class of foundation models is representation models (Radford et al., 2021; Jia et al., 2021; Karpukhin et al., 2020). Such models are trained at scale on large and heterogeneous datasets, often using self-supervision, to learn general-purpose representations that can be adapted to a wide range of downstream tasks.

A central paradigm underlying many successful representation models is *contrastive representation learning* (CRL). For two modalities $\mathcal{X}$ and $\mathcal{Y}$, CRL aims to learn a scoring function parameterized by $\mathbf{w}$, $s_{\mathbf{w}} : \mathcal{X} \times \mathcal{Y} \mapsto \mathbb{R}$, that assigns higher scores to positive pairs than to negative pairs. By pulling together semantically related pairs $(\mathbf{x}, \mathbf{y})$ and pushing apart mismatched pairs $(\mathbf{x}, \mathbf{y}')$ in the embedding space, CRL learns transferable and task-agnostic representations. This mechanism can be implemented by minimizing the following objective (Wang & Isola, 2020):

$$\mathcal{L}(s_{\mathbf{w}}) = \mathbb{E}_{\mathbf{x}}\mathbb{E}_{\mathbf{y}\sim p_{\mathbf{x}}^+} \tau \log \mathbb{E}_{\mathbf{y}'\sim p_{\mathbf{x}}^-} \exp\left(\tfrac{\Delta_{\mathbf{w}}(\mathbf{x},\mathbf{y},\mathbf{y}')}{\tau}\right), \quad (1)$$

where $p_{\mathbf{x}}^+$ and $p_{\mathbf{x}}^-$ denote the positive and negative data distributions for the anchor point $\mathbf{x}$, respectively, and $\Delta_{\mathbf{w}}(\mathbf{x}, \mathbf{y}, \mathbf{y}') := s_{\mathbf{w}}(\mathbf{x}, \mathbf{y}') - s_{\mathbf{w}}(\mathbf{x}, \mathbf{y})$ measures the difference in the similarity score between a negative pair and a positive pair. Typically the scoring function can be the inner product of the neural network encoder/representation $f_{\mathbf{w}}$, i.e., $s_{\mathbf{w}}(\mathbf{x}, \mathbf{y}) = f_{\mathbf{w}}(\mathbf{x})^\top f_{\mathbf{w}}(\mathbf{y})$.

CRL has demonstrated remarkable empirical success in computer vision, natural language processing, and multimodal learning (An et al., 2023; Chen et al., 2020; He et al., 2020; Radford et al., 2021). In particular, several large-scale vision–language and multimodal foundation models adopt contrastive objectives as their core pretraining mechanism, enabling superior zero-shot and few-shot generalization (Alayrac et al., 2022; Ramesh et al., 2022). These results suggest that CRL serves not merely as a task-specific technique but as a *foundational learning paradigm* for building general-purpose representations. Despite its impressive empirical success, a unified learning theory that rigorously characterizes the statistical behavior of CRL-based pretraining and its implications for downstream generalization remains limited.

A fundamental question is how would upstream pretraining affect downstream tasks. In a prominent line of research (Saunshi et al., 2019; Ash et al., 2022; Bao et al., 2022;

[1]University of Sydney [2]Texas A&M University. Correspondence to: Yiming Ying <yiming.ying@sydney.edu.au>.

*Proceedings of the 43rd International Conference on Machine Learning*, Seoul, South Korea. PMLR 306, 2026. Copyright 2026 by the author(s).

Nozawa et al., 2020), the downstream task is evaluated by the so-called *surrogate gap*, i.e., the discrepancy between the population risk of the contrastive objective and the mean supervised loss of the downstream task. Existing results establish that a small contrastive risk yields a small mean supervised loss in the downstream task, when using a linear classifier. However, these guarantees fall short of *statistical consistency*: whether, as the sample size grows, minimizers of the contrastive objective converge to predictors achieving the *least possible* downstream population risk.

In practice, Eq. (1) is usually estimated through finite samples, i.e., $n$ positive pairs and $m$ negative examples for each anchor point. Therefore, another important theoretical problem is to give a generalization analysis that captures the role of $m$ and $n$. Most existing generalization analyses for CRL typically scale as $O(m/\sqrt{n})$ (Saunshi et al., 2019) or $O(\log m/\sqrt{n})$ (Lei et al., 2023), which suggest that the generalization performance deteriorates as the number of negative samples $m$ increases. Such theoretical results are at odds with empirical practice, where using a large number of negative samples is known to significantly improve performance. For example, SimCLR (Chen et al., 2020) employs up to 8192 negative examples, while CLIP (Radford et al., 2021) further increases this number to 32,768. Consequently, existing theory fails to provide a satisfactory explanation for the strong generalization behavior observed in modern CRL methods.

**Our first main contribution** is to establish statistical consistency and a calibration-style inequality for CRL, which are among the most fundamental problems in the framework of Statistical Learning Theory (SLT) (Bartlett et al., 2006; Bousquet et al., 2003; Zhang, 2004; Vapnik, 2013). To motivate our study, we note that the contrastive objective in Eq. (1) is inherently a pairwise ranking loss, encouraging relevant items to receive higher scores than irrelevant ones for a given query. This observation naturally motivates us to evaluate CRL through a retrieval-based criterion rather than classification metrics (e.g. misclassification error or cross entropy loss). Specifically, we consider the following performance measure:

$$
\begin{aligned}
\mathcal{E}(s) =& \mathbb{E}_{\mathbf{x}} \, \mathbb{E}_{\mathbf{y} \sim p_{\mathbf{x}}^+, \, \mathbf{y}' \sim p_{\mathbf{x}}^-} \left[ \mathbb{I}[s(\mathbf{x}, \mathbf{y}) > s(\mathbf{x}, \mathbf{y}')] \right] \\
&+ \frac{1}{2} \mathbb{E}_{\mathbf{x}} \, \mathbb{E}_{\mathbf{y} \sim p_{\mathbf{x}}^+, \, \mathbf{y}' \sim p_{\mathbf{x}}^-} \left[ \mathbb{I}[s(\mathbf{x}, \mathbf{y}) = s(\mathbf{x}, \mathbf{y}')] \right],
\end{aligned}
\tag{2}
$$

which corresponds to an AUC-type measure (Agarwal et al., 2005; Clémençon et al., 2008; Cortes & Mohri, 2003; Ying et al., 2016) and represents the expected probability that a relevant item $\mathbf{y}$ is ranked above an irrelevant item $\mathbf{y}'$ given a query $\mathbf{x}$. A higher value of $\mathcal{E}(s)$ corresponds to better retrieval performance in the downstream task.

In this paper, we establish the *statistical consistency* of contrastive representation learning in the following sense.

For any sequence of scoring functions $\{s_n\}_{n \geq 1}$,

$$
\begin{aligned}
\mathcal{L}(s_n) \to \mathcal{L}^* &:= \inf_s \mathcal{L}(s) \\
&\implies \mathcal{E}(s_n) \to \mathcal{E}^* := \sup_s \mathcal{E}(s).
\end{aligned}
\tag{3}
$$

where the infimum and supremum are taken over all measurable scoring functions $s : \mathcal{X} \times \mathcal{Y} \to \mathbb{R}$. This result shows that convergence of the contrastive risk guarantees convergence to the optimal achievable retrieval performance in the downstream task.

Moreover, we establish a *calibration-style inequality* of the form

$$
\mathcal{E}^* - \mathcal{E}(s) \lesssim \sqrt{\mathcal{L}(s) - \mathcal{L}^*}, \qquad \forall \, s,
$$

which quantitatively links excess contrastive risk to suboptimality in retrieval performance and implies statistical consistency defined by Eq. (3). The above results provide a principled theoretical explanation for why minimizing the contrastive objective yields statistically optimal ranking performance, addressing a fundamental question that has remained largely unexplored in the existing literature.

**Our second main contribution** is a comprehensive generalization analysis that reveals how the performance of CRL relies on the number of anchor points and negative examples. In this paper, we consider two learning regimes widely used in practice. In supervised CRL, $m$ negative examples are sampled for each anchor point independently. While in self-supervised CRL, $m$ negative examples are *shared* across all of anchor points. To analyze the generalization performance, note that the contrastive loss in Eq. (1) exhibit a *compositional structure*: the *outer* structure centers on positive sample pairs $(\mathbf{x}, \mathbf{y})$, while the *inner* structure is formed by the negative samples used to contrast against each positive pair. Inspired by this, we decompose the generalization gap into inner error and outer error. The outer error is mainly responsible for the sampling of $n$ positive pairs and is of the order $1/\sqrt{n}$. The inner error is introduced to quantify the statistical discrepancy arising from comparing each anchor with only $m$ negative samples rather than the full population of negative examples, and reflects the bias induced by sampling a finite number of negative examples. To control the inner error, we apply uniform convergence theory to derive a bound of $O(1/\sqrt{m})$ for self-supervised CRL. For the inner error in supervised CRL, we reformulate contrastive losses as stochastic minimization problems, which enables us to analyze the resulting estimators using algorithmic stability theory for empirical risk minimization (ERM) (Bousquet & Elisseeff, 2002), yielding an $O(1/m)$ bound. Combining these analyses yield overall generalization bounds of $O(1/\sqrt{m} + 1/\sqrt{n})$ for self-supervised CRL and $O(1/m + 1/\sqrt{n})$ for supervised CRL. Our result has two important implications. First, it formally explains why generalization performance in CRL *improves* as the num-

ber of negative samples $m$ increases. Second, it reveals an explicit trade-off between $m$ and $n$.

## 1.1. Related Work

A foundational framework for analyzing the generalization error of contrastive representation learning (CRL) was introduced by Saunshi et al. (2019), who derived bounds scaling linearly with the number of negative examples $m$ using Rademacher complexity techniques (Bartlett & Mendelson, 2002). Subsequent work improved this dependence to $O(\log m)$ (Lei et al., 2023). Hieu et al. (2025) further derived generalization bounds for deep neural networks. Related generalization results have also been established for Transformer-based CRL (Oko et al., 2025) and adversarial contrastive learning (Zou & Liu, 2023).

Complementary to upstream generalization bounds, the surrogate gap was studied by Saunshi et al. (2019) and is further refined in (Bao et al., 2022; Ash et al., 2022; Nozawa & Sato, 2021). These works show that a small contrastive risk implies a small mean supervised loss under linear evaluation. However, such results do not address *statistical consistency*, namely whether minimizing the contrastive objective yields *the least* downstream performance in the population limit. In contrast, our work establishes statistical consistency and derives calibration-style inequalities that directly relate upstream and downstream excess risks.

CRL has also been studied from alternative perspectives, including PAC-Bayesian generalization (Nozawa et al., 2020), spectral analysis (HaoChen et al., 2021), optimization algorithms (Yuan et al., 2022; Qiu et al., 2023), and information-theoretic or geometric viewpoints (Tsai et al., 2021). Recent work has further explored CRL beyond linear evaluation, including zero-shot prediction and vision–language models (Chen et al., 2024; Mehta & Harchaoui, 2025; Oko et al., 2025). CRL can be regarded as conditional (compositional) stochastic optimization, for which generalization bounds have been studied for different structural assumptions (Hu et al., 2020; Yang et al., 2024).

Statistical consistency itself is a fundamental topic in SLT (Lin, 2004; Zhang, 2004; Bartlett et al., 2006), including extensions to top-$k$ classification (Fan et al., 2017; Yang & Koyejo, 2020; Zhu et al., 2023) and ranking losses (Calauzènes et al., 2013; Gao & Zhou, 2014). Our work differs from this literature in three key respects: (i) we study consistency in the context of modern foundation models trained via contrastive pretraining; (ii) our downstream task is retrieval rather than label prediction; and (iii) the performance measure $\mathcal{E}(s)$ has a compositional, pairwise ranking structure over triples $(\mathbf{x}, \mathbf{y}, \mathbf{y}')$, going beyond the standard instance–label setting.

*Table 1.* Summary of main notations used in this paper.

| Symbol | Description |
| --- | --- |
| $\mathcal{X}, \mathcal{Y}$ | Two data spaces of views or modalities |
| $(\mathbf{x}, \mathbf{y})$ | A data pair with $\mathbf{x} \in \mathcal{X}, \mathbf{y} \in \mathcal{Y}$ |
| $p$ | Joint distribution of $(\mathbf{x}, \mathbf{y})$ |
| $p_{\mathcal{X}}, p_{\mathcal{Y}}$ | Marginal distributions for $\mathcal{X}, \mathcal{Y}$. |
| $p_{\mathbf{x}}^{+}(\mathbf{y})$ | Distribution for positive example |
| $p_{\mathbf{x}}^{-}(\mathbf{y})$ | Distribution for negative example |
| $z$ | A binary variable indicating the relevance between $\mathbf{x}$ and $\mathbf{y}$ |
| $\mathcal{L}$ | Population risk for general contrastive representation learning |
| $\mathcal{L}^{*}$ | $\inf_{s} \mathcal{L}(s)$ over all measurable $s : \mathcal{X} \times \mathcal{Y} \to \mathbb{R}$ |
| $\mathcal{L}_{\text{S}}$ | Population risk for supervised contrastive representation learning |
| $\widehat{\mathcal{L}}_{\text{S}}$ | Empirical risk for supervised contrastive representation learning |
| $\mathcal{L}_{\text{SS}}$ | Population risk for self-supervised contrastive representation learning |
| $\widehat{\mathcal{L}}_{\text{SS}}$ | Empirical risk for self-supervised contrastive representation learning |
| $\mathbb{I}$ | The indicator function such that an event $E$ holds true if $\mathbb{I}[E] = 1$ |
| $\mathcal{E}$ | The AUC-type retrieval measure |
| $\mathcal{E}^{*}$ | $\sup_{s} \mathcal{E}(s)$ over all measurable $s : \mathcal{X} \times \mathcal{Y} \to \mathbb{R}$ |

## 2. Problem Formulation

We illustrate the CRL framework that encompasses both supervised and self-supervised settings. Let $\mathcal{X}, \mathcal{Y}$ be two data spaces of views or modalities. Denote by $(\mathbf{x}, \mathbf{y})$ a data pair with $\mathbf{x} \in \mathcal{X}$ and $\mathbf{y} \in \mathcal{Y}$. Given an anchor $\mathbf{x}$, we refer to $\mathbf{y}$ as positive if it is semantically relevant to $\mathbf{x}$, and negative otherwise. We denote by $p_{\mathbf{x}}^{+}$ and $p_{\mathbf{x}}^{-}$ the conditional distributions of positive and negative samples given $\mathbf{x}$, respectively. Let $p$ be the joint distribution of $\mathbf{x}, \mathbf{y}$ and $p_{\mathcal{X}}, p_{\mathcal{Y}}$ be the marginal distributions of $\mathcal{X}, \mathcal{Y}$, respectively. Denote by $\mathbb{I}$ the indicator function such that an event $E$ holds true if $\mathbb{I}[E] = 1$.

Depending on how $p_{\mathbf{x}}^{+}$ and $p_{\mathbf{x}}^{-}$ are defined and consequently how the empirical risk is constructed, we obtain two distinct instantiations of CRL: supervised and self-supervised learning settings.

### 2.1. Supervised CRL (SCRL)

In the supervised learning setting, relevance between $\mathbf{x}$ and $\mathbf{y}$ is explicitly indicated by a binary label $z \in \{-1, 1\}$, where $z = 1$ signifies that $\mathbf{y}$ is positive for $\mathbf{x}$, and $z = -1$ indicates negativity. Accordingly, we define the positive and

negative conditional distributions as

$$p_{\mathbf{x}}^+(\mathbf{y}) = p_+(\mathbf{y}|\mathbf{x}) := p(\mathbf{y}|\mathbf{x}, z = 1),$$
$$p_{\mathbf{x}}^-(\mathbf{y}) = p_-(\mathbf{y}|\mathbf{x}) := p(\mathbf{y}|\mathbf{x}, z = -1). \tag{4}$$

By the law of total probability, the marginal conditional distribution of $\mathbf{y}$ given $\mathbf{x}$ decomposes as

$$p(\mathbf{y}|\mathbf{x}) = p(z = 1|\mathbf{x})\, p_+(\mathbf{y}|\mathbf{x}) + p(z = -1|\mathbf{x})\, p_-(\mathbf{y}|\mathbf{x}).$$

The population risk for SCRL is then defined as

$$\mathcal{L}_{\mathrm{S}}(s_{\mathbf{w}}) \tag{5}$$
$$= \mathbb{E}_{\mathbf{x}}\mathbb{E}_{\mathbf{y}\sim p_+(\cdot|\mathbf{x})}\Big[\tau \log\mathbb{E}_{\mathbf{y}'\sim p_-(\cdot|\mathbf{x})} \exp\Big(\frac{\Delta_{\mathbf{w}}(\mathbf{x}, \mathbf{y}, \mathbf{y}')}{\tau}\Big)\Big].$$

To estimate this risk from data, we draw $n$ i.i.d. anchor points $\mathbf{x}_i \sim p_{\mathcal{X}}$ for $i \in [n]$. For each anchor $\mathbf{x}_i$, we sample one positive example $\mathbf{y}_i \sim p_+(\cdot|\mathbf{x}_i)$ and then $m$ negative examples $\{\mathbf{y}'_{ij} \sim p_-(\cdot|\mathbf{x}_i) : j \in [m]\}$. The model parameters $\mathbf{w}$ are learned by minimizing the following empirical risk

$$\widehat{\mathcal{L}}_{\mathrm{S}}(s_{\mathbf{w}}) = \frac{1}{n}\sum_{i=1}^{n}\tau \log\Big(\frac{1}{m}\sum_{j=1}^{m}\exp\Big(\frac{\Delta_{\mathbf{w}}(\mathbf{x}_i, \mathbf{y}_i, \mathbf{y}'_{ij})}{\tau}\Big)\Big). \tag{6}$$

**Multi-class classification.** Our general framework takes multi-class setting as a special case(Saunshi et al., 2019; Lei et al., 2023; Hieu & Ledent, 2025). Let $\mathcal{Y} = \mathcal{X}$ and $\mathcal{C}$ be the finite set of all classes. Suppose that $\rho$ is the discrete probability measure over $\mathcal{C}$. For any class $c \in \mathcal{C}$, we denote $\mathcal{D}_c$ as the class-conditional distribution of input vectors over $\mathcal{X}$ given that the vectors belong to class $c$. On the other hand, we define $\bar{\mathcal{D}}_c$ as the distribution of input vectors in $\mathcal{X}$ given that the vectors do not belong to class $c$:

$$\bar{\mathcal{D}}_c(\mathbf{x}) = \frac{\sum_{c' \in \mathcal{C}, c' \neq c}\rho(c')\mathcal{D}_{c'}(\mathbf{x})}{1 - \rho(c)}, \quad \forall \mathbf{x} \in \mathcal{X}.$$

In this case, $z = 1$ when $\mathbf{x}, \mathbf{y}$ have same labels. When the label of $\mathbf{x}$ is $c$, then $p_{\mathbf{x}}^+ = \mathcal{D}_c$ and $p_{\mathbf{x}}^- = \bar{\mathcal{D}}_c$. Then the population of SCRL becomes

$$\mathbb{E}_{c\sim\rho}\mathbb{E}_{\mathbf{x},\mathbf{y}\sim\mathcal{D}_c}\Big[\tau \log\mathbb{E}_{\mathbf{y}'\sim\bar{\mathcal{D}}_c} \exp\Big(\frac{\Delta_{\mathbf{w}}(\mathbf{x}, \mathbf{y}, \mathbf{y}')}{\tau}\Big)\Big].$$

In the empirical risk, the negatives are drawn i.i.d. from $\bar{\mathcal{D}}_c$.

*Remark* 2.1. The empirical risk Eq. (6) is different from InfoNCE, which includes the positive pair in its own denominator. However, the positive pair in the denominator of InfoNCE is arguably not a good feature. The Decoupled Contrastive Learning (DCL) (Yeh et al., 2022) explicitly removes the positive pair in the denominator and has been showed to behave better than InfoNCE. In addition, the difference between InfoNCE and Eq. (6) could be negligible

when the number of negative examples is very large since we average over all negative samples. Furthermore, the formulation of Eq. (6) enjoys a pairwise nature, which is more natural for us to define and analyze the consistency.

*Remark* 2.2. Many existing works take $\mathbb{E}\widehat{\mathcal{L}}_S(s_{\mathbf{w}})$ as the population risk. However, taking Eqs. (1),(5) as population risks facilitates both optimization and generalization analyses. i) Employing the full population of negative samples is well-motivated from an optimization perspective. For example, (Yuan et al., 2022) considers Global Contrastive Objective (GCL), which contrasts each positive pair with all negative pairs for an anchor point. They proposed SogCLR to optimize the objective to achieve better performance than SimCLR that just uses negative data in a mini-batch. ii) Defining Eq. (1) as the population risk also yields a better and meaningful generalization bound than existing work, which will be shown in Section 4 later.

## 2.2. Self-supervised CRL (SSCRL)

In practice, CRL is often deployed in a self-supervised setting. Specifically, given an anchor $\mathbf{x}$, we define the positive and negative conditional distributions as

$$p_{\mathbf{x}}^+(\mathbf{y}) = p(\mathbf{y}|\mathbf{x}), p_{\mathbf{x}}^-(\mathbf{y}) = p_{\mathcal{Y}}(\mathbf{y}). \tag{7}$$

i.e., positive pairs $(\mathbf{x}, \mathbf{y})$ are drawn from the joint distribution $p$, while negative examples $\mathbf{y}'$ are sampled independently from the marginal distribution $p_{\mathcal{Y}}$. For instance, $p$ is the joint distribution of image-text pairs. In augmentation-based contrastive learning, the conditional distribution $p(\cdot|\mathbf{x})$ could be the distribution of augmentation.

Under this construction, the population risk for SSCRL is given by

$$\mathcal{L}_{\mathrm{SS}}(s_{\mathbf{w}}) = \mathbb{E}_{\mathbf{x}}\mathbb{E}_{\mathbf{y}|\mathbf{x}}\tau \log \mathbb{E}_{\mathbf{y}'} \exp\Big(\frac{\Delta_{\mathbf{w}}(\mathbf{x}, \mathbf{y}, \mathbf{y}')}{\tau}\Big). \tag{8}$$

In CLIP model (Radford et al., 2021), a symmetric variant is used by treating $\mathbf{x}$ and $\mathbf{y}$ interchangeably:

$$\mathcal{L}'_{\mathrm{SS}}(s_{\mathbf{w}}) = \mathbb{E}_{\mathbf{x}}\mathbb{E}_{\mathbf{y}|\mathbf{x}}\tau\log\mathbb{E}_{\mathbf{y}'} \exp\Big(\frac{\Delta_{\mathbf{w}}(\mathbf{x}, \mathbf{y}, \mathbf{y}')}{\tau}\Big)$$
$$+ \mathbb{E}_{\mathbf{y}}\mathbb{E}_{\mathbf{x}|\mathbf{y}}\tau \log \mathbb{E}_{\mathbf{x}'} \exp\Big(\frac{\Delta_{\mathbf{w}}(\mathbf{y}, \mathbf{x}, \mathbf{x}')}{\tau}\Big). \tag{9}$$

It has been shown that minimizing $\mathcal{L}'_{\mathrm{SS}}(s)$ over all measurable scoring functions $s : \mathcal{X} \times \mathcal{Y} \to \mathbb{R}$ is equivalent to maximizing the mutual information between $\mathbf{x}$ and $\mathbf{y}$ (Zhang et al., 2023).

To get the empirical version of Eq. (8), a typical choice is mini-batch based InfoNCE loss (Chen et al., 2020; Radford et al., 2021). However, it only uses negative samples

within a mini-batch and thus requires a large batch size. In this paper, we follow the global contrastive loss (GCL) studied in Yuan et al. (2022), where each anchor point is contrasted with all negative points and advanced compositional optimization algorithm can be employed using only mini-batches per-iteration (Wei et al., 2024). We draw $n$ positive pairs $\{(\mathbf{x}_i, \mathbf{y}_i)\}_{i=1}^n \sim p$. Additionally, we sample $m$ negative examples $\{\mathbf{y}'_j\}_{j=1}^m \overset{\text{i.i.d.}}{\sim} p_{\mathcal{Y}}$. Each anchor $\mathbf{x}_i$ is then contrasted against with all $\mathbf{y}'_j, j \in [m]$. The empirical risk is given by

$$\widehat{\mathcal{L}}_{\text{SS}}(s_{\mathbf{w}}) \tag{10}$$
$$= \frac{1}{n}\sum_{i=1}^n \tau \log\left(\frac{1}{m}\sum_{j=1}^m \exp\left(\frac{\Delta_{\mathbf{w}}(\mathbf{x}_i, \mathbf{y}_i, \mathbf{y}'_j))}{\tau}\right)\right).$$

For example, $\{(\mathbf{x}_i, \mathbf{y}_i)\}_{i=1}^n$ may represent $n$ observed image–text pairs, while $\{\mathbf{y}'_1, \ldots, \mathbf{y}'_m\}$ forms a fixed text dictionary. It is worth noting that in this setting, negative samples are *shared* across all anchor data. While supervised CRL treats each anchor independently with its own set of negatives.

*Remark* 2.3. In practice, the negative examples of one term may be positive for another term. Similar independent assumptions have been adopted in existing theoretical work of CRL (Saunshi et al., 2019; Lei et al., 2023). How to make more realistic assumptions is generally an important problem in learning theory.

## 3. Statistical Consistency of CRL

In this section, we show that a scoring function learned via CRL generalizes well to downstream tasks. Specifically, we establish the *Fisher consistency* and *statistical consistency* of CRL, together with a *calibration-type inequality*, following the terminology of SLT developed for binary classification (Bartlett et al., 2006; Lin, 2004; Zhang, 2004). Leveraging this calibration-type inequality, we derive the explicit bounds for the excess risks for downstream retrieval tasks.

In Appendix E, we establish statistical consistency and calibration-type inequalities for a general CRL framework based on optimized certainty equivalent (OCE) losses (Ben-Tal & Teboulle, 2007), which instantiates the log-sum-exp loss used in standard CRL Eq. (1) as a special case.

### 3.1. Fisher consistency

We first establish the Fisher consistency of CRL,

**Theorem 3.1.** $\mathcal{L}(s)$ *is statistically consistent with* $\mathcal{E}(s)$*, i.e., a minimizer of* $\mathcal{L}(s)$ *is a maximizer of* $\mathcal{E}(s)$*.*

The proof of Theorem 3.1 is based on the following two lemmas.

**Lemma 3.2** (Characterization of minimizers of $\mathcal{L}$)**.** *Let* $\mathcal{L}(s)$ *be defined as in Eq.* (1)*. Then a measurable function* $s : \mathcal{X} \times \mathcal{Y} \to \mathbb{R}$ *minimizes* $\mathcal{L}(s)$ *if and only if it satisfies*

$$s(\mathbf{x}, \mathbf{y}) = \tau \log \frac{p_{\mathbf{x}}^+(\mathbf{y})}{p_{\mathbf{x}}^-(\mathbf{y})} + g(\mathbf{x}), \tag{11}$$

*for some measurable function* $g : \mathcal{X} \to \mathbb{R}$*.*

The proof of Lemma 3.2 is deferred to Appendix B.1.

**Lemma 3.3** (Characterization of maximizers of $\mathcal{E}$)**.** *Let* $\mathcal{E}(s)$ *be defined as in Eq.* (2)*. A measurable scoring function* $s : \mathcal{X} \times \mathcal{Y} \to \mathbb{R}$ *maximizes* $\mathcal{E}(s)$ *if and only if, for any* $\mathbf{x}, \mathbf{y}, \mathbf{y}'$ *with* $p_{\mathbf{x}}^+(\mathbf{y})/p_{\mathbf{x}}^-(\mathbf{y}) \neq p_{\mathbf{x}}^+(\mathbf{y}')/p_{\mathbf{x}}^-(\mathbf{y}')$*, there holds*

$$\left(\frac{p_{\mathbf{x}}^+(\mathbf{y})}{p_{\mathbf{x}}^-(\mathbf{y})} - \frac{p_{\mathbf{x}}^+(\mathbf{y}')}{p_{\mathbf{x}}^-(\mathbf{y}')}\right)(s(\mathbf{x}, \mathbf{y}) - s(\mathbf{x}, \mathbf{y}')) > 0.$$

The proof of Lemma 3.3 is deferred to Appendix B.2. From Lemma 3.2 we know that a minimizer of the contrastive loss $\mathcal{L}(s)$ is given by Eq. (11) for some $g : \mathcal{X} \to \mathbb{R}$. Since $\log(t)$ is an increasing function of $t$ over $\mathbb{R}$, it is easy to see that for $p_{\mathbf{x}}^+(\mathbf{y})/p_{\mathbf{x}}^-(\mathbf{y}) \neq p_{\mathbf{x}}^+(\mathbf{y}')/p_{\mathbf{x}}^-(\mathbf{y}')$, there holds

$$\left(\frac{p_{\mathbf{x}}^+(\mathbf{y})}{p_{\mathbf{x}}^-(\mathbf{y})} - \frac{p_{\mathbf{x}}^+(\mathbf{y}')}{p_{\mathbf{x}}^-(\mathbf{y}')}\right)(s(\mathbf{x}, \mathbf{y}) - s(\mathbf{x}, \mathbf{y}'))$$
$$= \tau\left(\frac{p_{\mathbf{x}}^+(\mathbf{y})}{p_{\mathbf{x}}^-(\mathbf{y})} - \frac{p_{\mathbf{x}}^+(\mathbf{y}')}{p_{\mathbf{x}}^-(\mathbf{y}')}\right)\left(\log\frac{p_{\mathbf{x}}^+(\mathbf{y})}{p_{\mathbf{x}}^-(\mathbf{y})} - \log\frac{p_{\mathbf{x}}^+(\mathbf{y}')}{p_{\mathbf{x}}^-(\mathbf{y}')}\right) > 0$$

and $s$ is therefore *indeed a maximizer* of the AUC-type functional $\mathcal{E}(s)$ according to Lemma 3.3. Consequently, every minimizer attaining $\inf_s \mathcal{L}(s)$ also guarantees to attain $\sup_s \mathcal{E}(s)$. In the terminology of statistical learning theory, this establishes the *Fisher consistency* of the contrastive loss with respect to the downstream ranking objective $\mathcal{E}(s)$; see, e.g., Bartlett et al. (2006); Lin (2004) for related discussions in the context of binary classification.

### 3.2. Calibration-style inequality

Theorem 3.1 establishes a qualitative notion of Fisher consistency. We next strengthen this result by deriving a quantitative relationship between the excess risk of the upstream contrastive objective and the excess risk of downstream performance.

The quantitative relationship between the excess risk of the upstream CRL objective and that of the downstream task is captured by the following calibration-style inequality. The proof of the theorem is provided in Appendix B.3.

**Theorem 3.4.** *For any scoring function* $s : \mathcal{X} \times \mathcal{Y} \to \mathbb{R}$*, there holds*

$$\mathcal{E}^* - \mathcal{E}(s) \leq \sqrt{2/\tau(\mathcal{L}(s) - \mathcal{L}^*)}.$$

*Remark* 3.5. Our general statistical consistency results for CRL naturally extend to both supervised (SCRL) and self-supervised (SSCRL) settings. Our statistical consistency theorems are established for general positive and negative distributions , so it applies to the specialized distributions used in both SCRL and SSCRL, which differ in how $p_{\mathbf{x}}^+$ and $p_{\mathbf{x}}^-$ are specified (see Eqs. (4) and (7)).

*Remark* 3.6. Recently, Ryu et al. (2025) have established Fisher consistency for InfoNCE loss. While we derive a quantitative calibraion-type inequality for CRL for the first time, which is practically important since the ideal optimal scoring function is often unattainable in real training. Some work shows that contrastive loss induces Gaussian distribution (Betser et al., 2026). We view our results and Gaussianity as complementary: our theory provides the statistical guarantee that optimal CRL leads to optimal retrieval performance, while Gaussianity reveals the latent structure that enables such good generalization.

## 4. Generalization Analysis for CRL

In this section, we analyze the generalization gap of contrastive representation learning (CRL) under both supervised and self-supervised learning objectives. To this end, we make the following assumption.

**Assumption 4.1.** Suppose that for any $\mathbf{x} \in \mathcal{X}, \mathbf{y} \in \mathcal{Y}$, $\|\mathbf{x}\|_2, \|\mathbf{y}\|_2 \leq 1$, $s_{\mathbf{w}}(\mathbf{x}, \mathbf{y}) = f_{\mathbf{w}}(\mathbf{x})^\top f_{\mathbf{w}}(\mathbf{y})$, where $f_{\mathbf{w}}(\mathbf{x})$ is a deep neural network. Specifically, let $W_l \in \mathbb{R}^{d_l \times d_{l-1}}, 1 \leq l \leq L$ be matrices, $\sigma$ be ReLU activation function. Then we define

$$f_{\mathbf{w}}(\mathbf{x}) = \sigma(W_L \sigma(\cdots \sigma(W_1 \mathbf{x}) \cdots)).$$

The hypothesis space $\mathcal{W}$ is the following space of matrices

$$\mathcal{W} = \{W_l \in \mathbb{R}^{d_l \times d_{l-1}} : \|W_l\|_2 \leq s_l, \forall l \in [L]\},$$

where $\|\cdot\|_2$ denotes the spectral norm. The hypothesis space of functions is given by

$$\mathcal{F} = \{\mathbf{x} \mapsto f_{\mathbf{w}}(\mathbf{x}) : \mathbf{w} \in \mathcal{W}\}.$$

In this case, $s_{\mathbf{w}}(\mathbf{x}, \mathbf{y}) \in [-B, B]$ with $B = \Pi_{l=1}^L s_l^2$.

### 4.1. Generalization analysis for SCRL

In this subsection, we focus on the generalization analysis of supervised contrastive representation learning (SCRL).

We aim to control the uniform bound for the generalization gap $\sup_{\mathbf{w} \in \mathcal{W}} |\mathcal{L}_{\mathrm{S}}(s_{\mathbf{w}}) - \widehat{\mathcal{L}}_{\mathrm{S}}(s_{\mathbf{w}})|$, where $\mathcal{L}_{\mathrm{S}}(s_{\mathbf{w}})$ and $\widehat{\mathcal{L}}_{\mathrm{S}}(s_{\mathbf{w}})$ are defined in Eqs. (5) and (6), respectively. A key observation is that the contrastive loss formulations in Eqs. (5) and (6) exhibit a *compositional structure*: the *outer* structure centers on positive sample pairs $(\mathbf{x}, \mathbf{y})$, while the *inner* structure is formed by the negative samples used to

contrast against each positive pair. This compositional property motivates us to decompose the generalization gap into inner and outer components, which we analyze separately. In particular, we have

$$\left|\mathcal{L}_{\mathrm{S}}(s_{\mathbf{w}}) - \widehat{\mathcal{L}}_{\mathrm{S}}(s_{\mathbf{w}})\right|$$
$$\leq \underbrace{\left|\mathcal{L}_{\mathrm{S}}(s_{\mathbf{w}}) - \mathbb{E}\widehat{\mathcal{L}}_{\mathrm{S}}(s_{\mathbf{w}})\right|}_{\text{inner error}} + \underbrace{\left|\mathbb{E}\widehat{\mathcal{L}}_{\mathrm{S}}(s_{\mathbf{w}}) - \widehat{\mathcal{L}}_{\mathrm{S}}(s_{\mathbf{w}})\right|}_{\text{outer error}}. \quad (12)$$

Here, we have

$$\mathcal{L}_{\mathrm{S}}(s_{\mathbf{w}}) - \mathbb{E}\widehat{\mathcal{L}}_{\mathrm{S}}(s_{\mathbf{w}})$$
$$= \mathbb{E}_{\mathbf{x}, \mathbf{y}, \{\mathbf{y}'_j\}} \left[ \tau \log \mathbb{E}_{\mathbf{y}'} \exp\left(\frac{\Delta_{\mathbf{w}}(\mathbf{x}, \mathbf{y}, \mathbf{y}')}{\tau}\right) \right.$$
$$\left. - \tau \log \frac{1}{m} \sum_{j=1}^m \exp\left(\frac{\Delta_{\mathbf{w}}(\mathbf{x}, \mathbf{y}, \mathbf{y}'_j)}{\tau}\right) \right],$$

where we omit the explicit specification of the underlying distributions for simplicity.

The inner error quantifies the bias introduced by using $m$ negative samples instead of the full population of negative examples.

**Estimation of the inner error.** One approach is to apply uniform convergence theory, which usually leads to a bound that scales with $1/\sqrt{m}$ (Lee et al., 2020). However, we could derive a more refined bound for the inner error by exploring the intrinsic structure of the log-sum-exponential loss in CRL. To be more specific, we can reformulate contrastive losses as minimization problems in the following lemma. The proof is deferred to Appendix C.1.

**Lemma 4.2.** *Let Assumption 4.1 hold, then we have*

$$\widehat{\mathcal{L}}_{\mathrm{S}}(s_{\mathbf{w}}) = -\tau +$$
$$\frac{1}{n} \sum_{i=1}^n \left[ \min_{|\mu_i| \leq 2B} \frac{\tau}{m} \sum_{j=1}^m \exp\left(\frac{\Delta(\mathbf{x}_i, \mathbf{y}_i, \mathbf{y}'_{ij}) - \mu_i}{\tau}\right) + \mu_i \right],$$
$$\quad (13)$$

$$\mathcal{L}_{\mathrm{S}}(s_{\mathbf{w}}) = -\tau +$$
$$\mathbb{E}_{\mathbf{x}, \mathbf{y}} \left[ \min_{|\mu| \leq 2B} \tau \mathbb{E}_{\mathbf{y}'} \exp\left(\frac{\Delta_{\mathbf{w}}(\mathbf{x}, \mathbf{y}, \mathbf{y}') - \mu}{\tau}\right) + \mu \right]. \quad (14)$$

We denote $\psi_{\mathbf{w}, \mathbf{x}, \mathbf{y}}(\mu; \mathbf{y}') = \mu + \tau[\exp((\Delta_{\mathbf{w}}(\mathbf{x}, \mathbf{y}, \mathbf{y}') - \mu)/\tau)]$, which is a strongly convex function for $\mu$ in the bounded domain $[-2B, 2B]$. Then we have

$$\mathcal{L}_{\mathrm{S}}(s_{\mathbf{w}}) - \mathbb{E}\widehat{\mathcal{L}}_{\mathrm{S}}(s_{\mathbf{w}}) = \mathbb{E}_{\mathbf{x}, \mathbf{y}, \{\mathbf{y}'_j\}} \left[ f(\mathbf{w}, \mathbf{x}, \mathbf{y}) - \hat{f}(\mathbf{w}, \mathbf{x}, \mathbf{y}) \right],$$

where $f(\mathbf{w}, \mathbf{x}, \mathbf{y}) = \min_{|\mu| \leq 2B} \mathbb{E}_{\mathbf{y}'} \psi_{\mathbf{w}, \mathbf{x}, \tau}(\mu; \mathbf{y}')$ and $\hat{f}(\mathbf{w}, \mathbf{x}, \mathbf{y}) = \min_{|\mu| \leq 2B} \frac{1}{m} \sum_{j=1}^m \psi_{\mathbf{w}, \mathbf{x}, \tau}(\mu; \mathbf{y}'_j)$.

It can be regarded as the generalization error of empirical risk minimization (ERM). Leveraging the algorithmic stability approach (Bousquet & Elisseeff, 2002), we derive an inner error bound of order $O(1/m)$. This result is formalized in the following lemma:

**Lemma 4.3.** *Let Assumption 4.1 hold. Then we have* $\sup_{\mathbf{w} \in \mathcal{W}} \left| \mathcal{L}(s_\mathbf{w}) - \mathbb{E}\widehat{\mathcal{L}}_S(s_\mathbf{w}) \right| \leq \frac{2 \exp(8B/\tau)\tau}{m}$.

The proof of Lemma 4.3 is provided in Appendix C.2.

**Estimation of the outer error.** The outer error can be written as

$$
\mathbb{E}\widehat{\mathcal{L}}_S(s_\mathbf{w}) - \widehat{\mathcal{L}}_S(s_\mathbf{w})
$$
$$
= \mathbb{E}_{\mathbf{x},\mathbf{y},\{\mathbf{y}'_j\}} \left[ \tau \log \frac{1}{m} \sum_{j=1}^m \exp\left( \frac{\Delta_\mathbf{w}(\mathbf{x},\mathbf{y},\mathbf{y}'_j)}{\tau} \right) \right]
$$
$$
- \frac{1}{n} \sum_{i=1}^n \tau \log \frac{1}{m} \sum_{j=1}^m \exp\left( \frac{\Delta_\mathbf{w}(\mathbf{x}_i,\mathbf{y}_i,\mathbf{y}'_{ij})}{\tau} \right).
$$

It mainly focuses on the generalization of the outer structure, capturing how would the number of anchor points affect the generalization performance.

For $\mathbf{w} \in \mathcal{W}$, we introduce $k_\mathbf{w} : \mathcal{X} \times \mathcal{Y}^{m+1} \to \mathbb{R}$ as follows:

$$
k_\mathbf{w}(\mathbf{x},\mathbf{y},\mathbf{y}'_1,\cdots,\mathbf{y}'_m) = \tau \log \frac{1}{m} \sum_{j=1}^m \exp\left( \frac{\Delta_\mathbf{w}(\mathbf{x},\mathbf{y},\mathbf{y}'_j)}{\tau} \right). \tag{15}
$$

Then we can bound the outer error through Rademacher complexity $\mathfrak{R}_S(\mathcal{K})$ (Bartlett & Mendelson, 2002), where

$$
\mathcal{K} = \Big\{ (\mathbf{x},\mathbf{y},\mathbf{y}'_1,\cdots,\mathbf{y}'_m)
$$
$$
\mapsto k_\mathbf{w}(\mathbf{x},\mathbf{y},\mathbf{y}'_1,\cdots,\mathbf{y}'_m) : \mathbf{w} \in \mathcal{W} \Big\}
$$

and $S = \left\{ (\mathbf{x}_i,\mathbf{y}_i,\mathbf{y}'_{i1},\cdots,\mathbf{y}'_{im}) : i \in [n] \right\}$.

Following standard techniques (Long & Sedghi, 2019; Graf et al., 2022), we establish the following outer error bound:

**Lemma 4.4.** *Let Assumption 4.1 hold. Then with probability at least $1 - \delta$, we have*

$$
\sup_{\mathbf{w} \in \mathcal{W}} \left| \widehat{\mathcal{L}}_S(s_\mathbf{w}) - \mathbb{E}\widehat{\mathcal{L}}_S(s_\mathbf{w}) \right| \lesssim
$$
$$
\frac{B\sqrt{\log(1/\delta)} + B\sqrt{\log(1 + 8nL\Pi_{l=1}^L s_l^2) \sum_{l=1}^L d_l d_{l-1}}}{\sqrt{n}}.
$$

The proof of Lemma 4.4 is provided in Appendix C.3.

**Combining Inner Error and Outer Error** Plugging Lemma 4.3, 4.4 into Eq. (12), we obtain

**Theorem 4.5.** *Let Assumption 4.1 hold. Then with probability at least $1 - \delta$, there holds*

$$
\sup_{\mathbf{w} \in \mathcal{W}} \left| \mathcal{L}_S(s_\mathbf{w}) - \widehat{\mathcal{L}}_S(s_\mathbf{w}) \right| \lesssim \frac{2\exp(8B/\tau)\tau}{m} +
$$
$$
\frac{B\sqrt{\log(1/\delta)} + B\sqrt{\log(1 + 8nL\Pi_{l=1}^L s_l^2) \sum_{l=1}^L d_l d_{l-1}}}{\sqrt{n}}.
$$

Below, we discuss the implication of the above theoretical results and compare them with existing works.

*Firstly*, most existing works analyze the generalization performance of CRL through the outer error (Saunshi et al., 2019; Lei et al., 2023; Hieu et al., 2025). These analyses typically yield a bound of $\widetilde{O}(1/\sqrt{n})$ that deteriorates as the number of negative samples $m$ increases. In contrast, our theory demystifies the role of $m$ in CRL by more refined error decomposition which involves the inner error. We show that the generalization error initially *decreases* as the number of negative samples $m$ grows, revealing the benefit of large $m$ and aligning with empirical practice. Moreover, our results $\widetilde{O}(1/m + 1/\sqrt{n})$ uncover an explicit trade-off between $m$ and $n$ in the sense of relative contribution of each term to the total error: the generalization error is bottlenecked by the larger term. When $m \gg \sqrt{n}$, the $1/\sqrt{n}$ term dominates, and further increasing $m$ yields diminishing returns, leading to the observed saturation. Further gains therefore require increasing the number of anchor points rather than merely adding more negatives. We conduct experiments to validate this theoretical insight in Section 5.

*Secondly*, the problem of bounding inner error is similar to generalization error of risk-averse learning in the literature (Lee et al., 2020), which applied uniform convergence arguments to obtain a $1/\sqrt{m}$ bound. Wang & Isola (2020) used the inner error to analyze uniformity and alignment in CRL, deriving a rate of $m^{-2/3}$. More recently, Oko et al. (2025) estimated the inner error to show that near-minimizers of CLIP are near-sufficient statistics; their Taylor-expansion-based analysis yields a $1/m$ rate, comparable to ours.

In contrast, our analysis is grounded in algorithmic stability and extends to a more general class of contrastive losses, as detailed in Section E.5. For the classical CRL with the standard log-sum-exp loss, this approach yields a sharp $O(1/m)$ bound on the inner error. More generally, for broad classes of OCE and nonlinear pairwise losses $\ell$, we obtain an $O(1/\sqrt{m})$ rate, highlighting both the strength of our method in the CRL setting and its versatility beyond the standard contrastive losses.

### 4.2. Generalization analysis for SSCRL

In this subsection, we turn to the generalization analysis of CRL in the self-supervised learning setting (SS-

CRL) where the population and empirical risks are given by (8), (10), respectively. Our objective is to bound $\sup_{\mathbf{w} \in \mathcal{W}} \left| \mathcal{L}_{\mathrm{SS}}(s_{\mathbf{w}}) - \widehat{\mathcal{L}}_{\mathrm{SS}}(s_{\mathbf{w}}) \right|$. A distinguished feature of SSCRL is that negative examples are shared by anchor points. Specifically, let $k_{\mathbf{w}}$ be defined as in Eq. (15), we have

$$\widehat{\mathcal{L}}_{\mathrm{SS}}(s_{\mathbf{w}}) = \frac{1}{n} \sum_{i=1}^{n} k_{\mathbf{w}}(\mathbf{x}_i, \mathbf{y}_i, \mathbf{y}'_1, \cdots, \mathbf{y}'_m).$$

Therefore, $k_{\mathbf{w}}(\mathbf{x}_i, \mathbf{y}_i, \mathbf{y}'_1, \cdots, \mathbf{y}'_m)$ share the same negative set $\{\mathbf{y}'_j\}$ and are thus dependent. This is different from SCRL. Because of this key difference, the previous analysis no longer works. We will resort to a different analysis for deriving the generalization bound. We define

$$h_{\mathbf{w}}(\mathbf{x}, \mathbf{y}) = \tau \log \mathbb{E}_{\mathbf{y}' \sim p_{\mathcal{Y}}} \exp(\Delta_{\mathbf{w}}(\mathbf{x}, \mathbf{y}, \mathbf{y}')/\tau),$$
$$\hat{h}_{\mathbf{w}}(\mathbf{x}, \mathbf{y}) = \tau \log \left( \frac{1}{m} \sum_{j=1}^{m} \exp \left( \frac{\Delta_{\mathbf{w}}(\mathbf{x}, \mathbf{y}, \mathbf{y}'_j)}{\tau} \right) \right),$$

which compares $(\mathbf{x}, \mathbf{y})$ with all negative examples and $m$ negative examples $\{\mathbf{y}'_j\}$, respectively. Therefore,

$$\mathcal{L}_{\mathrm{SS}}(s_{\mathbf{w}}) = \mathbb{E}_{\mathbf{x},\mathbf{y}} h_{\mathbf{w}}(\mathbf{x}, \mathbf{y}), \quad \widehat{\mathcal{L}}_{\mathrm{SS}}(s_{\mathbf{w}}) = \frac{1}{n} \sum_{i=1}^{n} \hat{h}_{\mathbf{w}}(\mathbf{x}_i, \mathbf{y}_i)$$

and we can make the following error decomposition:

$$|\mathcal{L}_{\mathrm{SS}}(s_{\mathbf{w}}) - \widehat{\mathcal{L}}_{\mathrm{SS}}(s_{\mathbf{w}})| = \left| \mathbb{E}_{\mathbf{x},\mathbf{y}} h_{\mathbf{w}}(\mathbf{x}, \mathbf{y}) - \frac{1}{n} \sum_{i=1}^{n} \hat{h}_{\mathbf{w}}(\mathbf{x}_i, \mathbf{y}_i) \right|$$
$$\leq \underbrace{\left| \mathbb{E}_{\mathbf{x},\mathbf{y}} h_{\mathbf{w}}(\mathbf{x}, \mathbf{y}) - \frac{1}{n} \sum_{i=1}^{n} h_{\mathbf{w}}(\mathbf{x}_i, \mathbf{y}_i) \right|}_{\text{outer error}}$$
$$+ \underbrace{\left| \frac{1}{n} \sum_{i=1}^{n} \left( h_{\mathbf{w}}(\mathbf{x}_i, \mathbf{y}_i) - \hat{h}_{\mathbf{w}}(\mathbf{x}_i, \mathbf{y}_i) \right) \right|}_{\text{inner error}}.$$

(16)

**Estimation of the outer error.** The outer error focuses on the generalization of the outer structure, i,e., the concentration property of $h_{\mathbf{w}}(\mathbf{x}, \mathbf{y})$ on $n$ positive pairs $(\mathbf{x}_i, \mathbf{y}_i)$. Leveraging standard uniform convergence arguments based on Rademacher complexity (Bartlett & Mendelson, 2002), we derive a bound for this component that scales with $1/\sqrt{n}$.

**Estimation of the inner error.** For each positive pair $(\mathbf{x}, \mathbf{y})$, $h_{\mathbf{w}}(\mathbf{x}, \mathbf{y}) - \hat{h}_{\mathbf{w}}(\mathbf{x}, \mathbf{y})$ reflects the bias caused by sampling $m$ negative examples $\{\mathbf{y}'_j\}$ instead of all negative data, thus characterizing the error arising from the inner structure, consistent with the core intuition of the inner error in SCRL (see Eq. (12)). A crucial distinction, however, is that the inner error in SCRL is a *population-level* expectation. In contrast, the inner error here is an *empirical* quantity, depending on

both the $n$ positive pairs and a shared set of $m$ negative samples. By the independence between the negative samples $\mathbf{y}'_j$ and the positive pairs $(\mathbf{x}_i, \mathbf{y}_i)$, we can estimate the inner error by analyzing the term $h_{\mathbf{w}}(\mathbf{x}_i, \mathbf{y}_i) - \hat{h}_{\mathbf{w}}(\mathbf{x}_i, \mathbf{y}_i)$ for each $i \in [n]$ independently. Consequently, we derive a bound for inner error that scales with $1/\sqrt{m}$.

Overall, the generalization gap for SSCRL is stated as follows, the detailed proof is deferred to Appendix D .

**Theorem 4.6.** *Let Assumption 4.1 hold. We have with probability at least $1 - \delta$,*

$$\sup_{\mathbf{w} \in \mathcal{W}} \left| \mathcal{L}_{\mathrm{SS}}(s_{\mathbf{w}}) - \widehat{\mathcal{L}}_{\mathrm{SS}}(s_{\mathbf{w}}) \right| \lesssim B\sqrt{\frac{\log(1/\delta)}{n}} +$$
$$B\sqrt{\frac{\log(1 + 8\exp(4B/\tau)nL\Pi_{l=1}^{L} s_l^2) \sum_{l=1}^{L} d_l d_{l-1}}{n}} +$$
$$\frac{\exp(4B/\tau)B\sqrt{\log(1 + 8mL\Pi_{l=1}^{L} s_l^2) \sum_{l=1}^{L} d_l d_{l-1}}}{\sqrt{m}} +$$
$$\frac{\exp(4B/\tau)\tau\sqrt{\log(n/\delta)}}{\sqrt{m}}.$$

*Remark* 4.7. CRL has a compositional structure analogous to conditional stochastic optimization (CSO), for which sample complexity bounds are well-established in Hu et al. (2020). The conditional and independent sampling strategies correspond directly to our SCRL and SSCRL settings, with qualitatively similar sample complexity results derived. Our work, however, has key advantages: First, CRL's log-sum-exp loss features a more complex compositional structure, requiring non-trivial technical treatments absent in standard CSO analysis. Second, we adopt distinct techniques from Hu et al. (2020)—notably, we leverage algorithmic stability to analyze the inner error of SCRL.

**Application to retrieval and zero-shot classification** Our generalization analysis (Theorem 4.5,4.6) can be used to show that minimizing the empirical CRL risk would generalize well in the downstream tasks. Let $\hat{\mathbf{w}} = \arg\min_{\mathbf{w} \in \mathcal{W}} \widehat{\mathcal{L}}_{\mathrm{S}}(s_{\mathbf{w}})$, A standard decomposition yields

$$\mathcal{L}_{\mathrm{S}}(s_{\hat{\mathbf{w}}}) - \inf_s \mathcal{L}_{\mathrm{S}}(s) \leq 2\underbrace{\sup_{\mathbf{w} \in \mathcal{W}} \left| \mathcal{L}_{\mathrm{S}}(s_{\mathbf{w}}) - \widehat{\mathcal{L}}_{\mathrm{S}}(s_{\mathbf{w}}) \right|}_{\text{generalization gap}}$$
$$+ \underbrace{\inf_{\mathbf{w} \in \mathcal{W}} \mathcal{L}_{\mathrm{S}}(s_{\mathbf{w}}) - \inf_s \mathcal{L}_{\mathrm{S}}(s)}_{\text{approximation error}}.$$

When $\mathcal{W}$ is sufficiently expressive, the approximation error is negligible (Oko et al., 2025). In this paper, we mainly focus on the statistical consistency, the generalization gap, and its role in transferring pretraining performance to downstream retrieval performance. Combining the above inequality with Theorem 3.4, 4.5, we can establish the excess risk

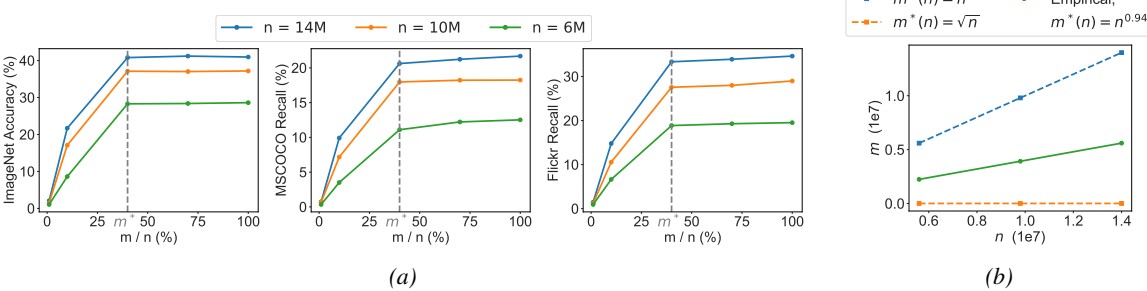

*Figure 1.* (a): Zero-shot classification results on ImageNet (left) and retrieval results on MSCOCO (middle) and Flickr (right) of CLIP training on different sizes of negative samples. $n$ denotes the size of the anchor dataset, while $m$ denotes the size of negative samples. (b): Critical size of $m$ at different $n$, compared with $m = \sqrt{n}$ and $m = n$.

for retrieval $\sup_s \mathcal{E}(s) - \mathcal{E}(s_{\hat{\mathbf{w}}})$. Our generalization gap can be also applied to zero-shot classification downstream task, following the framework in Oko et al. (2025). One can find more details in Section D.3 of the Appendix.

## 5. Empirical Verification

In this section, we conduct experiments to empirically demonstrate the validity of our results in Theorem 4.6. Specifically, we consider the CLIP training problem (Radford et al., 2021). The objective consists of two symmetric parts: in the first part the anchor data are images and each image is compared with a set of negative texts, while in the second part the anchor are texts and each text is contrasted with a set of negative images. We choose different sizes of anchor datasets, i.e., $n \in \{14M, 10M, 6M\}$, all of which are sampled from the DFN dataset (Fang et al., 2024). To control the number of negative samples, we randomly sample $m$ instances from the original dataset and use their images as negatives for all anchor text instances, and their texts as negatives for all anchor image instances. The value of $m$ is varied such that $m/n \in \{1.0, 0.7, 0.4, 0.1, 0.01\}$. For each choice of $n$ and $m$, we train a CLIP model using FastCLIP (Wei et al., 2024) by processing 320M samples in total, then evaluate (1) its zero-shot classification performance on ImageNet (Deng et al., 2009), and (2) its zero-shot retrieval performance on MSCOCO (Chen et al., 2015) and Flickr (Young et al., 2014). The reported metrics are (1) the top-1 accuracy on ImageNet and (2) the average of image recall@1 and text recall@1 on individual retrieval datasets.

We present the results in Figure 1a. From the figure, we observe that performance improves as the number of negative samples $m$ increases, but saturates once $m$ exceeds a certain threshold, referred to as the *critical negative data size*. This behavior is consistent with our generalization results, highlighting the trade-off between $m$ and $n$. In Figure 1b, we plot the critical negative data size of $m^*(n)$ as a function of $n$, and compare it with $m^*(n) = \sqrt{n}$ and $m^*(n) = n$. From the figure we can see that the empirical trend lies between $m^*(n) = \sqrt{n}$ and $m^*(n) = n$ but is

closer to $m^*(n) = n$. We provide a theoretical intuition for this observation based on our derived results.

Theorem 4.6 establishes that $m^* = n$ is the optimal scaling when negative examples are fully shared across all anchor points and independent of positive pairs. In contrast, Theorem 4.5 shows that $m^*$ should scale as $\sqrt{n}$ in the supervised setting, where negative samples are drawn independently for each anchor point. While in our experimental setup, negative samples are shared across anchor points (aligning with Theorem 4.6), the empirical scaling for $m^*$ shows that the generalization in practice indeed has a better dependence than $1/\sqrt{m}$ as predicted by the theory.

## 6. Conclusion

In this paper, we develop a statistical learning theory framework for CRL. We establish the statistical consistency of CRL by proving that minimizing the contrastive loss guarantees optimal performance on downstream retrieval objectives. We also derive a calibration-style inequality that quantifies the quantitative relationship between the excess risk of CRL during pretraining and that of downstream tasks. Additionally, we provide pretraining generalization bounds whose values decrease with the increasing number of negative examples.

Several promising directions for future work emerge from our findings. While we have provided a general framework for CRL based on OCE and pairwise loss functions in the Appendix, it is not known which choices of disutility function $\phi$ and pairwise loss $\ell$ perform best practically and theoretically. Additionally, it would be interesting to adapt our learning theory framework to analyze other foundation models, such as generalization analysis and statistical consistency for large language models (LLMs).

## Acknowledgments

The authors would like to thank anonymous reviewers and area chair for useful comments. T.Y. Yang and X.Y. Wei were partially supported by NSF award 2306572. The work

of Y. Ying and Y. Li was partially supported by the Australian Research Council (ARC) Discovery Project under DP250101359.

## Impact Statement

This paper presents work whose goal is to advance the field of Machine Learning. There are many potential societal consequences of our work, none which we feel must be specifically highlighted here.

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

# A. Rademacher complexity and covering numbers

In this section we introduce some useful facts and lemmas on Rademacher complexity and covering numbers. We first introduce notations of Rademacher complexity. For a hypothesis space $\mathcal{F}$ and a dataset $S = \{\mathbf{x}_1, \cdots, \mathbf{x}_n\}$, the empirical Rademacher complexity is defined as

$$\mathfrak{R}_S(\mathcal{F}) = \mathbb{E}_\epsilon \left[ \sup_{f \in \mathcal{F}} \frac{1}{n} \sum_{i=1}^n \epsilon_i f(\mathbf{x}_i) \right],$$

where $\epsilon = (\epsilon_i)_{i \in [n]} = \{\pm 1\}^n$ are independent Rademacher random variables. The following proposition controls the generalization error through Rademacher complexity.

**Proposition A.1** (Mohri et al. (2018)). *Let $\mathcal{F}$ be a class of functions $f : \mathcal{Z} \to [a, b]$. $\mathcal{D}$ is a distribution over $\mathcal{Z}$. Let $S = \{\mathbf{z}_1, \cdots, \mathbf{z}_n\}$ be i.i.d. sampled from $\mathcal{D}$. Then*

$$\mathbb{E}_S \sup_{f \in \mathcal{F}} \mathbb{E}_{\mathbf{z} \sim \mathcal{D}} f(\mathbf{z}) - \frac{1}{n} \sum_{i=1}^n f(\mathbf{z}_i) \leq 2 \mathbb{E}_S \mathfrak{R}_S(\mathcal{F}).$$

*Furthermore, with probability at least $1 - \delta$, there holds*

$$\sup_{f \in \mathcal{F}} \mathbb{E}_{\mathbf{z} \sim \mathcal{D}} f(\mathbf{z}) - \frac{1}{n} \sum_{i=1}^n f(\mathbf{z}_i) \leq 2 \mathfrak{R}_S(\mathcal{F}) + 3(b - a) \sqrt{\frac{\log(2/\delta)}{2n}}.$$

The following contraction lemma is useful for estimating Rademacher complexity.

**Lemma A.2** (Ledoux & Talagrand (2013)). *Suppose $\psi : \mathbb{R} \mapsto \mathbb{R}$ is contractive (i.e., $|\psi(t) - \psi(t')| \leq \eta |t - t'|$). Then we have*

$$\mathbb{E}_\epsilon \sup_{f \in \mathcal{F}} \sum_{i=1}^n \epsilon_i \psi(f(\mathbf{x}_i)) \leq \eta \mathbb{E}_\epsilon \sup_{f \in \mathcal{F}} \sum_{i=1}^n \epsilon_i f(\mathbf{x}_i).$$

We will use covering numbers to estimate Rademacher complexity. Now we introduce some facts about covering numbers. The empirical $\epsilon$-covering number of a real-valued function class w.r.t. $L_2(S)$ metric is denoted by $\mathcal{N}(\mathcal{G}, \epsilon, L_2(S))$, i.e.,

$$\mathcal{N}(\mathcal{G}, \epsilon, L_2(S)) = \min\{|\mathcal{C}| : \forall g \in \mathcal{G}, \exists \bar{g} \in \mathcal{C} \, s.t. \|g - \bar{g}\|_{L_2(S)} \leq \epsilon\},$$

where $\|g - \bar{g}\|_{L_2(S)} = \sqrt{\frac{1}{|S|} \sum_{x \in S} |g(x) - \bar{g}(x)|^2}$. We denote $\mathcal{N}(\mathcal{F}, \epsilon, L_\infty(S))$ as the empirical $\epsilon$-covering number of a real-valued function class w.r.t. $L_\infty(S)$ metric, i.e.,

$$\mathcal{N}(\mathcal{F}, \epsilon, L_\infty(S))$$
$$= \min\{|\mathcal{C}| : \forall f \in \mathcal{F}, \exists \bar{f} \in \mathcal{C} \, s.t. \|f - \bar{f}\|_{L_\infty(S)} \leq \epsilon\},$$

where $\|f - \bar{f}\|_{L_\infty(S)} = \max_{s \in S} |f(x) - \bar{f}(x)|$. The empirical $\epsilon$-covering number of a vector-valued function class w.r.t. $L_{\infty,2}(S)$ metric is denoted by $\mathcal{N}(\mathcal{G}, \epsilon, L_{\infty,2})$, i.e.,

$$\mathcal{N}(\mathcal{G}, \epsilon, L_{\infty,2}(S)) = \min\{|\mathcal{C}| : \forall g \in \mathcal{G}, \exists \bar{g} \in \mathcal{C} \, s.t. \|g - \bar{g}\|_{L_{\infty,2}} \leq \epsilon\},$$

where $\|g - \bar{g}\|_{L_{\infty,2}(S)} = \max_{x \in S} \|g(x) - \bar{g}(x)\|_2$. The relationship between Rademacher complexities and covering numbers is established in the following proposition.

**Proposition A.3** (Dudley's entropy integral, Lemma A.5 in Bartlett et al. (2017)). *Let $\mathcal{Z}$ be a vector space and $\mathcal{F}$ be a class of functions $f : \mathcal{Z} \mapsto \mathbb{R}$. Let $S = \{\mathbf{z}_1, \cdots, \mathbf{z}_n\}$ be a dataset. Denote $B_{\mathcal{F}} = \sup_{f \in \mathcal{F}} \|f\|_{L_2(S)}$, we have*

$$\mathfrak{R}_S(\mathcal{F}) \leq \inf_{\alpha > 0} 4\alpha + \frac{12}{\sqrt{n}} \int_\alpha^{B_{\mathcal{F}}} \sqrt{\log \mathcal{N}(\mathcal{F}, \epsilon, L_2(S))} d\epsilon.$$

Now we provide some results on the covering numbers for the hypothesis $\mathcal{W}$ defined in Assumption 4.1.

**Lemma A.4.** *Let Assumption 4.1 hold. Then for any $\varepsilon > 0$, there exists a cover of $\mathcal{W}$: $\mathcal{C}_\varepsilon = \{\tilde{\mathbf{w}}_1, \cdots, \tilde{\mathbf{w}}_{|\mathcal{C}_\varepsilon|}\}$ with*

$$\log |\mathcal{C}_\varepsilon| \leq \sum_{l=1}^{L} d_l d_{l-1} \log \left( 1 + \frac{2L\Pi_{l=1}^{L} s_l}{\varepsilon} \right).$$

*For any $\mathbf{w} \in \mathcal{W}$, there exists $\tilde{\mathbf{w}} \in \mathcal{C}_\varepsilon$, such that*

$$\|f_{\mathbf{w}}(\mathbf{x}) - f_{\tilde{\mathbf{w}}}(\mathbf{x})\|_2 \leq \varepsilon, \|f_{\mathbf{w}}(\mathbf{y}) - f_{\tilde{\mathbf{w}}}(\mathbf{y})\|_2 \leq \varepsilon \tag{17}$$

*holds for any $\mathbf{x} \in \mathcal{X}, \mathbf{y} \in \mathcal{Y}$.*

*Proof.* We denote the output of $l$-th layer by $h_l(\mathbf{x})$ and $h_0(\mathbf{x}) = \mathbf{x}$. Then $h_l(\mathbf{x}) = \sigma(W_l h_{l-1}(\mathbf{x})), l \in [L]$. For any $\mathbf{x} \in \mathcal{X}, \mathbf{w} \in \mathcal{W}$, we have

$$\|h_l(\mathbf{x})\|_2 = \|\sigma(W_l h_{l-1}(\mathbf{x})) - 0\|_2 \leq \|W_l h_{l-1}(\mathbf{x})\|_2 \leq s_l \|h_{l-1}(\mathbf{x})\|_2 \leq \cdots \leq \Pi_{r=1}^{l} s_l \|\mathbf{x}\|_2 \leq \Pi_{r=1}^{l} s_l. \tag{18}$$

For $\mathbf{w}' = (W'_L, \cdots, W'_1)$, we denote the output of $l$-th layer by $h'_l(\mathbf{x})$. Then we have,

$$\begin{aligned}
\|f_{\mathbf{w}}(\mathbf{x}) - f_{\mathbf{w}'}(\mathbf{x})\|_2 &= \|h_L(\mathbf{x}) - h'_L(\mathbf{x})\|_2 = \|\sigma(W_L h_{L-1}(\mathbf{x})) - \sigma(W'_L h'_{L-1}(\mathbf{x}))\|_2 \\
&\leq \|W_L h_{L-1}(\mathbf{x}) - W'_L h'_{L-1}(\mathbf{x})\|_2 \\
&\leq \|W_L(h_{L-1}(\mathbf{x}) - h'_{L-1}(\mathbf{x}))\|_2 + \|(W_L - W'_L)h'_{L-1}(\mathbf{x})\|_2 \\
&\leq \|W_L\|_2 \|h_{L-1}(\mathbf{x}) - h'_{L-1}(\mathbf{x})\|_2 + \|W_L - W'_L\|_2 \|h'_{L-1}(\mathbf{x})\|_2 \\
&\leq s_L \|h_{L-1}(\mathbf{x}) - h'_{L-1}(\mathbf{x})\|_2 + \Pi_{l=1}^{L-1} s_l \|W_L - W'_L\|_2 \\
&\leq \cdots \leq \Pi_{l=1}^{L} s_l \sum_{l=1}^{L} \frac{\|W_L - W'_l\|_2}{s_l}.
\end{aligned}$$

Given $\varepsilon > 0$, we define

$$\varepsilon_l = \frac{s_l \varepsilon}{L\Pi_{l=1}^{L} s_l}.$$

Let $\mathcal{W}_l = \{W_l \in \mathbb{R}^{d_l \times d_{l-1}} : \|W_l\|_2 \leq s_l\}$, according to Example 5.8 in Wainwright (2019), there exists a $\varepsilon_l$-cover of $\mathcal{W}_l$ w.r.t. $\|\cdot\|_2$-which we denote by $\mathcal{C}_l(\mathcal{W}_l, \varepsilon_l)$, satisfying

$$|\mathcal{C}_l(\mathcal{W}_l, \varepsilon_l)| \leq \left( 1 + \frac{2s_l}{\varepsilon_l} \right)^{d_l d l-1} = \left( 1 + \frac{2L\Pi_{l=1}^{L} s_l}{\varepsilon} \right)^{d_l d_{l-1}}.$$

Now we can construct a cover of $\mathcal{W}$ through $\mathcal{C}_\varepsilon = \mathcal{C}_1(\mathcal{W}_1, \varepsilon_1) \times \cdots \times \mathcal{C}_L(\mathcal{W}_L, \varepsilon)$, where $\times$ the Cartesian product. For any $\mathbf{w} \in \mathcal{W}$, there exists $\tilde{\mathbf{w}} = (\widetilde{W}_1, \cdots, \widetilde{W}_L) \in \mathcal{C}$, such that for any $l \in [L]$,

$$\|W_l - \widetilde{W}_l\|_2 \leq \varepsilon_l.$$

Hence, for any $\mathbf{x} \in \mathcal{X}$, we have

$$\|f_{\mathbf{w}}(\mathbf{x}) - f_{\tilde{\mathbf{w}}}(\mathbf{x})\|_2 \leq \Pi_{l=1}^{L} s_l \sum_{l=1}^{L} \frac{\|W_L - \widetilde{W}_l\|_2}{s_l} \leq \Pi_{l=1}^{L} s_l \sum_{l=1}^{L} \frac{\varepsilon_l}{s_l} = \varepsilon.$$

Similarly, for any $\mathbf{y} \in \mathcal{Y}$, $\|f_{\mathbf{w}}(\mathbf{y}) - f_{\tilde{\mathbf{w}}}(\mathbf{y})\|_2 \leq \varepsilon$. The covering number satisfies

$$\log |\mathcal{C}_\varepsilon| \leq \sum_{l=1}^{L} \log |\mathcal{C}_l(\mathcal{W}_l, \varepsilon_l)| \leq \sum_{l=1}^{L} \log \left( 1 + \frac{2L\Pi_{l=1}^{L} s_l}{\varepsilon} \right)^{d_l d_{l-1}} = \sum_{l=1}^{L} d_l d_{l-1} \log \left( 1 + \frac{2L\Pi_{l=1}^{L} s_l}{\varepsilon} \right).$$

The proof is completed. $\square$

We define

$$S_1 := \{(\mathbf{x}_i, \mathbf{y}_i, \mathbf{y}_{ij}^-) : 1 \le i \le n, 1 \le j \le m\}, \tag{19}$$

$$\mathcal{H} := \{(\mathbf{x}, \mathbf{y}, \mathbf{y}') \mapsto \Delta_{\mathbf{w}}(\mathbf{x}, \mathbf{y}, \mathbf{y}') : \mathbf{w} \in \mathcal{W}\}. \tag{20}$$

An upper bound of $\mathcal{N}(\mathcal{H}, \epsilon, L_\infty(S_1))$ in the following lemma.

**Lemma A.5.** *Let Assumption 4.1 hold. Then for any $\varepsilon > 0$,*

$$\log \mathcal{N}(\mathcal{H}, \epsilon, L_\infty(S_1)) \le \sum_{l=1}^{L} d_l d_{l-1} \log \left(1 + \frac{8L\Pi_{l=1}^{L} s_l^2}{\epsilon}\right).$$

*Proof.* For $\mathbf{w}, \mathbf{w}' \in \mathcal{W}$ and a triplet $(\mathbf{x}, \mathbf{y}, \mathbf{y}')$, we have

$$
\begin{aligned}
|\Delta_{\mathbf{w}}(\mathbf{x}, \mathbf{y}, \mathbf{y}') - \Delta_{\mathbf{w}'}(\mathbf{x}, \mathbf{y}, \mathbf{y}')| = & |(s_{\mathbf{w}}(\mathbf{x}, \mathbf{y}') - s_{\mathbf{w}}(\mathbf{x}, \mathbf{y})) - (s_{\mathbf{w}'}(\mathbf{x}, \mathbf{y}') - s_{\mathbf{w}'}(\mathbf{x}, \mathbf{y}))| \\
= & |f_{\mathbf{w}}(\mathbf{x})^\top (f_{\mathbf{w}}(\mathbf{y}') - f_{\mathbf{w}}(\mathbf{y})) - f_{\mathbf{w}'}(\mathbf{x})^\top (f_{\mathbf{w}'}(\mathbf{y}') - f_{\mathbf{w}'}(\mathbf{y}))| \\
\le & |(f_{\mathbf{w}}(\mathbf{x}) - f_{\mathbf{w}'}(\mathbf{x}))^\top (f_{\mathbf{w}}(\mathbf{y}') - f_{\mathbf{w}}(\mathbf{y}))| \\
& + |f_{\mathbf{w}'}(\mathbf{x})^\top (f_{\mathbf{w}}(\mathbf{y}') - f_{\mathbf{w}}(\mathbf{y}) - f_{\mathbf{w}'}(\mathbf{y}') + f_{\mathbf{w}'}(\mathbf{y}))| \\
\le & (\|f_{\mathbf{w}}(\mathbf{x})\|_2 + \|f_{\mathbf{w}'}(\mathbf{x})\|_2) \|f_{\mathbf{w}}(\mathbf{y}) - f_{\mathbf{w}'}(\mathbf{y})\|_2 \\
& + \|f_{\mathbf{w}'}(\mathbf{x})\|_2 (\|f_{\mathbf{w}'}(\mathbf{y}) - f_{\mathbf{w}}(\mathbf{y})\|_2 + \|f_{\mathbf{w}'}(\mathbf{y}') - f_{\mathbf{w}}(\mathbf{y}')\|_2) \\
\le & 4\Pi_{l=1}^{L} s_l \max_{\mathbf{v} \in \{\mathbf{x}, \mathbf{y}, \mathbf{y}'\}} \|f_{\mathbf{w}}(\mathbf{v}) - f_{\mathbf{w}'}(\mathbf{v})\|_2. 
\end{aligned}
\tag{21}
$$

we define

$$S_2 = \{\mathbf{x}_i, \mathbf{y}'_{ij}, \mathbf{y}_i : 1 \le n, 1 \le j \le m.\}$$

to be the set of all $n(m+2)$ training data. Therefore,

$$\mathcal{N}(\mathcal{H}, \epsilon, L_\infty(S_1)) \le \mathcal{N}(\mathcal{F}, \epsilon/(4\Pi_{l=1}^{L} s_l), L_{\infty,2}(S_2)).$$

According to Lemma A.4, suppose $\mathcal{C}_\varepsilon = \{\tilde{\mathbf{w}}_1, \cdots, \tilde{\mathbf{w}}_N\}$ is a cover of $\mathcal{W}$ such that Eq. (17) holds. Then

$$\|f_{\mathbf{w}} - f_{\tilde{\mathbf{w}}}\|_{L_{\infty,2}(S_2)} = \max_{\mathbf{v} \in S_2} \|f_{\mathbf{w}}(\mathbf{v}) - f_{\tilde{\mathbf{w}}}(\mathbf{v})\|_2 \le \sup_{\mathbf{v} \in \mathcal{X}, \mathcal{Y}} \|f_{\mathbf{w}}(\mathbf{v}) - f_{\tilde{\mathbf{w}}}(\mathbf{v})\|_2 \le \varepsilon.$$

Therefore, $\mathcal{N}(\mathcal{F}, \varepsilon, L_{\infty,2}(S_2)) \le |\mathcal{C}_\varepsilon|$. Letting $\varepsilon = \epsilon/(4\Pi_{l=1}^{L} s_l)$, we have

$$\log \mathcal{N}(\mathcal{H}, \epsilon, L_\infty(S_1)) \le \log |\mathcal{C}_{\epsilon/(4\Pi_{l=1}^{L} s_l)}| \le \sum_{l=1}^{L} d_l d_{l-1} \log \left(1 + \frac{8L\Pi_{l=1}^{L} s_l^2}{\epsilon}\right).$$

We have completed the proof of the lemma. $\qquad\square$

## B. Proofs in Section 3

### B.1. Proof of Lemma 3.2

*Proof of Lemma 3.2.* For a scoring function $s$ and an anchor point $\mathbf{x}$, we define the following distribution

$$q_{\mathbf{x}}^+(\mathbf{y}) = \frac{p_{\mathbf{x}}^-(\mathbf{y}) \exp(s(\mathbf{x}, \mathbf{y})/\tau)}{\mathbb{E}_{\mathbf{y}' \sim p_{\mathbf{x}}^-} \exp(s(\mathbf{x}, \mathbf{y}')/\tau)}. \tag{22}$$

Then we have

$$
\begin{aligned}
\tau \mathbb{E}_{\mathbf{x}} D_{\mathrm{KL}}(p_{\mathbf{x}}^+ \| q_{\mathbf{x}}^+) = & \tau \mathbb{E}_{\mathbf{x}} \mathbb{E}_{\mathbf{y} \sim p_{\mathbf{x}}^+} \log \frac{p_{\mathbf{x}}^+(\mathbf{y})}{q_{\mathbf{x}}^+(\mathbf{y})} = \tau \mathbb{E}_{\mathbf{x}} \mathbb{E}_{\mathbf{y} \sim p_{\mathbf{x}}^+} \log \frac{p_{\mathbf{x}}^+(\mathbf{y}) \mathbb{E}_{\mathbf{y}' \sim p_{\mathbf{x}}^-} \exp(s(\mathbf{x}, \mathbf{y}')/\tau)}{p_{\mathbf{x}}^-(\mathbf{y}) \exp(s(\mathbf{x}, \mathbf{y})/\tau)} \\
= & \tau \mathbb{E}_{\mathbf{x}} \mathbb{E}_{\mathbf{y} \sim p_{\mathbf{x}}^+} \log \frac{p_{\mathbf{x}}^+(\mathbf{y})}{p_{\mathbf{x}}^-(\mathbf{y})} + \tau \mathbb{E}_{\mathbf{x}} \mathbb{E}_{\mathbf{y} \sim p_{\mathbf{x}}^+} \log \mathbb{E}_{\mathbf{y}' \sim p_{\mathbf{x}}^-} \exp((s(\mathbf{x}, \mathbf{y}') - s(\mathbf{x}, \mathbf{y}))/\tau) \\
= & \mathcal{L}(s) - \tau \mathbb{E}_{\mathbf{x}} \mathbb{E}_{\mathbf{y} \sim p_{\mathbf{x}}^+} \log \frac{p_{\mathbf{x}}^-(\mathbf{y})}{p_{\mathbf{x}}^+(\mathbf{y})}. 
\end{aligned}
\tag{23}
$$

By the nonnegativity of KL-divergence, there holds

$$\mathcal{L}(s) \geq \tau \mathbb{E}_{\mathbf{x}} \mathbb{E}_{\mathbf{y} \sim p_{\mathbf{x}}^+} \log \frac{p_{\mathbf{x}}^-(\mathbf{y})}{p_{\mathbf{x}}^+(\mathbf{y})} := \mathcal{L}^*.$$

By the equality condition of KL-divergence, $\mathcal{L}(s) - \mathcal{L}^* = 0$ is achieved if and only if

$$p_{\mathbf{x}}^+(\mathbf{y}) = q_{\mathbf{x}}^+(\mathbf{y}) = \frac{p_{\mathbf{x}}^-(\mathbf{y}) \exp(s(\mathbf{x}, \mathbf{y})/\tau)}{\mathbb{E}_{\mathbf{y}' \sim p_{\mathbf{x}}^-} \exp(s(\mathbf{x}, \mathbf{y}')/\tau)}$$

for any $\mathbf{x} \in \mathcal{X}, \mathbf{y} \in \mathcal{Y}$. Therefore, the set of the minimizers of $\mathcal{L}$ is given by

$$\mathcal{A} = \left\{ s : \frac{p_{\mathbf{x}}^-(\mathbf{y}) \exp(s(\mathbf{x}, \mathbf{y})/\tau)}{p_{\mathbf{x}}^+(\mathbf{y})} = \mathbb{E}_{\mathbf{y}' \sim p_{\mathbf{x}}^-} \exp(s(\mathbf{x}, \mathbf{y}')/\tau) \right\}.$$

We also denote

$$\mathcal{A}' = \left\{ s : s(\mathbf{x}, \mathbf{y}) = \tau \log \frac{p_{\mathbf{x}}^+(\mathbf{y})}{p_{\mathbf{x}}^-(\mathbf{y})} + g(\mathbf{x}), \text{for some measurable function } g : \mathcal{X} \to \mathbb{R} \right\}.$$

We aim to show that $\mathcal{A} = \mathcal{A}'$.

For any $s \in \mathcal{A}$, let $g_s(\mathbf{x}) = \tau \log \mathbb{E}_{\mathbf{y}' \sim p_{\mathbf{x}}^-} \exp(s(\mathbf{x}, \mathbf{y}')/\tau)$ be a measurable function. Then there holds

$$s(\mathbf{x}, \mathbf{y}) = \tau \log \left( \frac{p_{\mathbf{x}}^+(\mathbf{y})}{p_{\mathbf{x}}^-(\mathbf{y})} \mathbb{E}_{\mathbf{y}' \sim p_{\mathbf{x}}^-} \exp(s(\mathbf{x}, \mathbf{y}')/\tau) \right) = \tau \log \frac{p_{\mathbf{x}}^+(\mathbf{y})}{p_{\mathbf{x}}^-(\mathbf{y})} + g_s(\mathbf{x}).$$

Therefore, $s \in \mathcal{A}'$, this implies that $\mathcal{A}' \subseteq \mathcal{A}'$.

To show that $\mathcal{A}' \subseteq \mathcal{A}$, we take a function $s \in \mathcal{A}'$. Then there exists a function $g$ such that

$$s(\mathbf{x}, \mathbf{y}) = \tau \log \frac{p_{\mathbf{x}}^+(\mathbf{y})}{p_{\mathbf{x}}^-(\mathbf{y})} + g(\mathbf{x}).$$

Hence, $\exp(s(\mathbf{x}, \mathbf{y})/\tau) = \frac{p_{\mathbf{x}}^+(\mathbf{y})}{p_{\mathbf{x}}^-(\mathbf{y})} \exp(g(\mathbf{x})/\tau)$ and

$$\frac{p_{\mathbf{x}}^-(\mathbf{y}) \exp(s(\mathbf{x}, \mathbf{y})/\tau)}{p_{\mathbf{x}}^+(\mathbf{y})} = \exp(g(\mathbf{x})/\tau).$$

It further holds

$$\mathbb{E}_{\mathbf{y}' \sim p_{\mathbf{x}}^-} \exp(s(\mathbf{x}, \mathbf{y}')/\tau) = \mathbb{E}_{\mathbf{y}' \sim p_{\mathbf{x}}^-} \left[ \frac{p_{\mathbf{x}}^+(\mathbf{y}')}{p_{\mathbf{x}}^-(\mathbf{y}')} \exp(g(\mathbf{x})/\tau) \right] = \int_{\mathcal{Y}} p_{\mathbf{x}}^-(\mathbf{y}') \frac{p_{\mathbf{x}}^+(\mathbf{y}')}{p_{\mathbf{x}}^-(\mathbf{y}')} \exp(g(\mathbf{x})/\tau) d\mathbf{y}'$$

$$= \int_{\mathcal{Y}} p_{\mathbf{x}}^+(\mathbf{y}') \exp(g(\mathbf{x})/\tau) d\mathbf{y}' = \exp(g(\mathbf{x})/\tau) \int_{\mathcal{Y}} p_{\mathbf{x}}^+(\mathbf{y}') d\mathbf{y}' = \exp(g(\mathbf{x})/\tau),$$

where the last inequality is because $p_{\mathbf{x}}^+$ is a density function and thus $\int_{\mathcal{Y}} p_{\mathbf{x}}^+(\mathbf{y}') d\mathbf{y}' = 1$.

As a result,

$$\frac{p_{\mathbf{x}}^-(\mathbf{y}) \exp(s(\mathbf{x}, \mathbf{y})/\tau)}{p_{\mathbf{x}}^+(\mathbf{y})} = \mathbb{E}_{\mathbf{y}' \sim p_{\mathbf{x}}^-} \exp(s(\mathbf{x}, \mathbf{y}')/\tau).$$

This indicates that $s \in \mathcal{A}$, and $\mathcal{A}' \subseteq \mathcal{A}$.

Consequently, $\mathcal{A} = \mathcal{A}'$. A scoring function $s$ is a minimizer of $\mathcal{L}$ *if and only if*

$$s(\mathbf{x}, \mathbf{y}) = \tau \log \frac{p_{\mathbf{x}}^+(\mathbf{y})}{p_{\mathbf{x}}^-(\mathbf{y})} + g(\mathbf{x}), \text{for some measurable function } g : \mathcal{X} \to \mathbb{R}.$$

The proof is completed.

$\square$

## B.2. Proof of Lemma 3.3

We first present the following technical lemma.

**Lemma B.1.** *For any* $\mathbf{x} \in \mathcal{X}, \mathbf{y}, \mathbf{y}' \in \mathcal{Y}$ *and socring function* $s$, *there holds*

$$
\mathbb{I}\left[\left(\frac{p_{\mathbf{x}}^+(\mathbf{y})}{p_{\mathbf{x}}^-(\mathbf{y})} - \frac{p_{\mathbf{x}}^+(\mathbf{y}')}{p_{\mathbf{x}}^-(\mathbf{y}')}\right)(s(\mathbf{x},\mathbf{y}) - s(\mathbf{x},\mathbf{y}')) \le 0\right]\left|\frac{p_{\mathbf{x}}^+(\mathbf{y})}{p_{\mathbf{x}}^-(\mathbf{y})} - \frac{p_{\mathbf{x}}^+(\mathbf{y}')}{p_{\mathbf{x}}^-(\mathbf{y}')}\right|
$$

$$
\le \left|\frac{p_{\mathbf{x}}^+(\mathbf{y})}{p_{\mathbf{x}}^-(\mathbf{y})} - \frac{p_{\mathbf{x}}^+(\mathbf{y}')}{p_{\mathbf{x}}^-(\mathbf{y}')}\right| - \left(\frac{p_{\mathbf{x}}^+(\mathbf{y})}{p_{\mathbf{x}}^-(\mathbf{y})} - \frac{p_{\mathbf{x}}^+(\mathbf{y}')}{p_{\mathbf{x}}^-(\mathbf{y}')}\right)(\mathbb{I}[s(\mathbf{x},\mathbf{y}) > s(\mathbf{x},\mathbf{y}')] - \mathbb{I}[s(\mathbf{x},\mathbf{y}') > s(\mathbf{x},\mathbf{y})]) \tag{24}
$$

$$
\le 2\mathbb{I}\left[\left(\frac{p_{\mathbf{x}}^+(\mathbf{y})}{p_{\mathbf{x}}^-(\mathbf{y})} - \frac{p_{\mathbf{x}}^+(\mathbf{y}')}{p_{\mathbf{x}}^-(\mathbf{y}')}\right)(s(\mathbf{x},\mathbf{y}) - s(\mathbf{x},\mathbf{y}')) \le 0\right]\left|\frac{p_{\mathbf{x}}^+(\mathbf{y})}{p_{\mathbf{x}}^-(\mathbf{y})} - \frac{p_{\mathbf{x}}^+(\mathbf{y}')}{p_{\mathbf{x}}^-(\mathbf{y}')}\right|.
$$

*Proof.* The above inequality holds true when $\frac{p_{\mathbf{x}}^+(\mathbf{y})}{p_{\mathbf{x}}^-(\mathbf{y})} - \frac{p_{\mathbf{x}}^+(\mathbf{y}')}{p_{\mathbf{x}}^-(\mathbf{y}')} = 0$. We ony need to consider the case $\frac{p_{\mathbf{x}}^+(\mathbf{y})}{p_{\mathbf{x}}^-(\mathbf{y})} - \frac{p_{\mathbf{x}}^+(\mathbf{y}')}{p_{\mathbf{x}}^-(\mathbf{y}')} \ne 0$. By the symmetry of $\mathbf{y}, \mathbf{y}'$, we assume that

$$
\frac{p_{\mathbf{x}}^+(\mathbf{y})}{p_{\mathbf{x}}^-(\mathbf{y})} - \frac{p_{\mathbf{x}}^+(\mathbf{y}')}{p_{\mathbf{x}}^-(\mathbf{y}')} > 0.
$$

Similar arguments also follows for $\frac{p_{\mathbf{x}}^+(\mathbf{y})}{p_{\mathbf{x}}^-(\mathbf{y})} - \frac{p_{\mathbf{x}}^+(\mathbf{y}')}{p_{\mathbf{x}}^-(\mathbf{y}')} < 0$. We verify Eq. (24) by considering three cases:

Case 1: $s(\mathbf{x}, \mathbf{y}) > s(\mathbf{x}, \mathbf{y}')$. We have

$$
\frac{p_{\mathbf{x}}^+(\mathbf{y})}{p_{\mathbf{x}}^-(\mathbf{y})} - \frac{p_{\mathbf{x}}^+(\mathbf{y}')}{p_{\mathbf{x}}^-(\mathbf{y}')} - \left(\frac{p_{\mathbf{x}}^+(\mathbf{y})}{p_{\mathbf{x}}^-(\mathbf{y})} - \frac{p_{\mathbf{x}}^+(\mathbf{y}')}{p_{\mathbf{x}}^-(\mathbf{y}')}\right)(\mathbb{I}[s(\mathbf{x},\mathbf{y}) > s(\mathbf{x},\mathbf{y}')] - \mathbb{I}[s(\mathbf{x},\mathbf{y}') > s(\mathbf{x},\mathbf{y})])
$$

$$
= \frac{p_{\mathbf{x}}^+(\mathbf{y})}{p_{\mathbf{x}}^-(\mathbf{y})} - \frac{p_{\mathbf{x}}^+(\mathbf{y}')}{p_{\mathbf{x}}^-(\mathbf{y}')} - \left(\frac{p_{\mathbf{x}}^+(\mathbf{y})}{p_{\mathbf{x}}^-(\mathbf{y})} - \frac{p_{\mathbf{x}}^+(\mathbf{y}')}{p_{\mathbf{x}}^-(\mathbf{y}')}\right)(1 - 0) = 0,
$$

$$
\mathbb{I}\left[\left(\frac{p_{\mathbf{x}}^+(\mathbf{y})}{p_{\mathbf{x}}^-(\mathbf{y})} - \frac{p_{\mathbf{x}}^+(\mathbf{y}')}{p_{\mathbf{x}}^-(\mathbf{y}')}\right)(s(\mathbf{x},\mathbf{y}) - s(\mathbf{x},\mathbf{y}')) \le 0\right]\left(\frac{p_{\mathbf{x}}^+(\mathbf{y})}{p_{\mathbf{x}}^-(\mathbf{y})} - \frac{p_{\mathbf{x}}^+(\mathbf{y}')}{p_{\mathbf{x}}^-(\mathbf{y}')}\right) = 0.
$$

Therefore, Eq. (24) becomes $0 \le 0 \le 0$, which holds true.

Case 2: $s(\mathbf{x}, \mathbf{y}) = s(\mathbf{x}, \mathbf{y}')$. We have

$$
\frac{p_{\mathbf{x}}^+(\mathbf{y})}{p_{\mathbf{x}}^-(\mathbf{y})} - \frac{p_{\mathbf{x}}^+(\mathbf{y}')}{p_{\mathbf{x}}^-(\mathbf{y}')} - \left(\frac{p_{\mathbf{x}}^+(\mathbf{y})}{p_{\mathbf{x}}^-(\mathbf{y})} - \frac{p_{\mathbf{x}}^+(\mathbf{y}')}{p_{\mathbf{x}}^-(\mathbf{y}')}\right)(\mathbb{I}[s(\mathbf{x},\mathbf{y}) > s(\mathbf{x},\mathbf{y}')] - \mathbb{I}[s(\mathbf{x},\mathbf{y}') > s(\mathbf{x},\mathbf{y})])
$$

$$
= \frac{p_{\mathbf{x}}^+(\mathbf{y})}{p_{\mathbf{x}}^-(\mathbf{y})} - \frac{p_{\mathbf{x}}^+(\mathbf{y}')}{p_{\mathbf{x}}^-(\mathbf{y}')} - \left(\frac{p_{\mathbf{x}}^+(\mathbf{y})}{p_{\mathbf{x}}^-(\mathbf{y})} - \frac{p_{\mathbf{x}}^+(\mathbf{y}')}{p_{\mathbf{x}}^-(\mathbf{y}')}\right)(0 - 0) = \frac{p_{\mathbf{x}}^+(\mathbf{y})}{p_{\mathbf{x}}^-(\mathbf{y})} - \frac{p_{\mathbf{x}}^+(\mathbf{y}')}{p_{\mathbf{x}}^-(\mathbf{y}')},
$$

$$
\mathbb{I}\left[\left(\frac{p_{\mathbf{x}}^+(\mathbf{y})}{p_{\mathbf{x}}^-(\mathbf{y})} - \frac{p_{\mathbf{x}}^+(\mathbf{y}')}{p_{\mathbf{x}}^-(\mathbf{y}')}\right)(s(\mathbf{x},\mathbf{y}) - s(\mathbf{x},\mathbf{y}')) \le 0\right]\left(\frac{p_{\mathbf{x}}^+(\mathbf{y})}{p_{\mathbf{x}}^-(\mathbf{y})} - \frac{p_{\mathbf{x}}^+(\mathbf{y}')}{p_{\mathbf{x}}^-(\mathbf{y}')}\right)
$$

$$
= \frac{p_{\mathbf{x}}^+(\mathbf{y})}{p_{\mathbf{x}}^-(\mathbf{y})} - \frac{p_{\mathbf{x}}^+(\mathbf{y}')}{p_{\mathbf{x}}^-(\mathbf{y}')}.
$$

In this case, Eq. (24) becomes

$$
\frac{p_{\mathbf{x}}^+(\mathbf{y})}{p_{\mathbf{x}}^-(\mathbf{y})} - \frac{p_{\mathbf{x}}^+(\mathbf{y}')}{p_{\mathbf{x}}^-(\mathbf{y}')} \le \frac{p_{\mathbf{x}}^+(\mathbf{y})}{p_{\mathbf{x}}^-(\mathbf{y})} - \frac{p_{\mathbf{x}}^+(\mathbf{y}')}{p_{\mathbf{x}}^-(\mathbf{y}')} \le 2\left(\frac{p_{\mathbf{x}}^+(\mathbf{y})}{p_{\mathbf{x}}^-(\mathbf{y})} - \frac{p_{\mathbf{x}}^+(\mathbf{y}')}{p_{\mathbf{x}}^-(\mathbf{y}')}\right),
$$

which holds true by noting that $\frac{p_{\mathbf{x}}^+(\mathbf{y})}{p_{\mathbf{x}}^-(\mathbf{y})} - \frac{p_{\mathbf{x}}^+(\mathbf{y}')}{p_{\mathbf{x}}^-(\mathbf{y}')} > 0$.

Case 3: $s(\mathbf{x}, \mathbf{y}) < s(\mathbf{x}, \mathbf{y}')$. We have

$$\frac{p_{\mathbf{x}}^+(\mathbf{y})}{p_{\mathbf{x}}^-(\mathbf{y})} - \frac{p_{\mathbf{x}}^+(\mathbf{y}')}{p_{\mathbf{x}}^-(\mathbf{y}')} - \left(\frac{p_{\mathbf{x}}^+(\mathbf{y})}{p_{\mathbf{x}}^-(\mathbf{y})} - \frac{p_{\mathbf{x}}^+(\mathbf{y}')}{p_{\mathbf{x}}^-(\mathbf{y}')}\right)(\mathbb{I}[s(\mathbf{x}, \mathbf{y}) > s(\mathbf{x}, \mathbf{y}')] - \mathbb{I}[s(\mathbf{x}, \mathbf{y}') > s(\mathbf{x}, \mathbf{y})])$$

$$=\frac{p_{\mathbf{x}}^+(\mathbf{y})}{p_{\mathbf{x}}^-(\mathbf{y})} - \frac{p_{\mathbf{x}}^+(\mathbf{y}')}{p_{\mathbf{x}}^-(\mathbf{y}')} - \left(\frac{p_{\mathbf{x}}^+(\mathbf{y})}{p_{\mathbf{x}}^-(\mathbf{y})} - \frac{p_{\mathbf{x}}^+(\mathbf{y}')}{p_{\mathbf{x}}^-(\mathbf{y}')}\right)(0-1) = 2\left(\frac{p_{\mathbf{x}}^+(\mathbf{y})}{p_{\mathbf{x}}^-(\mathbf{y})} - \frac{p_{\mathbf{x}}^+(\mathbf{y}')}{p_{\mathbf{x}}^-(\mathbf{y}')}\right),$$

$$\mathbb{I}\left[\left(\frac{p_{\mathbf{x}}^+(\mathbf{y})}{p_{\mathbf{x}}^-(\mathbf{y})} - \frac{p_{\mathbf{x}}^+(\mathbf{y}')}{p_{\mathbf{x}}^-(\mathbf{y}')}\right)(s(\mathbf{x}, \mathbf{y}) - s(\mathbf{x}, \mathbf{y}')) \le 0\right]\left(\frac{p_{\mathbf{x}}^+(\mathbf{y})}{p_{\mathbf{x}}^-(\mathbf{y})} - \frac{p_{\mathbf{x}}^+(\mathbf{y}')}{p_{\mathbf{x}}^-(\mathbf{y}')}\right)$$

$$=\frac{p_{\mathbf{x}}^+(\mathbf{y})}{p_{\mathbf{x}}^-(\mathbf{y})} - \frac{p_{\mathbf{x}}^+(\mathbf{y}')}{p_{\mathbf{x}}^-(\mathbf{y}')}.$$

Consequently, Eq. (24) becomes

$$\frac{p_{\mathbf{x}}^+(\mathbf{y})}{p_{\mathbf{x}}^-(\mathbf{y})} - \frac{p_{\mathbf{x}}^+(\mathbf{y}')}{p_{\mathbf{x}}^-(\mathbf{y}')} \le 2\left(\frac{p_{\mathbf{x}}^+(\mathbf{y})}{p_{\mathbf{x}}^-(\mathbf{y})} - \frac{p_{\mathbf{x}}^+(\mathbf{y}')}{p_{\mathbf{x}}^-(\mathbf{y}')}\right) \le 2\left(\frac{p_{\mathbf{x}}^+(\mathbf{y})}{p_{\mathbf{x}}^-(\mathbf{y})} - \frac{p_{\mathbf{x}}^+(\mathbf{y}')}{p_{\mathbf{x}}^-(\mathbf{y}')}\right),$$

which holds true.

Combining three cases together, we have completed the proof of the lemma. $\square$

Now we present the proof of Lemma 3.3.

*Proof of Lemma 3.3.* Recall the definition of $\mathcal{E}(s)$ in Eq. (2), there holds

$$\mathcal{E}(s) = \mathbb{E}_{\mathbf{x}}\mathbb{E}_{\mathbf{y}\sim p_{\mathbf{x}}^+, \mathbf{y}'\sim p_{\mathbf{x}}^-}[\mathbb{I}[s(\mathbf{x}, \mathbf{y}) > s(\mathbf{x}, \mathbf{y}')]] + \frac{1}{2}\mathbb{E}_{\mathbf{x}}\mathbb{E}_{\mathbf{y}\sim p_{\mathbf{x}}^+, \mathbf{y}'\sim p_{\mathbf{x}}^-}[\mathbb{I}[s(\mathbf{x}, \mathbf{y}) = s(\mathbf{x}, \mathbf{y}')]]$$

$$=\mathbb{E}_{\mathbf{x}}\left[\iint p_{\mathbf{x}}^+(\mathbf{y})p_{\mathbf{x}}^-(\mathbf{y}')(\mathbb{I}[s(\mathbf{x}, \mathbf{y}) > s(\mathbf{x}, \mathbf{y}')] + \frac{1}{2}\mathbb{I}[s(\mathbf{x}, \mathbf{y}) = s(\mathbf{x}, \mathbf{y}')])d\mathbf{y}d\mathbf{y}'\right]$$

$$=\mathbb{E}_{\mathbf{x}}\left[\iint p_{\mathbf{x}}^-(\mathbf{y})p_{\mathbf{x}}^-(\mathbf{y}')\frac{p_{\mathbf{x}}^+(\mathbf{y})}{p_{\mathbf{x}}^-(\mathbf{y})}(\mathbb{I}[s(\mathbf{x}, \mathbf{y}) > s(\mathbf{x}, \mathbf{y}')] + \frac{1}{2}\mathbb{I}[s(\mathbf{x}, \mathbf{y}) = s(\mathbf{x}, \mathbf{y}')])d\mathbf{y}d\mathbf{y}'\right]$$

$$=\mathbb{E}_{\mathbf{x}}\mathbb{E}_{\mathbf{y}\sim p_{\mathbf{x}}^-, \mathbf{y}'\sim p_{\mathbf{x}}^-}\left[\frac{p_{\mathbf{x}}^+(\mathbf{y})}{p_{\mathbf{x}}^-(\mathbf{y})}(\mathbb{I}[s(\mathbf{x}, \mathbf{y}) > s(\mathbf{x}, \mathbf{y}')] + \frac{1}{2}\mathbb{I}[s(\mathbf{x}, \mathbf{y}) = s(\mathbf{x}, \mathbf{y}')])\right]$$

$$=\frac{1}{2}\mathbb{E}_{\mathbf{x}}\mathbb{E}_{\mathbf{y}\sim p_{\mathbf{x}}^-, \mathbf{y}'\sim p_{\mathbf{x}}^-}\left[\frac{p_{\mathbf{x}}^+(\mathbf{y})}{p_{\mathbf{x}}^-(\mathbf{y})}\mathbb{I}[s(\mathbf{x}, \mathbf{y}) > s(\mathbf{x}, \mathbf{y}')] + \frac{p_{\mathbf{x}}^+(\mathbf{y}')}{p_{\mathbf{x}}^-(\mathbf{y}')}\mathbb{I}[s(\mathbf{x}, \mathbf{y}') > s(\mathbf{x}, \mathbf{y})]\right.$$

$$\left. + \frac{1}{4}\left(\frac{p_{\mathbf{x}}^+(\mathbf{y})}{p_{\mathbf{x}}^-(\mathbf{y})} + \frac{p_{\mathbf{x}}^+(\mathbf{y}')}{p_{\mathbf{x}}^-(\mathbf{y}')}\right)\mathbb{I}[s(\mathbf{x}, \mathbf{y}) = s(\mathbf{x}, \mathbf{y}')]\right],$$

where the last term is by the symmetry of $\mathbf{y}, \mathbf{y}'$. By noting that $\mathbb{I}[s(\mathbf{x}, \mathbf{y}') > s(\mathbf{x}, \mathbf{y})] + \mathbb{I}[s(\mathbf{x}, \mathbf{y}) > s(\mathbf{x}, \mathbf{y}')] + \mathbb{I}[s(\mathbf{x}, \mathbf{y}') = s(\mathbf{x}, \mathbf{y})] = 1$, we have

$$\mathcal{E}(s) = \frac{1}{2}\mathbb{E}_{\mathbf{x}}\mathbb{E}_{\mathbf{y}\sim p_{\mathbf{x}}^-, \mathbf{y}'\sim p_{\mathbf{x}}^-}\left[\frac{p_{\mathbf{x}}^+(\mathbf{y})}{p_{\mathbf{x}}^-(\mathbf{y})}\mathbb{I}[s(\mathbf{x}, \mathbf{y}) > s(\mathbf{x}, \mathbf{y}')] + \frac{p_{\mathbf{x}}^+(\mathbf{y}')}{p_{\mathbf{x}}^-(\mathbf{y}')}\mathbb{I}[s(\mathbf{x}, \mathbf{y}') > s(\mathbf{x}, \mathbf{y})]\right.$$

$$\left. + \frac{1}{4}\left(\frac{p_{\mathbf{x}}^+(\mathbf{y})}{p_{\mathbf{x}}^-(\mathbf{y})} + \frac{p_{\mathbf{x}}^+(\mathbf{y}')}{p_{\mathbf{x}}^-(\mathbf{y}')}\right)(1 - \mathbb{I}[s(\mathbf{x}, \mathbf{y}') > s(\mathbf{x}, \mathbf{y})] - \mathbb{I}[s(\mathbf{x}, \mathbf{y}) > s(\mathbf{x}, \mathbf{y}')])\right]$$

$$=\frac{1}{4}\mathbb{E}_{\mathbf{x}}\mathbb{E}_{\mathbf{y}\sim p_{\mathbf{x}}^-, \mathbf{y}'\sim p_{\mathbf{x}}^-}\left[\frac{p_{\mathbf{x}}^+(\mathbf{y})}{p_{\mathbf{x}}^-(\mathbf{y})} + \frac{p_{\mathbf{x}}^+(\mathbf{y}')}{p_{\mathbf{x}}^-(\mathbf{y}')} + \left(\frac{p_{\mathbf{x}}^+(\mathbf{y})}{p_{\mathbf{x}}^-(\mathbf{y})} - \frac{p_{\mathbf{x}}^+(\mathbf{y}')}{p_{\mathbf{x}}^-(\mathbf{y}')}\right)(\mathbb{I}[s(\mathbf{x}, \mathbf{y}) > s(\mathbf{x}, \mathbf{y}')] - \mathbb{I}[s(\mathbf{x}, \mathbf{y}') > s(\mathbf{x}, \mathbf{y})])\right]$$

$$\le \frac{1}{4}\mathbb{E}_{\mathbf{x}}\mathbb{E}_{\mathbf{y}\sim p_{\mathbf{x}}^-, \mathbf{y}'\sim p_{\mathbf{x}}^-}\left[\frac{p_{\mathbf{x}}^+(\mathbf{y})}{p_{\mathbf{x}}^-(\mathbf{y})} + \frac{p_{\mathbf{x}}^+(\mathbf{y}')}{p_{\mathbf{x}}^-(\mathbf{y}')} + \left|\frac{p_{\mathbf{x}}^+(\mathbf{y})}{p_{\mathbf{x}}^-(\mathbf{y})} - \frac{p_{\mathbf{x}}^+(\mathbf{y}')}{p_{\mathbf{x}}^-(\mathbf{y}')}\right|\right] := \mathcal{E}^* \tag{25}$$

Therefore,

$$\mathcal{E}^* - \mathcal{E}(s)$$
$$= \frac{1}{4} \mathbb{E}_{\mathbf{x}} \mathbb{E}_{\mathbf{y} \sim p_{\mathbf{x}}^-, \mathbf{y}' \sim p_{\mathbf{x}}^-} \left[ \left| \frac{p_{\mathbf{x}}^+(\mathbf{y})}{p_{\mathbf{x}}^-(\mathbf{y})} - \frac{p_{\mathbf{x}}^+(\mathbf{y}')}{p_{\mathbf{x}}^-(\mathbf{y}')} \right| - \left( \frac{p_{\mathbf{x}}^+(\mathbf{y})}{p_{\mathbf{x}}^-(\mathbf{y})} - \frac{p_{\mathbf{x}}^+(\mathbf{y}')}{p_{\mathbf{x}}^-(\mathbf{y}')} \right) (\mathbb{I}[s(\mathbf{x}, \mathbf{y}) > s(\mathbf{x}, \mathbf{y}')] - \mathbb{I}[s(\mathbf{x}, \mathbf{y}') > s(\mathbf{x}, \mathbf{y})]) \right].$$

Applying Lemma B.1 to the above equation, we conclude that

$$\frac{1}{4} \mathbb{E}_{\mathbf{x}} \mathbb{E}_{\mathbf{y} \sim p_{\mathbf{x}}^-, \mathbf{y}' \sim p_{\mathbf{x}}^-} \left[ \mathbb{I} \left[ \left( \frac{p_{\mathbf{x}}^+(\mathbf{y})}{p_{\mathbf{x}}^-(\mathbf{y})} - \frac{p_{\mathbf{x}}^+(\mathbf{y}')}{p_{\mathbf{x}}^-(\mathbf{y}')} \right) (s(\mathbf{x}, \mathbf{y}) - s(\mathbf{x}, \mathbf{y}')) \le 0 \right] \left| \frac{p_{\mathbf{x}}^+(\mathbf{y})}{p_{\mathbf{x}}^-(\mathbf{y})} - \frac{p_{\mathbf{x}}^+(\mathbf{y}')}{p_{\mathbf{x}}^-(\mathbf{y}')} \right| \right]$$
$$\le \mathcal{E}^* - \mathcal{E}(s) \tag{26}$$
$$\le \frac{1}{2} \mathbb{E}_{\mathbf{x}} \mathbb{E}_{\mathbf{y} \sim p_{\mathbf{x}}^-, \mathbf{y}' \sim p_{\mathbf{x}}^-} \left[ \mathbb{I} \left[ \left( \frac{p_{\mathbf{x}}^+(\mathbf{y})}{p_{\mathbf{x}}^-(\mathbf{y})} - \frac{p_{\mathbf{x}}^+(\mathbf{y}')}{p_{\mathbf{x}}^-(\mathbf{y}')} \right) (s(\mathbf{x}, \mathbf{y}) - s(\mathbf{x}, \mathbf{y}')) \le 0 \right] \left| \frac{p_{\mathbf{x}}^+(\mathbf{y})}{p_{\mathbf{x}}^-(\mathbf{y})} - \frac{p_{\mathbf{x}}^+(\mathbf{y}')}{p_{\mathbf{x}}^-(\mathbf{y}')} \right| \right].$$

Now we derive the necessary and sufficient condition for the optimal scoring function of $\mathcal{E}$.

**Necessary condition.** If $s$ is an optimal scoring function such that $\mathcal{E}^* - \mathcal{E}(s) = 0$, the by Eq. (26), there holds

$$0 \le \frac{1}{4} \mathbb{E}_{\mathbf{x}} \mathbb{E}_{\mathbf{y} \sim p_{\mathbf{x}}^-, \mathbf{y}' \sim p_{\mathbf{x}}^-} \left[ \mathbb{I} \left[ \left( \frac{p_{\mathbf{x}}^+(\mathbf{y})}{p_{\mathbf{x}}^-(\mathbf{y})} - \frac{p_{\mathbf{x}}^+(\mathbf{y}')}{p_{\mathbf{x}}^-(\mathbf{y}')} \right) (s(\mathbf{x}, \mathbf{y}) - s(\mathbf{x}, \mathbf{y}')) \le 0 \right] \left| \frac{p_{\mathbf{x}}^+(\mathbf{y})}{p_{\mathbf{x}}^-(\mathbf{y})} - \frac{p_{\mathbf{x}}^+(\mathbf{y}')}{p_{\mathbf{x}}^-(\mathbf{y}')} \right| \right]$$
$$\le \mathcal{E}^* - \mathcal{E}(s) \le 0.$$

Therefore,

$$\mathbb{E}_{\mathbf{x}} \mathbb{E}_{\mathbf{y} \sim p_{\mathbf{x}}^-, \mathbf{y}' \sim p_{\mathbf{x}}^-} \left[ \mathbb{I} \left[ \left( \frac{p_{\mathbf{x}}^+(\mathbf{y})}{p_{\mathbf{x}}^-(\mathbf{y})} - \frac{p_{\mathbf{x}}^+(\mathbf{y}')}{p_{\mathbf{x}}^-(\mathbf{y}')} \right) (s(\mathbf{x}, \mathbf{y}) - s(\mathbf{x}, \mathbf{y}')) \le 0 \right] \left| \frac{p_{\mathbf{x}}^+(\mathbf{y})}{p_{\mathbf{x}}^-(\mathbf{y})} - \frac{p_{\mathbf{x}}^+(\mathbf{y}')}{p_{\mathbf{x}}^-(\mathbf{y}')} \right| \right] = 0,$$

implying that for any $\mathbf{x}, \mathbf{y}, \mathbf{y}'$ with $p_{\mathbf{x}}^+(\mathbf{y})/p_{\mathbf{x}}^-(\mathbf{y}) \ne p_{\mathbf{x}}^+(\mathbf{y}')/p_{\mathbf{x}}^-(\mathbf{y}')$, there holds

$$\mathbb{I} \left[ \left( \frac{p_{\mathbf{x}}^+(\mathbf{y})}{p_{\mathbf{x}}^-(\mathbf{y})} - \frac{p_{\mathbf{x}}^+(\mathbf{y}')}{p_{\mathbf{x}}^-(\mathbf{y}')} \right) (s(\mathbf{x}, \mathbf{y}) - s(\mathbf{x}, \mathbf{y}')) \le 0 \right] = 0,$$
$$\implies \left( \frac{p_{\mathbf{x}}^+(\mathbf{y})}{p_{\mathbf{x}}^-(\mathbf{y})} - \frac{p_{\mathbf{x}}^+(\mathbf{y}')}{p_{\mathbf{x}}^-(\mathbf{y}')} \right) (s(\mathbf{x}, \mathbf{y}) - s(\mathbf{x}, \mathbf{y}')) > 0.$$

**Sufficient condition.** Let a scoring function $s$ satisfies that, for any $\mathbf{x}, \mathbf{y}, \mathbf{y}'$ with $p_{\mathbf{x}}^+(\mathbf{y})/p_{\mathbf{x}}^-(\mathbf{y}) \ne p_{\mathbf{x}}^+(\mathbf{y}')/p_{\mathbf{x}}^-(\mathbf{y}')$, there holds

$$\left( \frac{p_{\mathbf{x}}^+(\mathbf{y})}{p_{\mathbf{x}}^-(\mathbf{y})} - \frac{p_{\mathbf{x}}^+(\mathbf{y}')}{p_{\mathbf{x}}^-(\mathbf{y}')} \right) (s(\mathbf{x}, \mathbf{y}) - s(\mathbf{x}, \mathbf{y}')) > 0.$$

Then for any $\mathbf{x}, \mathbf{y}, \mathbf{y}'$, there holds

$$\mathbb{I} \left[ \left( \frac{p_{\mathbf{x}}^+(\mathbf{y})}{p_{\mathbf{x}}^-(\mathbf{y})} - \frac{p_{\mathbf{x}}^+(\mathbf{y}')}{p_{\mathbf{x}}^-(\mathbf{y}')} \right) (s(\mathbf{x}, \mathbf{y}) - s(\mathbf{x}, \mathbf{y}')) \le 0 \right] \left| \frac{p_{\mathbf{x}}^+(\mathbf{y})}{p_{\mathbf{x}}^-(\mathbf{y})} - \frac{p_{\mathbf{x}}^+(\mathbf{y}')}{p_{\mathbf{x}}^-(\mathbf{y}')} \right| = 0.$$

Then according to Eq. (26), we have

$$\mathcal{E}^* - \mathcal{E}(s)$$
$$\le \frac{1}{2} \mathbb{E}_{\mathbf{x}} \mathbb{E}_{\mathbf{y} \sim p_{\mathbf{x}}^-, \mathbf{y}' \sim p_{\mathbf{x}}^-} \left[ \mathbb{I} \left[ \left( \frac{p_{\mathbf{x}}^+(\mathbf{y})}{p_{\mathbf{x}}^-(\mathbf{y})} - \frac{p_{\mathbf{x}}^+(\mathbf{y}')}{p_{\mathbf{x}}^-(\mathbf{y}')} \right) (s(\mathbf{x}, \mathbf{y}) - s(\mathbf{x}, \mathbf{y}')) \le 0 \right] \left| \frac{p_{\mathbf{x}}^+(\mathbf{y})}{p_{\mathbf{x}}^-(\mathbf{y})} - \frac{p_{\mathbf{x}}^+(\mathbf{y}')}{p_{\mathbf{x}}^-(\mathbf{y}')} \right| \right] = 0,$$

indicating that $\mathcal{E}^* - \mathcal{E}(s) = 0$. This shows that $s$ is an optimal scoring function.

Hence, $s$ is an optimal scoring function *if and only if* for any $\mathbf{x}, \mathbf{y}, \mathbf{y}'$ with $p_\mathbf{x}^+(\mathbf{y})/p_\mathbf{x}^-(\mathbf{y}) \neq p_\mathbf{x}^+(\mathbf{y}')/p_\mathbf{x}^-(\mathbf{y}')$, there holds

$$\left( \frac{p_\mathbf{x}^+(\mathbf{y})}{p_\mathbf{x}^-(\mathbf{y})} - \frac{p_\mathbf{x}^+(\mathbf{y}')}{p_\mathbf{x}^-(\mathbf{y}')} \right) (s(\mathbf{x}, \mathbf{y}) - s(\mathbf{x}, \mathbf{y}')) > 0.$$

The proof is completed. $\qquad\qquad\square$

### B.3. Proof of Theorem 3.4

Before we provide the proof of Theorem 3.4, we introduce Pinsker's inequality which relates KL divergence to total variation.

**Lemma B.2** (Pinsker's Inequality). *Let $\mu$ and $\nu$ be two probability measures on a measurable space $(\Omega, \mathcal{F})$, with $\nu \ll \mu$ (i.e., $\nu$ is absolutely continuous with respect to $\mu$) so that the Radon-Nikodym derivative $\frac{d\nu}{d\mu}$ exists. The KL divergence of $\nu$ with respect to $\mu$ is defined as:*

$$D_{KL}(\nu \parallel \mu) = \int_\Omega \log \left( \frac{d\nu}{d\mu} \right) d\nu,$$

*where $D_{KL}(\nu \parallel \mu) = +\infty$ if the integral is infinite. The total variation distance between $\mu$ and $\nu$ is:*

$$\|\mu - \nu\|_{TV} = \sup_{A \in \mathcal{F}} |\mu(A) - \nu(A)| = \frac{1}{2} \int_\Omega \left| \frac{d\nu}{d\mu} - 1 \right| d\mu.$$

*Pinsker's inequality relates these two quantities by the bound:*

$$\|\mu - \nu\|_{TV}^2 \leq \frac{1}{2} D_{KL}(\nu \parallel \mu).$$

*Proof of Theorem 3.4.* According to Lemma 3.2, we choose an optimal scoring function

$$s_*(\mathbf{x}, \mathbf{y}) = \tau \log \frac{p_\mathbf{x}^+(\mathbf{y})}{p_\mathbf{x}^-(\mathbf{y})}. \tag{27}$$

Then $\mathcal{L}^* = \mathcal{L}(s_*) = \tau \mathbb{E}_\mathbf{x} \mathbb{E}_{\mathbf{y} \sim p_\mathbf{x}^+} \log \frac{p_\mathbf{x}^-(\mathbf{y})}{p_\mathbf{x}^+(\mathbf{y})}$. For a scoring function $s$, we define

$$\tilde{s}(\mathbf{x}, \mathbf{y}) = s(\mathbf{x}, \mathbf{y}) - \tau \log \mathbb{E}_{\mathbf{y}' \sim p_\mathbf{x}^-} \exp(s(\mathbf{x}, \mathbf{y}')/\tau).$$

Then $\mathbb{E}_{\mathbf{y}' \sim p_\mathbf{x}^-} \exp(\tilde{s}(\mathbf{x}, \mathbf{y}')/\tau) = 1$. Let $g(\mathbf{x}) = \tau \log \mathbb{E}_{\mathbf{y}' \sim p_\mathbf{x}^-} \exp(s(\mathbf{x}, \mathbf{y}')/\tau)$. We have

$$\begin{aligned}
\mathcal{E}(\tilde{s}) &= \mathbb{E}_\mathbf{x} \mathbb{E}_{\mathbf{y} \sim p_\mathbf{x}^+, \mathbf{y}' \sim p_\mathbf{x}^-} \mathbb{I}[\tilde{s}(\mathbf{x}, \mathbf{y}) > \tilde{s}(\mathbf{x}, \mathbf{y}')] = \mathbb{E}_\mathbf{x} \mathbb{E}_{\mathbf{y} \sim p_\mathbf{x}^+, \mathbf{y}' \sim p_\mathbf{x}^-} \mathbb{I}[s(\mathbf{x}, \mathbf{y}) - g(\mathbf{x}) > s(\mathbf{x}, \mathbf{y}') - g(\mathbf{x})] \\
&= \mathbb{E}_\mathbf{x} \mathbb{E}_{\mathbf{y} \sim p_\mathbf{x}^+, \mathbf{y}' \sim p_\mathbf{x}^-} \mathbb{I}[s(\mathbf{x}, \mathbf{y}) > s(\mathbf{x}, \mathbf{y}')] = \mathcal{E}(s).
\end{aligned}$$

There also holds

$$\begin{aligned}
\mathcal{L}(\tilde{s}) &= \tau \mathbb{E}_\mathbf{x} \mathbb{E}_{\mathbf{y} \sim p_\mathbf{x}^+} \log \mathbb{E}_{\mathbf{y}' \sim p_\mathbf{x}^-} \exp((\tilde{s}(\mathbf{x}, \mathbf{y}') - \tilde{s}(\mathbf{x}, \mathbf{y}))/\tau) \\
&= \tau \mathbb{E}_\mathbf{x} \mathbb{E}_{\mathbf{y} \sim p_\mathbf{x}^+} \log \mathbb{E}_{\mathbf{y}' \sim p_\mathbf{x}^-} \exp((s(\mathbf{x}, \mathbf{y}') - g(\mathbf{x}) + g(\mathbf{x}) - s(\mathbf{x}, \mathbf{y}))/\tau) \\
&= \tau \mathbb{E}_\mathbf{x} \mathbb{E}_{\mathbf{y} \sim p_\mathbf{x}^+} \log \mathbb{E}_{\mathbf{y}' \sim p_\mathbf{x}^-} \exp((s(\mathbf{x}, \mathbf{y}') - s(\mathbf{x}, \mathbf{y}))/\tau) = \mathcal{L}(s).
\end{aligned}$$

Therefore, $\mathcal{E}(\tilde{s}) = \mathcal{E}(s), \mathcal{L}(\tilde{s}) = \mathcal{L}(s)$ we can replace $s$ by $\tilde{s}$. In other words, without loss of generality, we can make the following regularity assumption

$$\mathbb{E}_{\mathbf{y}' \sim p_\mathbf{x}^-} \exp(s(\mathbf{x}, \mathbf{y}')/\tau) = 1. \tag{28}$$

Let the distribution $q_\mathbf{x}^+$ be defined in Eq. (22), i.e.,

$$q_\mathbf{x}^+(\mathbf{y}) = \frac{p_\mathbf{x}^-(\mathbf{y}) \exp(s(\mathbf{x}, \mathbf{y})/\tau)}{\mathbb{E}_{\mathbf{y}' \sim p_\mathbf{x}^-} \exp(s(\mathbf{x}, \mathbf{y}')/\tau)} = p_\mathbf{x}^-(\mathbf{y}) \exp(s(\mathbf{x}, \mathbf{y})/\tau).$$

By Eq. (23), we have

$$\mathcal{L}(s) - \mathcal{L}(s_*) = \tau \mathbb{E}_{\mathbf{x}} D_{\mathrm{KL}}(p_{\mathbf{x}}^+ \| q_{\mathbf{x}}^+).$$

According to Pinsker's inequality (Lemma B.2), we can bound the KL-divergence by total variation distance:

$$D_{\mathrm{KL}}(p_{\mathbf{x}}^+ \| q_{\mathbf{x}}^+) \geq \frac{1}{2} \left( \int_{\mathcal{Y}} |p_{\mathbf{x}}^+(\mathbf{y}) - q_{\mathbf{x}}^+(\mathbf{y})| d\mathbf{y} \right)^2.$$

Therefore,

$$\mathcal{L}(s) - \mathcal{L}(s_*) \geq \mathbb{E}_{\mathbf{x}} \frac{\tau}{2} \left( \int_{\mathcal{Y}} |p_{\mathbf{x}}^+(\mathbf{y}) - q_{\mathbf{x}}^+(\mathbf{y})| d\mathbf{y} \right)^2 = \mathbb{E}_{\mathbf{x}} \frac{\tau}{2} \left( \int_{\mathcal{Y}} |p_{\mathbf{x}}^+(\mathbf{y}) - p_{\mathbf{x}}^-(\mathbf{y}) \exp(s(\mathbf{x},\mathbf{y})/\tau)| d\mathbf{y} \right)^2$$

$$= \mathbb{E}_{\mathbf{x}} \frac{\tau}{2} \left( \int_{\mathcal{Y}} p_{\mathbf{x}}^-(\mathbf{y}) \left| \frac{p_{\mathbf{x}}^+(\mathbf{y})}{p_{\mathbf{x}}^-(\mathbf{y})} - \exp(s(\mathbf{x},\mathbf{y})/\tau) \right| d\mathbf{y} \right)^2$$

$$= \mathbb{E}_{\mathbf{x}} \frac{\tau}{2} \left( \int_{\mathcal{Y}} p_{\mathbf{x}}^-(\mathbf{y}) \left| \exp(s_*(\mathbf{x},\mathbf{y})/\tau)) - \exp(s(\mathbf{x},\mathbf{y})/\tau) \right| d\mathbf{y} \right)^2$$

$$= \mathbb{E}_{\mathbf{x}} \frac{\tau}{2} \left( \mathbb{E}_{\mathbf{y} \sim p_{\mathbf{x}}^-} \left| \exp(s_*(\mathbf{x},\mathbf{y})/\tau)) - \exp(s(\mathbf{x},\mathbf{y})/\tau) \right| \right)^2, \tag{29}$$

where the third equality is due to Eq. (27).

Then we make the following decomposition that introduces the pair $\mathbf{y}, \mathbf{y}'$:

$$2\mathbb{E}_{\mathbf{y} \sim p_{\mathbf{x}}^-} | \exp(s_*(\mathbf{x},\mathbf{y})/\tau)) - \exp(s(\mathbf{x},\mathbf{y})/\tau)|$$
$$= \mathbb{E}_{\mathbf{y} \sim p_{\mathbf{x}}^-} | \exp(s_*(\mathbf{x},\mathbf{y})/\tau)) - \exp(s(\mathbf{x},\mathbf{y})/\tau)| + \mathbb{E}_{\mathbf{y}' \sim p_{\mathbf{x}}^-} | \exp(s_*(\mathbf{x},\mathbf{y}')/\tau)) - \exp(s(\mathbf{x},\mathbf{y}')/\tau)|$$
$$= \mathbb{E}_{\mathbf{y} \sim p_{\mathbf{x}}^-, \mathbf{y}' \sim p_{\mathbf{x}}^-} [| \exp(s_*(\mathbf{x},\mathbf{y})/\tau)) - \exp(s(\mathbf{x},\mathbf{y})/\tau)| + | \exp(s_*(\mathbf{x},\mathbf{y}')/\tau)) - \exp(s(\mathbf{x},\mathbf{y}')/\tau)|]. \tag{30}$$

We will show that for any $\mathbf{x}, \mathbf{y}, \mathbf{y}'$,

$$|\exp(s_*(\mathbf{x},\mathbf{y})/\tau)) - \exp(s(\mathbf{x},\mathbf{y})/\tau)| + |\exp(s_*(\mathbf{x},\mathbf{y}')/\tau)) - \exp(s(\mathbf{x},\mathbf{y}')/\tau)|$$

$$\geq \mathbb{I} \left[ \left( \frac{p_{\mathbf{x}}^+(\mathbf{y})}{p_{\mathbf{x}}^-(\mathbf{y})} - \frac{p_{\mathbf{x}}^+(\mathbf{y}')}{p_{\mathbf{x}}^-(\mathbf{y}')} \right) (s(\mathbf{x},\mathbf{y}) - s(\mathbf{x},\mathbf{y}')) \leq 0 \right] \left| \frac{p_{\mathbf{x}}^+(\mathbf{y})}{p_{\mathbf{x}}^-(\mathbf{y})} - \frac{p_{\mathbf{x}}^+(\mathbf{y}')}{p_{\mathbf{x}}^-(\mathbf{y}')} \right|. \tag{31}$$

It is trivial when RHS is equal to $0$. We consider the case

$$\left( \frac{p_{\mathbf{x}}^+(\mathbf{y})}{p_{\mathbf{x}}^-(\mathbf{y})} - \frac{p_{\mathbf{x}}^+(\mathbf{y}')}{p_{\mathbf{x}}^-(\mathbf{y}')} \right) (s(\mathbf{x},\mathbf{y}) - s(\mathbf{x},\mathbf{y}')) \leq 0, \quad \frac{p_{\mathbf{x}}^+(\mathbf{y})}{p_{\mathbf{x}}^-(\mathbf{y})} - \frac{p_{\mathbf{x}}^+(\mathbf{y}')}{p_{\mathbf{x}}^-(\mathbf{y}')} \neq 0.$$

Without loss of generality, suppose that

$$\frac{p_{\mathbf{x}}^+(\mathbf{y})}{p_{\mathbf{x}}^-(\mathbf{y})} > \frac{p_{\mathbf{x}}^+(\mathbf{y}')}{p_{\mathbf{x}}^-(\mathbf{y}')}, \quad s(\mathbf{x},\mathbf{y}) - s(\mathbf{x},\mathbf{y}') \leq 0.$$

By the definition of $s_*$ in Eq. (27), we have

$$\exp(s_*(\mathbf{x},\mathbf{y})/\tau) - \exp(s_*(\mathbf{x},\mathbf{y}')/\tau) = \frac{p_{\mathbf{x}}^+(\mathbf{y})}{p_{\mathbf{x}}^-(\mathbf{y})} - \frac{p_{\mathbf{x}}^+(\mathbf{y}')}{p_{\mathbf{x}}^-(\mathbf{y}')} > 0. \tag{32}$$

and $\exp(s(\mathbf{x},\mathbf{y}')/\tau)) - \exp(s(\mathbf{x},\mathbf{y})/\tau) \geq 0$. Therefore,

$$|\exp(s_*(\mathbf{x},\mathbf{y})/\tau)) - \exp(s(\mathbf{x},\mathbf{y})/\tau)| + |\exp(s_*(\mathbf{x},\mathbf{y}')/\tau)) - \exp(s(\mathbf{x},\mathbf{y}')/\tau)|$$
$$\geq |\exp(s_*(\mathbf{x},\mathbf{y})/\tau)) - \exp(s(\mathbf{x},\mathbf{y})/\tau) - \exp(s_*(\mathbf{x},\mathbf{y}')/\tau)) + \exp(s(\mathbf{x},\mathbf{y}')/\tau)|$$
$$= |(\exp(s_*(\mathbf{x},\mathbf{y})/\tau) - \exp(s_*(\mathbf{x},\mathbf{y}')/\tau)) + (\exp(s(\mathbf{x},\mathbf{y}')/\tau) - \exp(s(\mathbf{x},\mathbf{y})/\tau))|$$
$$= (\exp(s_*(\mathbf{x},\mathbf{y})/\tau) - \exp(s_*(\mathbf{x},\mathbf{y}')/\tau)) + (\exp(s(\mathbf{x},\mathbf{y}')/\tau) - \exp(s(\mathbf{x},\mathbf{y})/\tau))$$
$$\geq \exp(s_*(\mathbf{x},\mathbf{y})/\tau) - \exp(s_*(\mathbf{x},\mathbf{y}')/\tau) = \frac{p_{\mathbf{x}}^+(\mathbf{y})}{p_{\mathbf{x}}^-(\mathbf{y})} - \frac{p_{\mathbf{x}}^+(\mathbf{y}')}{p_{\mathbf{x}}^-(\mathbf{y}')}.$$

As a result, Eq. (31) holds true. Plugging Eq. (31) into Eq. (30) yields

$$
\mathbb{E}_{\mathbf{y}\sim p_{\mathbf{x}}^-} |\exp(s_*(\mathbf{x},\mathbf{y})/\tau)) - \exp(s(\mathbf{x},\mathbf{y})/\tau)|
$$

$$
\geq \frac{1}{2}\mathbb{E}_{\mathbf{y}\sim p_{\mathbf{x}}^-,\mathbf{y}'\sim p_{\mathbf{x}}^-} \mathbb{I}\left[\left(\frac{p_{\mathbf{x}}^+(\mathbf{y})}{p_{\mathbf{x}}^-(\mathbf{y})} - \frac{p_{\mathbf{x}}^+(\mathbf{y}')}{p_{\mathbf{x}}^-(\mathbf{y}')}\right)(s(\mathbf{x},\mathbf{y}) - s(\mathbf{x},\mathbf{y}')) \leq 0\right] \left|\frac{p_{\mathbf{x}}^+(\mathbf{y})}{p_{\mathbf{x}}^-(\mathbf{y})} - \frac{p_{\mathbf{x}}^+(\mathbf{y}')}{p_{\mathbf{x}}^-(\mathbf{y}')}\right|. \tag{33}
$$

Combining Eqs. (29), (33), we have

$$
\mathcal{L}(s) - \mathcal{L}(s_*) \geq \frac{\tau}{2}\mathbb{E}_{\mathbf{x}}\left(\mathbb{E}_{\mathbf{y}\sim p_{\mathbf{x}}^-} |\exp(s_*(\mathbf{x},\mathbf{y})/\tau)) - \exp(s(\mathbf{x},\mathbf{y})/\tau)|\right)^2
$$

$$
\geq \frac{\tau}{2}\mathbb{E}_{\mathbf{x}}\left(\frac{1}{2}\mathbb{E}_{\mathbf{y}\sim p_{\mathbf{x}}^-,\mathbf{y}'\sim p_{\mathbf{x}}^-}\mathbb{I}\left[\left(\frac{p_{\mathbf{x}}^+(\mathbf{y})}{p_{\mathbf{x}}^-(\mathbf{y})} - \frac{p_{\mathbf{x}}^+(\mathbf{y}')}{p_{\mathbf{x}}^-(\mathbf{y}')}\right)(s(\mathbf{x},\mathbf{y}) - s(\mathbf{x},\mathbf{y}')) \leq 0\right]\left|\frac{p_{\mathbf{x}}^+(\mathbf{y})}{p_{\mathbf{x}}^-(\mathbf{y})} - \frac{p_{\mathbf{x}}^+(\mathbf{y}')}{p_{\mathbf{x}}^-(\mathbf{y}')}\right|\right)^2
$$

$$
\geq \frac{\tau}{2}\left(\frac{1}{2}\mathbb{E}_{\mathbf{x}}\mathbb{E}_{\mathbf{y}\sim p_{\mathbf{x}}^-,\mathbf{y}'\sim p_{\mathbf{x}}^-}\mathbb{I}[(\frac{p_{\mathbf{x}}^+(\mathbf{y})}{p_{\mathbf{x}}^-(\mathbf{y})} - \frac{p_{\mathbf{x}}^+(\mathbf{y}')}{p_{\mathbf{x}}^-(\mathbf{y}')})(s(\mathbf{x},\mathbf{y}) - s(\mathbf{x},\mathbf{y}')) \leq 0]\left|\frac{p_{\mathbf{x}}^+(\mathbf{y})}{p_{\mathbf{x}}^-(\mathbf{y})} - \frac{p_{\mathbf{x}}^+(\mathbf{y}')}{p_{\mathbf{x}}^-(\mathbf{y}')}\right|\right)^2
$$

$$
\geq \frac{\tau}{2}(\mathcal{E}^* - \mathcal{E}(s))^2,
$$

where the second inequality is due to Jensen's inequality and the last inequality results from Eq. (26). As a result,

$$
\mathcal{E}^* - \mathcal{E}(s) \leq \sqrt{2/\tau}\sqrt{\mathcal{L}(s) - \mathcal{L}(s_*)}.
$$

The proof is completed by noting that $s_*$ is a minimizer of $\mathcal{L}(s)$ and thus $\mathcal{L}(s_*) = \mathcal{L}^*$. □

## C. Proofs in Section 4.1

Recall the error decomposition

$$
\left|\mathcal{L}_{\mathrm{S}}(s_{\mathbf{w}}) - \widehat{\mathcal{L}}_{\mathrm{S}}(s_{\mathbf{w}})\right|
$$

$$
\leq \underbrace{\left|\mathcal{L}_{\mathrm{S}}(s_{\mathbf{w}}) - \mathbb{E}\widehat{\mathcal{L}}_{\mathrm{S}}(s_{\mathbf{w}})\right|}_{\text{inner error}} + \underbrace{\left|\mathbb{E}\widehat{\mathcal{L}}_{\mathrm{S}}(s_{\mathbf{w}}) - \widehat{\mathcal{L}}_{\mathrm{S}}(s_{\mathbf{w}})\right|}_{\text{outer error}}.
$$

We will control inner error and outer error separately. Before that, we provide the proof of Lemma 4.2.

### C.1. Proof of Lemma 4.2

*Proof of Lemma 4.2.* For $\mu_i \in \mathbb{R}$, we define

$$
d_i(\mu_i) = \frac{\tau}{m}\sum_{j=1}^m \exp\left(\frac{\Delta_{\mathbf{w}}(\mathbf{x}_i,\mathbf{y}_i,\mathbf{y}'_{ij}) - \mu_i}{\tau}\right) + \mu_i - \tau.
$$

Then we have

$$
d_i'(\mu_i) = \frac{\tau}{m}\sum_{j=1}^m \exp\left(\frac{\Delta_{\mathbf{w}}(\mathbf{x}_i,\mathbf{y}_i,\mathbf{y}'_{ij}) - \mu_i}{\tau}\right)\left(\frac{-1}{\tau}\right) + 1 = 1 - \frac{1}{m}\sum_{j=1}^m \exp\left(\frac{\Delta_{\mathbf{w}}(\mathbf{x}_i,\mathbf{y}_i,\mathbf{y}'_{ij}) - \mu_i}{\tau}\right).
$$

By taking $d_i'(\mu_i) = 0$, we obtain the minimizer of $d_i$:

$$
\mu_i^* = \tau \log \frac{1}{m}\sum_{j=1}^m \exp\left(\frac{\Delta_{\mathbf{w}}(\mathbf{x}_i,\mathbf{y}_i,\mathbf{y}'_{ij})}{\tau}\right).
$$

Plugging it into $d_i$, we have

$$\min_{\mu_i \in \mathbb{R}} \frac{\tau}{m} \sum_{j=1}^{m} \exp\left(\frac{\Delta_{\mathbf{w}}(\mathbf{x}_i, \mathbf{y}_i, \mathbf{y}'_{ij}) - \mu_i}{\tau}\right) + \mu_i - \tau = d_i(\mu_i^*) = \tau \log \frac{1}{m} \sum_{j=1}^{m} \exp\left(\frac{\Delta_{\mathbf{w}}(\mathbf{x}_i, \mathbf{y}_i, \mathbf{y}'_{ij})}{\tau}\right).$$

Since $s_{\mathbf{w}}(\mathbf{x}, \mathbf{y}) \in [-B, B]$ for all $\mathbf{x} \in \mathcal{X}, \mathbf{y} \in \mathcal{Y}, \mathbf{w} \in \mathcal{W}$, $\Delta_{\mathbf{w}}(\mathbf{x}_i, \mathbf{y}_i, \mathbf{y}'_{ij}) = s_{\mathbf{w}}(\mathbf{x}_i, \mathbf{y}'_{ij}) - s_{\mathbf{w}}(\mathbf{x}_i, \mathbf{y}_i) \in [-2B, 2B]$, implying that $\mu_i^* \in [-2B, 2B]$. Therefore,

$$\min_{\mu_i \in [-2B, 2B]} \frac{\tau}{m} \sum_{j=1}^{m} \exp\left(\frac{\Delta_{\mathbf{w}}(\mathbf{x}_i, \mathbf{y}_i, \mathbf{y}'_{ij}) - \mu_i}{\tau}\right) + \mu_i - \tau = \tau \log \frac{1}{m} \sum_{j=1}^{m} \exp\left(\frac{\Delta_{\mathbf{w}}(\mathbf{x}_i, \mathbf{y}_i, \mathbf{y}'_{ij})}{\tau}\right).$$

Taking summation over all $i \in [n]$, we get

$$\widehat{\mathcal{L}}_{\mathrm{S}}(s_{\mathbf{w}}) = \frac{1}{n} \sum_{i=1}^{n} \tau \log \frac{1}{m} \sum_{j=1}^{m} \exp\left(\frac{\Delta_{\mathbf{w}}(\mathbf{x}_i, \mathbf{y}_i, \mathbf{y}'_{ij})}{\tau}\right)$$

$$= -\tau + \frac{1}{n} \sum_{i=1}^{n} \left[ \min_{|\mu_i| \le 2B} \frac{\tau}{m} \sum_{j=1}^{m} \exp\left(\frac{\Delta(\mathbf{x}_i, \mathbf{y}_i, \mathbf{y}'_{ij}) - \mu_i}{\tau}\right) + \mu_i \right].$$

This completes the proof of Eq. (13).

To prove Eq. (14), we define

$$d(\mu) = \tau \mathbb{E}_{\mathbf{y}'} \exp\left(\frac{\Delta_{\mathbf{w}}(\mathbf{x}, \mathbf{y}, \mathbf{y}') - \mu}{\tau}\right) + \mu - \tau, \qquad \mu \in \mathbb{R}.$$

Then

$$d'(\mu) = -\mathbb{E}_{\mathbf{y}'} \exp\left(\frac{\Delta_{\mathbf{w}}(\mathbf{x}, \mathbf{y}, \mathbf{y}') - \mu}{\tau}\right) + 1.$$

Taking $d'(\mu) = 0$ obtains the optimal value $\mu^* = \tau \log \mathbb{E}_{\mathbf{y}'} \exp\left(\frac{\Delta_{\mathbf{w}}(\mathbf{x}, \mathbf{y}, \mathbf{y}')}{\tau}\right)$. Therefore,

$$\min_{\mu \in \mathbb{R}} \tau \mathbb{E}_{\mathbf{y}'} \exp\left(\frac{\Delta_{\mathbf{w}}(\mathbf{x}, \mathbf{y}, \mathbf{y}') - \mu}{\tau}\right) + \mu = d(\mu^*) = \tau \log \mathbb{E}_{\mathbf{y}'} \exp\left(\frac{\Delta_{\mathbf{w}}(\mathbf{x}, \mathbf{y}, \mathbf{y}')}{\tau}\right) - \tau.$$

By noting that $\Delta_{\mathbf{w}}(\mathbf{x}, \mathbf{y}, \mathbf{y}') = s_{\mathbf{w}}(\mathbf{x}, \mathbf{y}') - s_{\mathbf{w}}(\mathbf{x}, \mathbf{y}) \in [-2B, 2B]$, there holds

$$\min_{|\mu| \in [-2B, 2B]} \tau \mathbb{E}_{\mathbf{y}'} \exp\left(\frac{\Delta_{\mathbf{w}}(\mathbf{x}, \mathbf{y}, \mathbf{y}') - \mu}{\tau}\right) + \mu = \tau \log \mathbb{E}_{\mathbf{y}'} \exp\left(\frac{\Delta_{\mathbf{w}}(\mathbf{x}, \mathbf{y}, \mathbf{y}')}{\tau}\right) - \tau.$$

Taking expectation w.r.t. $\mathbf{x}, \mathbf{y}$ and combining with Eq. (5) complete the proof of the lemma. $\square$

### C.2. Proof of Lemma 4.3

*Proof of Lemma 4.3.* By Eqs. (13), (14), there holds

$$\mathcal{L}_{\mathrm{S}}(s_{\mathbf{w}}) - \mathbb{E}\widehat{\mathcal{L}}_{\mathrm{S}}(s_{\mathbf{w}})$$

$$= \mathbb{E}_{\mathbf{x}, \mathbf{y}} \left[ \min_{\mu \in [-2B, 2B]} \tau \mathbb{E}_{\mathbf{y}'} \exp\left(\frac{\Delta_{\mathbf{w}}(\mathbf{x}, \mathbf{y}, \mathbf{y}') - \mu}{\tau}\right) - \tau + \mu \right]$$

$$- \mathbb{E}_{\{\mathbf{x}_i, \mathbf{y}_i, \mathbf{y}'_{ij}\}} \frac{1}{n} \sum_{i=1}^{n} \left[ \min_{\mu_i \in [-2B, 2B]} \frac{\tau}{m} \sum_{j=1}^{m} \exp\left(\frac{\Delta_{\mathbf{w}}(\mathbf{x}_i, \mathbf{y}_i, \mathbf{y}'_{ij}) - \mu_i}{\tau}\right) - \tau + \mu_i \right]$$

$$= \mathbb{E}_{\mathbf{x}, \mathbf{y}} \left\{ \left[ \min_{\mu \in [-2B, 2B]} \tau \mathbb{E}_{\mathbf{y}'} \exp\left(\frac{\Delta_{\mathbf{w}}(\mathbf{x}, \mathbf{y}, \mathbf{y}') - \mu}{\tau}\right) - \tau + \mu \right] \right.$$

$$\left. - \mathbb{E}_{\{\mathbf{y}'_j\}} \left[ \min_{\mu \in [-2B, 2B]} \frac{\tau}{m} \sum_{j=1}^{m} \exp\left(\frac{\Delta_{\mathbf{w}}(\mathbf{x}, \mathbf{y}, \mathbf{y}'_j) - \mu}{\tau}\right) - \tau + \mu \right] \right\}.$$

Let $\psi_{\mathbf{w},\mathbf{x},\mathbf{y}}(\mu;\mathbf{y}') = \mu + \tau[\exp(\Delta_{\mathbf{w}}(\mathbf{x},\mathbf{y},\mathbf{y}') - \mu)/\tau) - 1]$. Since $\Delta_{\mathbf{w}}(\mathbf{x},\mathbf{y},\mathbf{y}') \in [-2B, 2B]$, $\psi_{\mathbf{w},\mathbf{x},\mathbf{y}}(\mu;\mathbf{y}')$ is $\exp(-4B/\tau)/\tau$-strongly convex and $\exp(4B/\tau)$-Lipschitz w.r.t. $\mu$ over the interval $[-2B, 2B]$. Therefore,

$$\mathcal{L}_{\mathrm{S}}(\mathbf{w}) - \mathbb{E}\widehat{\mathcal{L}}_{\mathrm{S}}(\mathbf{w}) = \mathbb{E}_{\mathbf{x},\mathbf{y}}\left[\min_{\mu\in[-2B,2B]}\mathbb{E}_{\mathbf{y}'}\psi_{\mathbf{w},\mathbf{x},\mathbf{y}}(\mu;\mathbf{y}') - \mathbb{E}_{\{\mathbf{y}'_j\}}\min_{\mu\in[-2B,2B]}\frac{1}{m}\sum_{j=1}^{m}\psi_{\mathbf{w},\mathbf{x},\mathbf{y}}(\mu;\mathbf{y}'_j)\right].$$

For simplicity, we define $S' = \{\mathbf{y}'_1, \cdots, \mathbf{y}'_m\}$, and

$$\bar{\mu}_{\mathbf{w},\mathbf{x},\mathbf{y}}(S') = \underset{\mu\in[-2B,2B]}{\arg\min}\frac{1}{m}\sum_{j=1}^{m}\psi_{\mathbf{w},\mathbf{x},\mathbf{y}}(\mu;\mathbf{y}'_j).$$

Then for any $S'$,

$$\min_{\mu\in[-2B,2B]}\mathbb{E}_{\mathbf{y}'}\psi_{\mathbf{w},\mathbf{x},\mathbf{y}}(\mu;\mathbf{y}') \le \mathbb{E}_{\mathbf{y}'}\psi_{\mathbf{w},\mathbf{x},\mathbf{y}}(\bar{\mu}(\{S'\});\mathbf{y}').$$

Taking expectation for $S'$ yields

$$\min_{\mu\in[-2B,2B]}\mathbb{E}_{\mathbf{y}'}\psi_{\mathbf{w},\mathbf{x},\mathbf{y}}(\mu;\mathbf{y}') \le \mathbb{E}_{S'}\mathbb{E}_{\mathbf{y}'}\psi_{\mathbf{w},\mathbf{x},\mathbf{y}}(\bar{\mu}(S');\mathbf{y}'). \tag{34}$$

By the $\exp(-4B/\tau)/\tau$-strong convexity of $\psi_{\mathbf{w},\mathbf{x},\mathbf{y}}(\mu;\mathbf{y}')$, we can define a convex function

$$\hat{\psi}_{\mathbf{w},\mathbf{x},\mathbf{y}}(\mu;\mathbf{y}') := \psi_{\mathbf{w},\mathbf{x},\mathbf{y}}(\mu;\mathbf{y}') - \frac{\exp(-4B/\tau)}{2\tau}\mu^2.$$

Then

$$\bar{\mu}_{\mathbf{w},\mathbf{x},\mathbf{y}}(S') = \underset{\mu\in[-2B,2B]}{\arg\min}\frac{1}{m}\sum_{j=1}^{m}\psi_{\mathbf{w},\mathbf{x},\mathbf{y}}(\mu;\mathbf{y}'_j) = \underset{\mu\in[-2B,2B]}{\arg\min}\left\{\frac{1}{m}\sum_{j=1}^{m}\hat{\psi}_{\mathbf{w},\mathbf{x},\mathbf{y}}(\mu;\mathbf{y}'_j) + \frac{\exp(-4B/\tau)}{2\tau}\mu^2\right\},$$

which is the regularized empirical risk minimization algorithm. According to Theorem 12, 22 in Bousquet & Elisseeff (2002), for all $\mathbf{x}, \mathbf{y}, \mathbf{w}$, we have

$$\mathbb{E}_{S'}\left[\mathbb{E}_{\mathbf{y}'}\hat{\psi}_{\mathbf{w},\mathbf{x},\mathbf{y}}(\bar{\mu}_{\mathbf{w},\mathbf{x},\mathbf{y}}(S');\mathbf{y}') - \frac{1}{m}\sum_{j=1}^{m}\hat{\psi}_{\mathbf{w},\mathbf{x},\mathbf{y}}(\bar{\mu}_{\mathbf{w},\mathbf{x},\mathbf{y}}(S');\mathbf{y}'_j)\right] \le \frac{2\exp(4B/\tau)\tau}{\exp(-4B/\tau)m} = \frac{2\exp(8B/\tau)\tau}{m}.$$

As a result,

$$\mathbb{E}_{S'}\left[\mathbb{E}_{\mathbf{y}'}\psi_{\mathbf{w},\mathbf{x},\mathbf{y}}(\bar{\mu}_{\mathbf{w},\mathbf{x},\mathbf{y}}(S');\mathbf{y}') - \frac{1}{m}\sum_{j=1}^{m}\psi_{\mathbf{w},\mathbf{x},\mathbf{y}}(\bar{\mu}_{\mathbf{w},\mathbf{x},\mathbf{y}}(S');\mathbf{y}'_j)\right]$$

$$=\mathbb{E}_{S'}\left[\mathbb{E}_{\mathbf{y}'}\hat{\psi}_{\mathbf{w},\mathbf{x},\mathbf{y}}(\bar{\mu}_{\mathbf{w},\mathbf{x},\mathbf{y}}(S');\mathbf{y}') + \frac{\exp(-4B/\tau)}{2\tau}(\bar{\mu}_{\mathbf{w},\mathbf{x},\mathbf{y}}(S'))^2\right.$$

$$\left.-\frac{1}{m}\sum_{j=1}^{m}\hat{\psi}_{\mathbf{w},\mathbf{x},\mathbf{y}}(\bar{\mu}_{\mathbf{w},\mathbf{x},\mathbf{y}}(S');\mathbf{y}'_j) - \frac{\exp(-4B/\tau)}{2\tau}(\bar{\mu}_{\mathbf{w},\mathbf{x},\mathbf{y}}(S'))^2\right]$$

$$=\mathbb{E}_{S'}\left[\mathbb{E}_{\mathbf{y}'}\hat{\psi}_{\mathbf{w},\mathbf{x},\mathbf{y}}(\bar{\mu}_{\mathbf{w},\mathbf{x},\mathbf{y}}(S');\mathbf{y}') - \frac{1}{m}\sum_{j=1}^{m}\hat{\psi}_{\mathbf{w},\mathbf{x},\mathbf{y}}(\bar{\mu}_{\mathbf{w},\mathbf{x},\mathbf{y}}(S');\mathbf{y}'_j)\right] \le \frac{2\exp(8B/\tau)\tau}{m}.$$

Combined with Eq. (34), for any $\mathbf{x}, \mathbf{y}, \mathbf{w}$, there holds

$$\min_{\mu\in[-2B,2B]}\mathbb{E}_{\mathbf{y}'}\psi_{\mathbf{w},\mathbf{x},\mathbf{y}}(\mu;\mathbf{y}') - \mathbb{E}_{\mathbf{y}'_j}\min_{\mu\in[-2B,2B]}\frac{1}{m}\sum_{j=1}^{m}\psi_{\mathbf{w},\mathbf{x},\mathbf{y}}(\mu;\mathbf{y}'_j)$$

$$=\min_{\mu\in[-2B,2B]}\mathbb{E}_{\mathbf{y}'}\psi_{\mathbf{w},\mathbf{x},\mathbf{y}}(\mu;\mathbf{y}') - \mathbb{E}_{S'}\frac{1}{m}\sum_{j=1}^{m}\psi_{\mathbf{w},\mathbf{x},\mathbf{y}}(\bar{\mu}_{\mathbf{w},\mathbf{x},\mathbf{y}}(S');\mathbf{y}'_j)$$

$$\le\mathbb{E}_{S'}\left[\mathbb{E}_{\mathbf{y}}\psi_{\mathbf{w},\mathbf{x},\mathbf{y}}(\bar{\mu}_{\mathbf{w},\mathbf{x},\mathbf{y}}(S');\mathbf{y}') - \frac{1}{m}\sum_{j=1}^{m}\psi_{\mathbf{w},\mathbf{x},\mathbf{y}}(\bar{\mu}_{\mathbf{w},\mathbf{x},\mathbf{y}}(S');\mathbf{y}'_j)\right] \le \frac{2\exp(8B/\tau)\tau}{m}.$$

As a result,

$$
\begin{aligned}
\sup_{\mathbf{w}\in\mathcal{W}}\left[\mathcal{L}(\mathbf{w})-\mathbb{E}\widehat{\mathcal{L}}_S(\mathbf{w})\right] &= \sup_{\mathbf{w}\in\mathcal{W}}\mathbb{E}_{\mathbf{x},\mathbf{y}}\left[\min_{\mu\in[-2B,2B]}\mathbb{E}_{\mathbf{y}'}\psi_{\mathbf{w},\mathbf{x},\mathbf{y}}(\mu;\mathbf{y}') - \mathbb{E}_{\mathbf{y}'_j}\min_{\mu\in[-2B,2B]}\frac{1}{m}\sum_{j=1}^{m}\psi_{\mathbf{w},\mathbf{x},\mathbf{y}}(\mu;\mathbf{y}'_j)\right] \\
&\leq \frac{2\exp(8B/\tau)\tau}{m}.
\end{aligned}
$$

The proof is completed. $\qquad\square$

*Remark* C.1. The term $\exp(8B/\tau)$ arises from standard Lipschitz and strong convexity arguments applied to the exponential function over bounded domains, and is often standard in the generalization theory (Hardt et al., 2016). The exponential dependences also appear in previous theoretical analyses of CRL (Wang & Isola, 2020; Lei et al., 2023). In practical implementations such as CLIP and SimCLR, the scoring function typically uses cosine similarity, which naturally bounds $B \in [-1, 1]$. The primary focus of our work is to characterize how $m$ and $n$ influence the scaling behavior of generalization, rather than obtaining tight numerical constants. We believe it would be an interesting future problem to achieve tighter bounds and remove these exponential dependencies.

## C.3. Proof of Lemma 4.4

*Proof of Lemma 4.4.* For $\mathbf{w} \in \mathcal{W}$, we introduce $k_{\mathbf{w}} : \mathcal{X} \times \mathcal{Y}^{m+1} \to \mathbb{R}$ as follows:

$$
k_{\mathbf{w}}(\mathbf{x}, \mathbf{y}, \mathbf{y}'_1, \cdots, \mathbf{y}'_m) = \tau\log\frac{1}{m}\sum_{j=1}^{m}\exp\left(\frac{\Delta_{\mathbf{w}}(\mathbf{x}, \mathbf{y}, \mathbf{y}'_j)}{\tau}\right).
$$

Then we have

$$
\begin{aligned}
&\mathbb{E}\widehat{\mathcal{L}}_S(\mathbf{w}) - \widehat{\mathcal{L}}_S(\mathbf{w}) \\
=&\mathbb{E}_{\mathbf{x},\mathbf{y},\mathbf{y}'_1,\cdots,\mathbf{y}'_m}\frac{1}{n}\sum_{i=1}^{n}\left[\tau\log\frac{1}{m}\sum_{j=1}^{m}\exp\left(\frac{\Delta_{\mathbf{w}}(\mathbf{x},\mathbf{y},\mathbf{y}'_j)}{\tau}\right)\right] - \frac{1}{n}\sum_{i=1}^{n}\left[\tau\log\frac{1}{m}\sum_{j=1}^{m}\exp\left(\frac{\Delta_{\mathbf{w}}(\mathbf{x}_i,\mathbf{y}_i,\mathbf{y}'_{ij})}{\tau}\right)\right] \\
=&\mathbb{E}_{\mathbf{x},\mathbf{y},\mathbf{y}'_1,\cdots,\mathbf{y}'_m}\left[k_{\mathbf{w}}(\mathbf{x},\mathbf{y},\mathbf{y}'_1,\cdots,\mathbf{y}'_m)\right] - \frac{1}{n}\sum_{i=1}^{n}k_{\mathbf{w}}(\mathbf{x}_i,\mathbf{y}_i,\mathbf{y}'_{i1},\cdots,\mathbf{y}'_{im}).
\end{aligned}
$$

By Proposition A.1, it suffices to bound Rademacher complexity $\mathfrak{R}_S(\mathcal{K})$, where

$$
\begin{aligned}
\mathcal{K} &= \{(\mathbf{x},\mathbf{y},\mathbf{y}'_1,\cdots,\mathbf{y}'_m)\mapsto k_{\mathbf{w}}(\mathbf{x},\mathbf{y},\mathbf{y}'_1,\cdots,\mathbf{y}'_m) : \mathbf{w}\in\mathcal{W}\}, \\
S &= \{(\mathbf{x}_1,\mathbf{y}_1,\mathbf{y}'_{11},\cdots,\mathbf{y}'_{1m}),\cdots,(\mathbf{x}_n,\mathbf{y}_n,\mathbf{y}'_{n1},\cdots,\mathbf{y}'_{nm})\}.
\end{aligned}
$$

Recall that

$$
\begin{aligned}
S_1 &:= \{(\mathbf{x}_i,\mathbf{y}_i,\mathbf{y}^-_{ij}) : 1\leq i\leq n, 1\leq j\leq m\}, \\
\mathcal{H} &:= \{(\mathbf{x},\mathbf{y},\mathbf{y}')\mapsto\Delta_{\mathbf{w}}(\mathbf{x},\mathbf{y},\mathbf{y}') : \mathbf{w}\in\mathcal{W}\}.
\end{aligned}
$$

Let $k_{\mathbf{w}}, k_{\mathbf{w}'}$ be two functions in $\mathcal{K}$. By Lemma E.4, the logistic loss $\tau\log\frac{1}{m}\sum_{j=1}^{m}\exp(t_j/\tau)$ is $1$-$\ell_\infty$-Lipschitz, i.e.

$$
\left|\tau\log\frac{1}{m}\sum_{j=1}^{m}\exp(t_j/\tau) - \tau\log\frac{1}{m}\sum_{j=1}^{m}\exp(t'_j/\tau)\right| \leq \max_{j}|t_j - t'_j|.
$$

Therefore,

$$
\begin{aligned}
\|k_{\mathbf{w}} - k_{\mathbf{w}'}\|_{L_2(S)} &= \sqrt{\frac{1}{n}\sum_{i=1}^{n}(k_{\mathbf{w}}(\mathbf{x}_i,\mathbf{y}_i,\mathbf{y}'_{i1},\cdots,\mathbf{y}'_{im}) - k_{\mathbf{w}'}(\mathbf{x}_i,\mathbf{y}_i,\mathbf{y}'_{i1},\cdots,\mathbf{y}'_{im}))^2} \\
&\leq \max_{i}|k_{\mathbf{w}}(\mathbf{x}_i,\mathbf{y}_i,\mathbf{y}'_{i1},\cdots,\mathbf{y}'_{im}) - k_{\mathbf{w}'}(\mathbf{x}_i,\mathbf{y}_i,\mathbf{y}'_{i1},\cdots,\mathbf{y}'_{im})| \\
&\leq \max_{i}\max_{j}|\Delta_{\mathbf{w}}(\mathbf{x}_i,\mathbf{y}_i,\mathbf{y}'_{ij}) - \Delta_{\mathbf{w}'}(\mathbf{x}_i,\mathbf{y}_i,\mathbf{y}'_{ij})| \\
&\leq \max_{(\hat{\mathbf{x}},\hat{\mathbf{y}},\hat{\mathbf{y}}')\in S_1}|\Delta_{\mathbf{w}}(\hat{\mathbf{x}},\hat{\mathbf{y}},\hat{\mathbf{y}}') - \Delta_{\mathbf{w}'}(\hat{\mathbf{x}},\hat{\mathbf{y}},\hat{\mathbf{y}}')| = \|\Delta_{\mathbf{w}} - \Delta_{\mathbf{w}'}\|_{L_\infty(S_1)}.
\end{aligned}
$$

As a result, for any $\epsilon > 0$,
$$\mathcal{N}(\mathcal{K}, \epsilon, L_2(S)) \le \mathcal{N}(\mathcal{H}, \epsilon, L_\infty(S_1)).$$

From Assumption 4.1, $\Delta_{\mathbf{w}}(\mathbf{x}, \mathbf{y}, \mathbf{y}') \in [-2B, 2B]$ for all $\mathbf{x} \in \mathcal{X}, \mathbf{y}, \mathbf{y}' \in \mathcal{Y}$, then $k_{\mathbf{w}} \in [-2B, 2B]$, implying that
$$B_{\mathcal{K}} := \sup_{\mathbf{w} \in \mathcal{W}} \|k_{\mathbf{w}}\|_{L_2(S)} \le 2B.$$

According to Proposition A.3 and Lemma A.5, there holds

$$\mathfrak{R}_S(\mathcal{K}) \le \inf_{\alpha > 0} 4\alpha + \frac{12}{\sqrt{n}} \int_\alpha^{B_{\mathcal{K}}} \sqrt{\log \mathcal{N}(\mathcal{K}, \epsilon, L_2(S))} d\epsilon \le \frac{4}{n} + \frac{12}{\sqrt{n}} \int_{\frac{1}{n}}^{2B} \sqrt{\log \mathcal{N}(\mathcal{K}, \epsilon, L_2(S))} d\epsilon$$

$$\le \frac{4}{n} + \frac{12}{\sqrt{n}} \int_{\frac{1}{n}}^{2B} \sqrt{\log \mathcal{N}(\mathcal{H}, \epsilon, L_\infty(S_1))} d\epsilon \le \frac{4}{n} + \frac{12}{\sqrt{n}} \int_{\frac{1}{n}}^{2B} \sqrt{\sum_{l=1}^{L} d_l d_{l-1} \log\left(1 + \frac{8L\Pi_{l=1}^L s_l^2}{\epsilon}\right)} d\epsilon$$

$$\le \frac{4}{n} + \frac{12}{\sqrt{n}} \sqrt{\sum_{l=1}^{L} d_l d_{l-1} \log(1 + 8nL\Pi_{l=1}^L s_l^2)(2B - 1/n)} \lesssim \frac{B\sqrt{\log(1 + 8nL\Pi_{l=1}^L s_l^2)\sum_{l=1}^L d_l d_{l-1}}}{n}.$$

Using Proposition A.1, with probability at least $1 - \delta$, we have

$$\sup_{\mathbf{w} \in \mathcal{W}} \left[\mathbb{E}\widehat{\mathcal{L}}_S(\mathbf{w}) - \widehat{\mathcal{L}}_S(\mathbf{w})\right]$$

$$= \sup_{\mathbf{w} \in \mathcal{W}} \mathbb{E}_{\mathbf{x}, \mathbf{y}, \mathbf{y}'_1, \cdots, \mathbf{y}'_m} \left[k_{\mathbf{w}}(\mathbf{x}, \mathbf{y}, \mathbf{y}'_1, \cdots, \mathbf{y}'_m)\right] - \frac{1}{n} \sum_{i=1}^{n} k_{\mathbf{w}}(\mathbf{x}_i, \mathbf{y}_i, \mathbf{y}'_{i1}, \cdots, \mathbf{y}'_{im}).$$

$$\le 2\mathfrak{R}_S(\mathcal{K}) + 12B\sqrt{\frac{\log(2/\delta)}{2n}} \lesssim \frac{B\sqrt{\log(1/\delta)} + B\sqrt{\log(1 + 8nL\Pi_{l=1}^L s_l^2)\sum_{l=1}^L d_l d_{l-1}}}{\sqrt{n}}.$$

The inequality also holds for the other side following a similar manner. The proof is completed. $\qquad\square$

## D. Proof of Theorem 4.6

Recall that

$$|\mathcal{L}_{SS}(s_{\mathbf{w}}) - \widehat{\mathcal{L}}_{SS}(s_{\mathbf{w}})| \le \underbrace{\left|\mathbb{E}_{\mathbf{x},\mathbf{y}} h_{\mathbf{w}}(\mathbf{x}, \mathbf{y}) - \frac{1}{n} \sum_{i=1}^{n} h_{\mathbf{w}}(\mathbf{x}_i, \mathbf{y}_i)\right|}_{\text{outer error}} + \underbrace{\left|\frac{1}{n} \sum_{i=1}^{n} \left(h_{\mathbf{w}}(\mathbf{x}_i, \mathbf{y}_i) - \hat{h}_{\mathbf{w}}(\mathbf{x}_i, \mathbf{y}_i)\right)\right|}_{\text{inner error}},$$

where

$$h_{\mathbf{w}}(\mathbf{x}, \mathbf{y}) = \tau \log \mathbb{E}_{\mathbf{y}'} \exp\left(\frac{\Delta_{\mathbf{w}}(\mathbf{x}, \mathbf{y}, \mathbf{y}')}{\tau}\right), \quad \hat{h}_{\mathbf{w}}(\mathbf{x}, \mathbf{y}) = \tau \log\left(\frac{1}{m} \sum_{j=1}^{m} \exp\left(\frac{\Delta_{\mathbf{w}}(\mathbf{x}, \mathbf{y}, \mathbf{y}'_j)}{\tau}\right)\right).$$

We will control each component respectively.

### D.1. Estimation of the outer error

We define the following function class and data space:

$$\mathcal{H}_1 = \{(\mathbf{x}, \mathbf{y}) \mapsto h_{\mathbf{w}}(\mathbf{x}, \mathbf{y}), \mathbf{w} \in \mathcal{W}\}, \quad S_4 = \{(\mathbf{x}_1, \mathbf{y}_1), \cdots, (\mathbf{x}_n, \mathbf{y}_n)\}.$$

To bound the outer error, we control the covering number $\log \mathcal{N}(\mathcal{H}_1, \epsilon, L_2(S_4))$.

**Lemma D.1.** *Let Assumption 4.1 hold, then for any $\epsilon > 0$, we have*

$$\log \mathcal{N}(\mathcal{H}_1, \epsilon, L_2(S_4)) \le \sum_{l=1}^{L} d_l d_{l-1} \log\left(1 + \frac{\exp(4B/\tau) 8L\Pi_{l=1}^L s_l^2}{\epsilon}\right).$$

*Proof.* Let $h_{\mathbf{w}}, h_{\mathbf{w}'}$ be two functions in $\mathcal{H}_1$. For $\mathbf{x} \in \mathcal{X}, \mathbf{y} \in \mathcal{Y}$, we aim bound $|h_{\mathbf{w}}(\mathbf{x}, \mathbf{y}) - h_{\mathbf{w}'}(\mathbf{x}, \mathbf{y})|$. Without loss of generality, we assume $h_{\mathbf{w}}(\mathbf{x}, \mathbf{y}) \geq h_{\mathbf{w}'}(\mathbf{x}, \mathbf{y})$. Then

$$
\begin{aligned}
|h_{\mathbf{w}}(\mathbf{x}, \mathbf{y}) - h_{\mathbf{w}'}(\mathbf{x}, \mathbf{y})| &= h_{\mathbf{w}}(\mathbf{x}, \mathbf{y}) - h_{\mathbf{w}'}(\mathbf{x}, \mathbf{y}) \\
&= \tau \log \mathbb{E}_{\mathbf{y}'} \exp\left(\frac{\Delta_{\mathbf{w}}(\mathbf{x}, \mathbf{y}, \mathbf{y}')}{\tau}\right) - \tau \log \mathbb{E}_{\mathbf{y}'} \exp\left(\frac{\Delta_{\mathbf{w}'}(\mathbf{x}, \mathbf{y}, \mathbf{y}')}{\tau}\right) \\
&= \tau \log \frac{\mathbb{E}_{\mathbf{y}'} \exp(\Delta_{\mathbf{w}}(\mathbf{x}, \mathbf{y}, \mathbf{y}')/\tau)}{\mathbb{E}_{\mathbf{y}'} \exp(\Delta_{\mathbf{w}'}(\mathbf{x}, \mathbf{y}, \mathbf{y}')/\tau)} \\
&\leq \tau \frac{\mathbb{E}_{\mathbf{y}'} \exp(\Delta_{\mathbf{w}}(\mathbf{x}, \mathbf{y}, \mathbf{y}')/\tau) - \mathbb{E}_{\mathbf{y}'} \exp(\Delta_{\mathbf{w}'}(\mathbf{x}, \mathbf{y}, \mathbf{y}')/\tau)}{\mathbb{E}_{\mathbf{y}'} \exp(\Delta_{\mathbf{w}'}(\mathbf{x}, \mathbf{y}, \mathbf{y}')/\tau)} \\
&\leq \tau \frac{\mathbb{E}_{\mathbf{y}'} \exp(\Delta_{\mathbf{w}}(\mathbf{x}, \mathbf{y}, \mathbf{y}')/\tau) - \mathbb{E}_{\mathbf{y}'} \exp(\Delta_{\mathbf{w}'}(\mathbf{x}, \mathbf{y}, \mathbf{y}')/\tau)}{\exp(-2B/\tau)} \\
&\leq \tau \frac{\mathbb{E}_{\mathbf{y}'} |\exp(\Delta_{\mathbf{w}}(\mathbf{x}, \mathbf{y}, \mathbf{y}')/\tau) - \exp(\Delta_{\mathbf{w}'}(\mathbf{x}, \mathbf{y}, \mathbf{y}')/\tau)|}{\exp(-2B/\tau)} \\
&\leq \tau \frac{\mathbb{E}_{\mathbf{y}'} \exp(2B/\tau)|\Delta_{\mathbf{w}}(\mathbf{x}, \mathbf{y}, \mathbf{y}')/\tau - \Delta_{\mathbf{w}'}(\mathbf{x}, \mathbf{y}, \mathbf{y}')/\tau|}{\exp(-2B/\tau)} \\
&= \exp(4B/\tau) \mathbb{E}_{\mathbf{y}'} |\Delta_{\mathbf{w}}(\mathbf{x}, \mathbf{y}, \mathbf{y}') - \Delta_{\mathbf{w}'}(\mathbf{x}, \mathbf{y}, \mathbf{y}')|,
\end{aligned}
$$

where the first inequality is due to $\log(t) \leq t - 1$, the second inequality results from $\Delta_{\mathbf{w}'}(\mathbf{x}, \mathbf{y}, \mathbf{y}') = s_{\mathbf{w}'}(\mathbf{x}, \mathbf{y}') - s_{\mathbf{w}'}(\mathbf{x}, \mathbf{y}) \in [-2B, 2B]$, the last inequality is by the $\exp(2B/\tau)$-Lipschitzness of $\exp(t)$ over the interval $[-2B/\tau, 2B/\tau]$.

We choose a cover $\mathcal{C}_{\varepsilon}$ of $\mathcal{W}$ such that Eq. (17) holds. By Eq. (21), we know that

$$
\begin{aligned}
|h_{\mathbf{w}}(\mathbf{x}, \mathbf{y}) - h_{\mathbf{w}'}(\mathbf{x}, \mathbf{y})| &\leq \exp(4B/\tau) \mathbb{E}_{\mathbf{y}'} |\Delta_{\mathbf{w}}(\mathbf{x}, \mathbf{y}, \mathbf{y}') - \Delta_{\mathbf{w}'}(\mathbf{x}, \mathbf{y}, \mathbf{y}')| \\
&\leq 4 \exp(4B/\tau) \Pi_{l=1}^{L} s_l \mathbb{E}_{\mathbf{y}'} \max_{\mathbf{v} \in \{\mathbf{x}, \mathbf{y}, \mathbf{y}'\}} \|f_{\mathbf{w}}(\mathbf{v}) - f_{\mathbf{w}'}(\mathbf{v})\|_2 \\
&\leq 4 \exp(4B/\tau) \Pi_{l=1}^{L} s_l \max_{\mathbf{v} \in \mathcal{X}, \mathcal{Y}} \|f_{\mathbf{w}}(\mathbf{v}) - f_{\mathbf{w}'}(\mathbf{v})\|_2 \leq 4 \exp(4B/\tau) \Pi_{l=1}^{L} s_l \varepsilon.
\end{aligned}
$$

As a result,

$$
\|h_{\mathbf{w}} - h_{\mathbf{w}'}\|_{L_2(S_4)} \leq 4 \exp(4B/\tau) \Pi_{l=1}^{L} s_l \varepsilon
$$

For $\epsilon > 0$, choosing

$$
\varepsilon = \frac{\epsilon}{4 \exp(4B/\tau) \Pi_{l=1}^{L} s_l}.
$$

Then by Lemma A.4, we have

$$
\log \mathcal{N}(\mathcal{H}_1, \epsilon, L_2(S_4)) \leq \log |\mathcal{C}_{\varepsilon}| \leq \sum_{l=1}^{L} d_l d_{l-1} \log\left(1 + \frac{2L\Pi_{l=1}^{L} s_l}{\varepsilon}\right) = \sum_{l=1}^{L} d_l d_{l-1} \log\left(1 + \frac{8 \exp(4B/\tau) L \Pi_{l=1}^{L} s_l^2}{\epsilon}\right).
$$

The proof is completed. $\qquad\square$

The outer error can be bounded as below.

**Lemma D.2.** *Let Assumption 4.1 hold, then with probability at least $1 - \delta$, there holds*

$$
\left| \mathbb{E}_{\mathbf{x}, \mathbf{y}} h_{\mathbf{w}}(\mathbf{x}, \mathbf{y}) - \frac{1}{n} \sum_{i=1}^{n} h_{\mathbf{w}}(\mathbf{x}_i, \mathbf{y}_i) \right| \lesssim \frac{B\sqrt{\sum_{l=1}^{L} d_l d_{l-1} \log(1 + 8 \exp(4B/\tau) n L \Pi_{l=1}^{L} s_l^2)} + B\sqrt{\log(1/\delta)}}{\sqrt{n}}.
$$

*Proof.* Since $\Delta_{\mathbf{w}}(\mathbf{x}, \mathbf{y}, \mathbf{y}') = s_{\mathbf{w}}(\mathbf{x}, \mathbf{y}') - s_{\mathbf{w}}(\mathbf{x}, \mathbf{y}) \in [-2B, 2B]$, we have $h_{\mathbf{w}}(\mathbf{x}, \mathbf{y}) \in [-2B, 2B]$. Then by Proposi-

tion A.3 and Lemma D.1,

$$\mathfrak{R}_{S_4}(\mathcal{H}_1) \leq \inf_{\alpha>0} 4\alpha + \frac{12}{\sqrt{n}} \int_\alpha^{2B} \sqrt{\log \mathcal{N}(\mathcal{H}_1, \epsilon, L_2(S_4))} d\epsilon \leq \frac{4}{n} + \frac{12}{\sqrt{n}} \int_{1/n}^{2B} \sqrt{\log \mathcal{N}(\mathcal{H}_1, \epsilon, L_2(S_4))} d\epsilon$$

$$\leq \frac{4}{n} + \frac{12}{\sqrt{n}} \int_{1/n}^{2B} \sqrt{\sum_{l=1}^L d_l d_{l-1} \log\left(1 + \frac{8\exp(4B/\tau)L\Pi_{l=1}^L s_l^2}{\epsilon}\right)} d\epsilon$$

$$\leq \frac{4}{n} + \frac{12}{\sqrt{n}} \sqrt{\sum_{l=1}^L d_l d_{l-1} \log(1 + 8\exp(4B/\tau)nL\Pi_{l=1}^L s_l^2)(2B - 1/n)}$$

$$\lesssim \frac{B}{\sqrt{n}} \sqrt{\sum_{l=1}^L d_l d_{l-1} \log(1 + 8\exp(4B/\tau)nL\Pi_{l=1}^L s_l^2)}$$

Combined with Proposition A.1, we have with probability at least $1 - \delta$,

$$\mathbb{E}_{\mathbf{x},\mathbf{y}} h_{\mathbf{w}}(\mathbf{x}, \mathbf{y}) - \frac{1}{n}\sum_{i=1}^n h_{\mathbf{w}}(\mathbf{x}_i, \mathbf{y}_i) \leq 2\mathfrak{R}_{S_4}(\mathcal{H}_1) + 12B\sqrt{\frac{\log(1/\delta)}{n}} \tag{35}$$

$$\lesssim \frac{B\sqrt{\sum_{l=1}^L d_l d_{l-1} \log(1 + 8\exp(4B/\tau)nL\Pi_{l=1}^L s_l^2)} + B\sqrt{\log(1/\delta)}}{\sqrt{n}}. \tag{36}$$

Similar inequality holds for another direction. The proof is completed. $\qquad\square$

### D.2. Estimation of the inner error

For any $\mathbf{x} \in \mathcal{X}, \mathbf{y} \in \mathcal{Y}$, we define the following function class and dataset:

$$\mathcal{G}_{\mathbf{x},\mathbf{y}} := \{\mathbf{y}' \to \Delta_{\mathbf{w}}(\mathbf{x}, \mathbf{y}, \mathbf{y}'), \mathbf{w} \in \mathcal{W}\}, \quad S' = \{\mathbf{y}'_1, \cdots, \mathbf{y}'_m\}. \tag{37}$$

We bound the covering number $\mathcal{N}(\mathcal{G}_{\mathbf{x},\mathbf{y}}, \epsilon, L_2(S'))$ in the following lemma.

**Lemma D.3.** *Let Assumption 4.1 hold, then for any* $\mathbf{x}, \mathbf{y}, S' = \{\mathbf{y}'_1, \cdots \mathbf{y}'_m\}$, *there holds*

$$\log \mathcal{N}(\mathcal{G}_{\mathbf{x},\mathbf{y}}, \epsilon, L_2(S')) \leq \sum_{l=1}^L d_l d_{l-1} \log\left(1 + \frac{8L\Pi_{l=1}^L s_l^2}{\epsilon}\right).$$

*Proof.* We define $S_3 = \{\mathbf{x}, \mathbf{y}, \mathbf{y}'_1, \cdots, \mathbf{y}'_m\}$. Let $\Delta_{\mathbf{w}}(\mathbf{x}, \mathbf{y}, \cdot), \Delta_{\mathbf{w}'}(\mathbf{x}, \mathbf{y}, \cdot)$ be two functions in $\mathcal{G}_{\mathbf{x},\mathbf{y}}$. We choose a cover $\mathcal{C}_\varepsilon$ of $\mathcal{W}$ such that Eq. (17) holds. According to Eq. (21), for any $\mathbf{x} \in \mathcal{X}, \mathbf{y}, \mathbf{y}' \in \mathcal{Y}$, we have

$$|\Delta_{\mathbf{w}}(\mathbf{x}, \mathbf{y}, \mathbf{y}') - \Delta_{\mathbf{w}'}(\mathbf{x}, \mathbf{y}, \mathbf{y}')| \leq 4\Pi_{l=1}^L s_l \max_{\mathbf{v} \in \{\mathbf{x}, \mathbf{y}, \mathbf{y}'\}} \|f_{\mathbf{w}}(\mathbf{v}) - f_{\mathbf{w}'}(\mathbf{v})\|_2 \leq 4\Pi_{l=1}^L s_l \varepsilon.$$

It implies that

$$\|\Delta_{\mathbf{w}}(\mathbf{x}, \mathbf{y}, \cdot) - \Delta_{\mathbf{w}'}(\mathbf{x}, \mathbf{y}, \cdot)\|_{L_2(S')} = \sqrt{\frac{1}{m}\sum_{j=1}^m (\Delta_{\mathbf{w}}(\mathbf{x}, \mathbf{y}, \mathbf{y}'_j) - \Delta_{\mathbf{w}'}(\mathbf{x}, \mathbf{y}, \mathbf{y}'_j))^2}$$

$$\leq \sqrt{\frac{1}{m}\sum_{j=1}^m \left(4\Pi_{l=1}^L s_l \max_{\mathbf{v} \in S_3} \|f_{\mathbf{w}}(\mathbf{v}) - f_{\mathbf{w}'}(\mathbf{v})\|_2\right)^2}$$

$$= 4\Pi_{l=1}^L s_l \max_{\mathbf{v} \in S_3} \|f_{\mathbf{w}}(\mathbf{v}) - f_{\mathbf{w}'}(\mathbf{v})\|_2 \leq 4\Pi_{l=1}^L s_l \varepsilon.$$

Letting $\varepsilon = \epsilon/(\leq 4\Pi_{l=1}^L s_l)$ yields

$$\log \mathcal{N}(\mathcal{G}_{\mathbf{x},\mathbf{y}}, \epsilon, L_2(S')) \leq \log \mathcal{N}(\mathcal{F}, \epsilon/(4\Pi_{l=1}^L s_l), L_{\infty,2}(S_3)) \leq \log|C_\varepsilon| \leq \sum_{l=1}^L d_l d_{l-1} \log\left(1 + \frac{8L\Pi_{l=1}^L s_l^2}{\epsilon}\right).$$

The proof is completed. $\qquad\square$

The inner error can be estimated in the following lemma.

**Lemma D.4.** *Let Assumption 4.1 hold, then with probability at least $1 - \delta$, we have*

$$\sup_{\mathbf{w} \in \mathcal{W}} \left| \frac{1}{n} \sum_{i=1}^{n} (h_{\mathbf{w}}(\mathbf{x}_i, \mathbf{y}_i) - \hat{h}_{\mathbf{w}}(\mathbf{x}_i, \mathbf{y}_i)) \right| = \widetilde{O}\left( \frac{\exp(4B/\tau)(B\sqrt{\sum_{l=1}^{L} d_l d_{l-1}} + \tau\sqrt{\log(n/\delta)})}{\sqrt{m}} \right).$$

*Proof.* We define

$$\mathcal{I}_{\mathbf{x}, \mathbf{y}} := \left\{ \mathbf{y}' \to \tau \exp\left( \frac{\Delta_{\mathbf{w}}(\mathbf{x}, \mathbf{y}, \mathbf{y}') - \mu}{\tau} \right), \mathbf{w} \in \mathcal{W}, \mu \in [-2B, 2B] \right\}.$$

Then we have $\frac{\Delta_{\mathbf{w}}(\mathbf{x}, \mathbf{y}, \mathbf{y}') - \mu}{\tau} \in [-4B/\tau, 4B/\tau]$ and $\tau \exp(\cdot/\tau)$ is $\exp(4B/\tau)$-Lipschitz over this interval. For any $\mathbf{w} \in \mathcal{W}, \mu \in [-2B, 2B]$, there holds

$$0 \leq \tau \exp\left( \frac{\Delta_{\mathbf{w}}(\mathbf{x}, \mathbf{y}, \mathbf{y}') - \mu}{\tau} \right) \leq \tau \exp(4B/\tau).$$

Then By Eqs. (13), (14) and Proposition A.1, we have

$$\sup_{\mathbf{w} \in \mathcal{W}} |h_{\mathbf{w}}(\mathbf{x}, \mathbf{y}) - \hat{h}_{\mathbf{w}}(\mathbf{x}, \mathbf{y})|$$

$$= \sup_{\mathbf{w} \in \mathcal{W}} \tau \log \mathbb{E}_{\mathbf{y}'} \exp\left( \frac{\Delta_{\mathbf{w}}(\mathbf{x}, \mathbf{y}, \mathbf{y}')}{\tau} \right) - \tau \log\left( \frac{1}{m} \sum_{j=1}^{m} \exp\left( \frac{\Delta_{\mathbf{w}}(\mathbf{x}, \mathbf{y}, \mathbf{y}'_j)}{\tau} \right) \right)$$

$$= \sup_{\mathbf{w} \in \mathcal{W}} \left[ \min_{|\mu| \leq 2B} \tau \mathbb{E}_{\mathbf{y}'} \exp\left( \frac{\Delta_{\mathbf{w}}(\mathbf{x}, \mathbf{y}, \mathbf{y}') - \mu}{\tau} \right) - \tau + \mu \right] - \left[ \min_{|\mu| \leq 2B} \frac{\tau}{m} \sum_{j=1}^{m} \exp\left( \frac{\Delta(\mathbf{x}, \mathbf{y}, \mathbf{y}'_j) - \mu}{\tau} \right) - \tau + \mu \right]$$

$$\leq \sup_{\mathbf{w} \in \mathcal{W}, |\mu| \leq 2B} \tau \mathbb{E}_{\mathbf{y}'} \exp\left( \frac{\Delta_{\mathbf{w}}(\mathbf{x}, \mathbf{y}, \mathbf{y}') - \mu}{\tau} \right) - \frac{\tau}{m} \sum_{j=1}^{m} \exp\left( \frac{\Delta(\mathbf{x}, \mathbf{y}, \mathbf{y}'_j) - \mu}{\tau} \right)$$

$$\leq 2\mathfrak{R}_{S'}(\mathcal{I}_{\mathbf{x}, \mathbf{y}}) + 3\tau \exp(4B/\tau)\sqrt{\frac{\log(2/\delta)}{2n}}$$

with probability at least $1 - \delta/2$. We define

$$\mathcal{K}_{\mathbf{x}, \mathbf{y}} := \{\mathbf{y}' \to \Delta_{\mathbf{w}}(\mathbf{x}, \mathbf{y}, \mathbf{y}') - \mu, \mathbf{w} \in \mathcal{W}, \mu \in [-2B, 2B]\}.$$

According to Lemma A.2, we have

$$\sup_{\mathbf{w} \in \mathcal{W}} |h_{\mathbf{w}}(\mathbf{x}, \mathbf{y}) - \hat{h}_{\mathbf{w}}(\mathbf{x}, \mathbf{y})|$$

$$\leq 2\exp(4B/\tau)\mathfrak{R}_{S'}(\mathcal{K}_{\mathbf{x}, \mathbf{y}}) + 3\tau \exp(4B/\tau)\sqrt{\frac{\log(2/\delta)}{2n}}. \tag{38}$$

Note that

$$\mathfrak{R}_{S'}(\mathcal{K}_{\mathbf{x}, \mathbf{y}}) = \mathbb{E}_{\epsilon} \sup_{\mathbf{w} \in \mathcal{W}, \mu \in [-2B, 2B]} \frac{1}{m} \sum_{j=1}^{m} \epsilon_j (\Delta_{\mathbf{w}}(\mathbf{x}, \mathbf{y}, \mathbf{y}'_j) - \mu)$$

$$\leq \mathbb{E}_{\epsilon} \sup_{\mathbf{w} \in \mathcal{W}} \frac{1}{m} \sum_{j=1}^{m} \epsilon_j \Delta_{\mathbf{w}}(\mathbf{x}, \mathbf{y}, \mathbf{y}'_j) + \mathbb{E}_{\epsilon} \sup_{\mu \in [-2B, 2B]} \frac{1}{m} \sum_{j=1}^{m} \epsilon_j \mu.$$

We will control two terms respectively.

$$\mathbb{E}_{\epsilon} \sup_{\mu \in [-2B, 2B]} \frac{1}{m} \sum_{j=1}^{m} \epsilon_j \mu \leq 2B\mathbb{E}_{\epsilon} \left| \frac{1}{m} \sum_{j=1}^{m} \epsilon_j \right| \leq 2B\sqrt{\mathbb{E}_{\epsilon} \left| \frac{1}{m} \sum_{j=1}^{m} \epsilon_j \right|^2} = \frac{2B}{\sqrt{m}}.$$

We apply Proposition A.3 and Lemma D.3 to obtain

$$\mathbb{E}_\epsilon \sup_{\mathbf{w}} \frac{1}{m} \sum_{j=1}^m \epsilon_j \Delta_{\mathbf{w}}(\mathbf{x}, \mathbf{y}, \mathbf{y}'_j) \le \inf_{\alpha>0} 4\alpha + \frac{12}{\sqrt{m}} \int_\alpha^{2B} \sqrt{\log \mathcal{N}(\mathcal{G}_{\mathbf{x},\mathbf{y}}, \epsilon, L_2(S'))} d\epsilon \tag{39}$$

$$\le \frac{4}{m} + \frac{12}{\sqrt{m}} \int_{1/m}^{2B} \sqrt{\log \mathcal{N}(\mathcal{G}_{\mathbf{x},\mathbf{y}}, \epsilon, L_2(S'))} d\epsilon \tag{40}$$

$$\le \frac{4}{m} + \frac{12}{\sqrt{m}} \int_{1/m}^{2B} \sqrt{\sum_{l=1}^L d_l d_{l-1} \log\left(1 + \frac{8L\Pi_{l=1}^L s_l^2}{\epsilon}\right)} d\epsilon$$

$$\le \frac{4}{m} + \frac{12}{\sqrt{m}} \sqrt{\sum_{l=1}^L d_l d_{l-1} \log(1 + 8mL\Pi_{l=1}^L s_l^2)(2B - 1/m)}$$

$$\lesssim \frac{B}{\sqrt{m}} \sqrt{\sum_{l=1}^L d_l d_{l-1} \log(1 + 8mL\Pi_{l=1}^L s_l^2)}. \tag{41}$$

Therefore,

$$\mathfrak{R}_{S'}(\mathcal{K}_{\mathbf{x},\mathbf{y}}) \lesssim \frac{B}{\sqrt{m}} \sqrt{\sum_{l=1}^L d_l d_{l-1} \log(1 + 8mL\Pi_{l=1}^L s_l^2)}.$$

Plugging it into Eq. (38) yields

$$\sup_{\mathbf{w} \in \mathcal{W}} h_{\mathbf{w}}(\mathbf{x}, \mathbf{y}) - \hat{h}_{\mathbf{w}}(\mathbf{x}, \mathbf{y}) \lesssim \frac{\exp(4B/\tau)(B\sqrt{\sum_{l=1}^L \log(1 + 8mL\Pi_{l=1}^L s_l^2)d_l d_{l-1}} + \tau\sqrt{\log(1/\delta)})}{\sqrt{m}} \tag{42}$$

with probability at least $1 - \delta$. By the independence of $S'$ and $\mathbf{x}_i, \mathbf{y}_i$, with probability at least $1 - \delta/n$, the following holds for all $i \in [n]$:

$$\sup_{\mathbf{w} \in \mathcal{W}} h_{\mathbf{w}}(\mathbf{x}_i, \mathbf{y}_i) - \hat{h}_{\mathbf{w}}(\mathbf{x}_i, \mathbf{y}_i) \lesssim \frac{\exp(4B/\tau)(B\sqrt{\sum_{l=1}^L \log(1 + 8mL\Pi_{l=1}^L s_l^2)d_l d_{l-1}} + \tau\sqrt{\log(1/\delta)})}{\sqrt{m}}.$$

Taking summation yields

$$\sup_{\mathbf{w} \in \mathcal{W}} \frac{1}{n} \sum_{i=1}^n (h_{\mathbf{w}}(\mathbf{x}_i, \mathbf{y}_i) - \hat{h}_{\mathbf{w}}(\mathbf{x}_i, \mathbf{y}_i)) \lesssim \frac{\exp(4B/\tau)(B\sqrt{\sum_{l=1}^L \log(1 + 8mL\Pi_{l=1}^L s_l^2)d_l d_{l-1}} + \tau\sqrt{\log(1/\delta)})}{\sqrt{m}}.$$

The other side of the inequality holds for the same reason. The proof of the lemma is completed. $\qquad\square$

Now we prove Theorem 4.6.

*Proof of Theorem 4.6.* Plugging Lemma 4.4, D.4 into the error decomposition Eq. (16), with probability at least $1 - \delta$, there holds

$$\sup_{\mathbf{w} \in \mathcal{W}} |\mathcal{L}_{\text{SS}}(s_{\mathbf{w}}) - \widehat{\mathcal{L}}_{\text{SS}}(s_{\mathbf{w}})| \le \left|\mathbb{E}_{\mathbf{x},\mathbf{y}} h_{\mathbf{w}}(\mathbf{x}, \mathbf{y}) - \frac{1}{n} \sum_{i=1}^n h_{\mathbf{w}}(\mathbf{x}_i, \mathbf{y}_i)\right| + \left|\frac{1}{n} \sum_{i=1}^n (h_{\mathbf{w}}(\mathbf{x}_i, \mathbf{y}_i) - \hat{h}_{\mathbf{w}}(\mathbf{x}_i, \mathbf{y}_i))\right|$$

$$\lesssim \frac{B\sqrt{\sum_{l=1}^L d_l d_{l-1} \log(1 + 8\exp(4B/\tau)nL\Pi_{l=1}^L s_l^2)} + B\sqrt{\log(1/\delta)}}{\sqrt{n}}$$

$$+ \frac{\exp(4B/\tau)(B\sqrt{\sum_{l=1}^L \log(1 + 8mL\Pi_{l=1}^L s_l^2)d_l d_{l-1}} + \tau\sqrt{\log(1/\delta)})}{\sqrt{m}}.$$

The proof is completed. $\qquad\square$

**D.3. Zero-shot Classification in SSCRL**

In this subsection, we study the generalization performance of SSCRL in zero-shot classification task. We follow the framework in Oko et al. (2025). Given a scoring function $s$, the goal is to predict the label $c$ of $\mathbf{x}$ without training on a task-specific dataset. Suppose there are $C$ classes. For each class $c$, the distributions of data on two modalities $\mathcal{X}, \mathcal{Y}$ are denoted by $\mathcal{D}_{\mathcal{X}}(c), \mathcal{D}_{\mathcal{Y}}(c)$ respectively. we study zero-shot classification performance in a probabilistic modeling regime. Specifically, for a label $c$, we model the Bayes rule $\mathbb{P}(c|\mathbf{x})$ by

$$\mathbb{P}_s(c|\mathbf{x}) = \frac{\mathbb{E}_{\mathbf{y}\sim\mathcal{D}_{\mathcal{Y}}(c)}p_{\mathcal{Y}}(\mathbf{y})\exp(s(\mathbf{x},\mathbf{y})/\tau)}{\mathbb{E}_{\mathbf{y}}\exp(s(\mathbf{x},\mathbf{y})/\tau)}. \tag{43}$$

The performance of zero-shot classification is measured by KL-divergence between the Bayes rule and the modeled probability:

$$\mathcal{R}(s) = \tau\mathbb{E}_{\mathbf{x}}D_{\mathrm{KL}}(\mathbb{P}(c|\mathbf{x})||\mathbb{P}_s(c|\mathbf{x})).$$

Following Oko et al. (2025), we assume a conditional independence property of the data distribution.

**Assumption D.5.** For the joint distribution $(\mathbf{x}, \mathbf{y}, c)$, $\mathbf{x}$ and $c$ are conditionally independent given $\mathbf{y}$. An special case is when $c$ is a deterministic function of $\mathbf{y}$.

The following proposition shows that zero-shot classification error can be bounded by the pretraining excess risk.

**Proposition D.6** (Proposition 2 in Oko et al. (2025)). *Let Assumption D.5 hold, then the error for zero-shot classification can be bounded by the excess risk of CRL.*

$$\mathcal{R}(s) \le \mathcal{L}_{\mathrm{SS}}(s) - \inf_s \mathcal{L}_{\mathrm{SS}}(s).$$

We can adapt our generalization bound (Theorem 4.6) and obtain

**Theorem D.7.** *Let Assumptions 4.1, D.5 hold. Let $\hat{\mathbf{w}}_1 \in \arg\min_{\mathbf{w}\in\mathcal{W}}\widehat{\mathcal{L}}_{\mathrm{SS}}(\mathbf{w})$. Then with probability $1 - \delta$, there holds*

$$\mathcal{R}(s_{\hat{\mathbf{w}}_1}) \le \inf_{\mathbf{w}\in\mathcal{W}}\mathcal{L}_{\mathrm{SS}}(s_{\mathbf{w}}) - \inf_s \mathcal{L}_{\mathrm{SS}}(s) + \widetilde{O}\left(\frac{1}{\sqrt{n}} + \frac{1}{\sqrt{m}}\right).$$

# E. Extension to a general class of contrastive losses

In Lemma 4.2, we formulate contrastive loss as minimization problems, which forms the theoretical foundation of our generalization analysis. In fact, it is closely related to the so-called *optimized certainty equivalent* (OCE). Another observation is that CRL inherently relies on a *pairwise* structure, comparing positive and negative sample pairs via $\Delta_{\mathbf{w}}(\mathbf{x}, \mathbf{y}, \mathbf{y}')$. In this section, we unify the OCE framework and general pairwise loss functions to formalize a broad family of CRL losses, where the canonical contrastive loss in Eq. (1) emerges as a natural special case. Building on this general formulation, we extend our earlier analyses of statistical consistency and generalization bounds to this larger class of losses—thereby enhancing the generality and applicability of our theoretical findings beyond the standard CRL setting.

**Optimized certainty equivalent**  We first introduce the OCE framework (Ben-Tal & Teboulle, 2007), a powerful tool for risk-sensitive analysis.

**Definition E.1** (OCE Risk). Let $\phi : \mathbb{R} \to \mathbb{R} \cup \{+\infty\}$ be a nondecreasing, closed, convex disutility function satisfying $\phi(0) = 0$ and $1 \in \partial\phi(0)$. Suppose $\tau > 0$ is a hyperparameter. For a random variable $\mathbf{z}$ and a measurable function $f$, the corresponding OCE risk is defined as

$$\mathrm{oce}^{\phi}(f) = \min_{\mu\in\mathbb{R}}\left\{\tau\mathbb{E}_{\mathbf{z}}\left[\phi\left(\frac{f(\mathbf{z}) - \mu}{\tau}\right)\right] + \mu\right\}.$$

This formulation encodes a risk-averse perspective by penalizing large realizations of the loss $f(\mathbf{z})$. Given $m$ i.i.d. samples $\mathbf{z}_1, \ldots, \mathbf{z}_m$, the empirical OCE risk is

$$\mathrm{oce}^{\phi}_m(f) = \min_{\mu\in\mathbb{R}}\left\{\frac{\tau}{m}\sum_{j=1}^{m}\phi\left(\frac{f(\mathbf{z}_j) - \mu}{\tau}\right) + \mu\right\}.$$

*Table 2.* **Examples of OCE risk**. For $\alpha \in (0,1)$, $\mathrm{VaR}_\alpha(f(\mathbf{z}))$ denotes the $\alpha$-quantile of the distribution of $f(\mathbf{z})$, $[t]_+ = \max\{0, t\}$.

| Name | $\phi(t)$ | $\mathrm{oce}^\phi(f)$ |
|------|-----------|------------------------|
| Expected loss | $t$ | $\mathbb{E}[f(\mathbf{z})]$ |
| Entropy risk | $\exp(t) - 1$ | $\log \mathbb{E}[\exp(f(\mathbf{z}))]$ |
| Mean-variance | $\frac{1}{2}t^2 + t(t \geq -1)$ | $\mathbb{E}[f(\mathbf{z})] + \frac{1}{2}\mathrm{Var}(f(\mathbf{z}))$ |
| Conditional Value-at-Risk (CVaR) | $\frac{1}{\alpha}[t]_+$ | $\mathbb{E}[f(\mathbf{z}) \mid f(\mathbf{z}) \geq \mathrm{VaR}_\alpha(f(\mathbf{z}))]$ |

The OCE framework subsumes a broad class of well-known risk measures as special cases, with illustrative examples provided in Table 2 (Lee et al., 2020). Notably, Lemma 4.2 establishes that the canonical contrastive loss corresponds to the OCE with the exponential disutility function $\phi(t) = \exp(t) - 1$, meaning standard contrastive loss is itself a specific case of the more general OCE risk.

A key advantage of framing CRL through the OCE lens lies in its ability to explain the hard negative mining property of contrastive loss (Qiu et al., 2023). In fact, a key property of the OCE risk is its inherent risk aversion: the property $\phi(x) \geq x$ implies $\mathrm{oce}^\phi(f) \geq \mathbb{E}[f(\mathbf{z})]$, making it a stronger criterion than the standard expected risk. This risk-averse nature compels CRL to prioritize hard negative examples, i.e., those with large $\Delta_\mathbf{w}$ that are more challenging to distinguish.

**Pairwise loss**   Next, we note the pairwise nature of contrastive loss function. The goal of CRL is to discriminate positive data from negative data, i.e. increasing the score difference $\Delta_\mathbf{w}(\mathbf{x}, \mathbf{y}, \mathbf{y}')$ between positive pairs and negative pairs. This aligns naturally with the pairwise learning paradigm, which employs pairwise losses for discriminative tasks (Gao & Zhou, 2014), motivating us to adopt a general pairwise loss formulation. Specifically, we introduce a general pairwise loss: $\ell(\Delta_\mathbf{w}(\mathbf{x}, \mathbf{y}, \mathbf{y}'))$, where $\ell : \mathbb{R} \to \mathbb{R}$ is a nondecreasing function. Representative instantiations of $\ell$ include $\ell(t) = \exp(t)$, $\ell(t) = \log(\exp(t) + 1)$, and $\ell(t) = (\max(0, 1 + t))^2$.

By integrating the OCE framework with general pairwise loss, we propose the following contrastive loss for a pairwise loss $\ell$ and a general disutility function $\phi$:

$$\mathcal{L}^{\phi,\ell}(s_\mathbf{w}) = \mathbb{E}_\mathbf{x} \mathbb{E}_{\mathbf{y} \sim p_\mathbf{x}^+} \left[ \min_{\mu \in \mathbb{R}} \tau \mathbb{E}_{\mathbf{y}' \sim p_\mathbf{x}^-} \phi\left( \frac{\ell(\Delta_\mathbf{w}(\mathbf{x}, \mathbf{y}, \mathbf{y}')) - \mu}{\tau} \right) + \mu \right]. \tag{44}$$

Notably, the contrastive loss Eq. (1) is a special case of our general contrastive loss framework when choosing $\ell(t) = t$, $\phi(t) = \exp(t) - 1$.

Most existing work considers pairwise loss of the form

$$T(s_\mathbf{w}) = \mathbb{E}_\mathbf{x} \mathbb{E}_{\mathbf{y} \sim p_\mathbf{x}^+} \mathbb{E}_{\mathbf{y}' \sim p_\mathbf{x}^-} \left[ \ell(\Delta_\mathbf{w}(\mathbf{x}, \mathbf{y}, \mathbf{y}')) \right].$$

However, this standard expected risk has a critical limitation in practical CRL scenarios: the amount of negative examples is much larger than that of positive examples, and it fails to prioritize hard negative examples. However, the OCE-augmented pairwise loss $\mathcal{L}^{\phi,\ell}(s_\mathbf{w})$ inherits the advantages of both components: it retains the pairwise nature of CRL loss while leveraging the risk-averse property of OCE to focus more on hard negative examples. Compared to the standard expected risk $T(s_\mathbf{w})$, $\mathcal{L}^{\phi,\ell}(s_\mathbf{w})$ is more robust in imbalanced sample scenarios. Formally, we have the inequality $\mathcal{L}^{\phi,\ell}(s_\mathbf{w}) \geq T(s_\mathbf{w})$, which implies that $\mathcal{L}^{\phi,\ell}(s_\mathbf{w})$ is a stronger risk measure. This strength entails a one-way implication: minimizing $\mathcal{L}^{\phi,\ell}(s_\mathbf{w})$ guarantees a small $T(s_\mathbf{w})$, but a small $T(s_\mathbf{w})$ does not necessarily imply a small $\mathcal{L}^{\phi,\ell}(s_\mathbf{w})$.

In the rest of the paper, we assume $\phi$ satisfies Definition E.1.

### E.1. Problem formulations

Now we extend the settings in Section 2 to $\mathcal{L}^{\phi,\ell}(s_\mathbf{w})$ and define the statistical consistency.

E.1.1. SUPERVISED SETTING

Similar to Lemma 4.2, the population risk for SCRL with $\phi, \ell$ is given by

$$\mathcal{L}_{\mathrm{S}}^{\phi,\ell}(s_{\mathbf{w}}) = \mathbb{E}_{\mathbf{x}}\mathbb{E}_{\mathbf{y}\sim p_+(\cdot|\mathbf{x})}\left[\min_{\mu\in\mathbb{R}}\tau\mathbb{E}_{\mathbf{y}'\sim p_-(\cdot|\mathbf{x})}\phi\left(\frac{\ell(\Delta_{\mathbf{w}}(\mathbf{x},\mathbf{y},\mathbf{y}'))-\mu}{\tau}\right)+\mu\right]. \tag{45}$$

To estimate it, we draw $n$ i.i.d. anchor points $\mathbf{x}_i \sim p_{\mathcal{X}}$ for $i \in [n]$. For each anchor $\mathbf{x}_i$, we sample one positive example $\mathbf{y}_i \sim p_+(\cdot|\mathbf{x}_i)$ and then $m$ negative examples $\{\mathbf{y}'_{ij} \sim p_-(\cdot|\mathbf{x}_i) : j \in [m]\}$. The empirical risk is defined by

$$\widehat{\mathcal{L}}_{\mathrm{S}}^{\phi,\ell}(s_{\mathbf{w}}) = \frac{1}{n}\sum_{i=1}^n\left[\min_{\mu_i\in\mathbb{R}}\frac{\tau}{m}\sum_{j=1}^m\phi\left(\frac{\ell(\Delta_{\mathbf{w}}(\mathbf{x}_i,\mathbf{y}_i,\mathbf{y}'_{ij}))-\mu_i}{\tau}\right)+\mu_i\right]. \tag{46}$$

An interesting property of OCE is its relationship with distributional robust optimization (DRO) (Ben-Tal & Teboulle, 2007). Let $\mathbf{p} = (p_1,\cdots,p_m) \in \Delta_m$, where $\Delta_m$ is the simplex satisfying $\Delta_m = \{\mathbf{p} \in \mathbb{R}^m : \sum_{j=1}^m p_j = 1, p_j \geq 0\}$. Let $\varphi$ be a proper closed convex function and has a minimum value zero that is attained at $t = 1$. The $\varphi$-divergence is defined as $D_\varphi(\mathbf{p},\mathbf{q}) = \sum_{j=1}^m q_j\varphi(p_j/q_j)$. Then we have

**Proposition E.2.** *Let $\varphi^*(s) = \max_{t\geq 0} ts - \varphi(t)$ be the convex conjugate of $\varphi$. For $\phi = \varphi^*$, we have*

$$\widehat{\mathcal{L}}_{\mathrm{S}}^{\phi,\ell}(s_{\mathbf{w}}) = \frac{1}{n}\sum_{i=1}^n\left[\max_{\mathbf{p}_i\in\Delta_m}\sum_{j=1}^m p_{ij}\ell(\Delta_{\mathbf{w}}(\mathbf{x}_i,\mathbf{y}_i,\mathbf{y}'_{ij})) - \tau D_\varphi\left(\mathbf{p}_i,\frac{1}{m}\right)\right]. \tag{47}$$

The DRO objective considers a distribution shift over an uncertainty set, thus helping the model to be more robust. The term $\tau D_\varphi\left(\mathbf{p}_i,\frac{1}{m}\right)$ controls the deviation from the uniform distribution $\frac{1}{m}$. When $\varphi(t) = t\log t - t + 1$, $\varphi$-divergence corresponds to KL divergence. In this case, we can formulate Eq. (6) as DRO.

$$\widehat{\mathcal{L}}_{\mathrm{S}}(s_{\mathbf{w}}) = \frac{1}{n}\sum_{i=1}^n\left[\max_{\mathbf{p}_i\in\Delta_m}\sum_{j=1}^m p_{ij}(\Delta_{\mathbf{w}}(\mathbf{x}_i,\mathbf{y}_i,\mathbf{y}'_{ij})) - \tau D_{\mathrm{KL}}\left(\mathbf{p}_i,\frac{1}{m}\right)\right]. \tag{48}$$

*Proof of Proposition E.2.* We first prove that for any $z_1,\cdots,z_m$, there holds

$$\max_{\mathbf{p}\in\triangle_m}\sum_{j=1}^m p_j z_j - \tau D_\varphi\left(\mathbf{p},\frac{1}{m}\right) = \min_{\mu\in\mathbb{R}}\mu + \frac{\tau}{m}\sum_{j=1}^m\varphi^*((z_j-\mu)/\tau). \tag{49}$$

By Lagrangian duality theory, we have

$$\max_{\mathbf{p}\in\triangle_m}\sum_{j=1}^m p_j z_j - \tau D_\varphi\left(\mathbf{p},\frac{1}{m}\right) = \min_{\mu\in\mathbb{R}}\max_{p_j\geq 0}\mu(1-\sum_{j=1}^m p_j) + \sum_{j=1}^m p_j z_j - \tau D_\varphi\left(\mathbf{p},\frac{1}{m}\right)$$

$$= \min_{\mu\in\mathbb{R}}\max_{p_j\geq 0}\mu(1-\sum_{j=1}^m p_j) + \sum_{j=1}^m p_j z_j - \frac{\tau}{m}\sum_{j=1}^m\varphi(mp_j)$$

$$= \min_{\mu\in\mathbb{R}}\left\{\mu + \max_{p_j\geq 0}\frac{\tau}{m}\sum_{j=1}^m(mp_j(z_j-\mu)/\tau - \varphi(mp_j))\right\}$$

$$= \min_{\mu\in\mathbb{R}}\left\{\mu + \frac{\tau}{m}\sum_{j=1}^m\max_{p_j\geq 0}(mp_j(z_j-\mu)/\tau - \varphi(mp_j))\right\}$$

$$= \min_{\mu\in\mathbb{R}}\mu + \frac{\tau}{m}\sum_{j=1}^m\varphi^*((z_j-\mu)/\tau),$$

where the last equality is by the definition of $\varphi^*$. Hence, Eq. (49) holds true. Back to Eq. (47), for $i \in [n]$, let $z_j = \ell(\Delta_{\mathbf{w}}(\mathbf{x}_i, \mathbf{y}_i, \mathbf{y}'_{ij}))$, applying Eq. (49) implies that

$$\max_{\mathbf{p}_i \in \Delta_m} \sum_{j=1}^{m} p_{ij} \ell(\Delta_{\mathbf{w}}(\mathbf{x}_i, \mathbf{y}_i, \mathbf{y}'_{ij})) - \tau D_{\varphi}\left(\mathbf{p}_i, \frac{\mathbf{1}}{m}\right) = \min_{\mu_i \in \mathbb{R}} \frac{\tau}{m} \sum_{j=1}^{m} \varphi^*\left(\frac{\ell(\Delta_{\mathbf{w}}(\mathbf{x}_i, \mathbf{y}_i, \mathbf{y}'_{ij})) - \mu_i}{\tau}\right) + \mu_i.$$

Taking summation over $i$ completes the proof. $\qquad\square$

The above proposition also helps to connect OCE with other fields, for instance, the general scoring rule analyzed in Ryu et al. (2025).

### E.1.2. SELF-SUPERVISED SETTING

Given an anchor $\mathbf{x}$, we define the positive and negative conditional distributions as

$$p_{\mathbf{x}}^+(\mathbf{y}) = p(\mathbf{y}|\mathbf{x}), p_{\mathbf{x}}^-(\mathbf{y}) = p_{\mathcal{Y}}(\mathbf{y}). \tag{50}$$

We define the population risk as

$$\mathcal{L}_{\mathrm{SS}}^{\phi,\ell}(s_{\mathbf{w}}) = \mathbb{E}_{\mathbf{x}}\mathbb{E}_{\mathbf{y}|\mathbf{x}}\left[\min_{\mu \in \mathbb{R}} \tau \mathbb{E}_{\mathbf{y}'} \phi\left(\frac{\ell(\Delta_{\mathbf{w}}(\mathbf{x}, \mathbf{y}, \mathbf{y}')) - \mu}{\tau}\right) + \mu\right]. \tag{51}$$

To estimate it from data, we draw $n$ positive pairs $\{(\mathbf{x}_i, \mathbf{y}_i)\}_{i=1}^{n} \sim p$. Additionally, we sample $m$ negative examples $\{\mathbf{y}'_j\}_{j=1}^{m} \overset{\text{i.i.d.}}{\sim} p_{\mathcal{Y}}$. Each anchor $\mathbf{x}_i$ is then contrasted against with all $\mathbf{y}'_j, j \in [m]$. The empirical risk is defined to be

$$\widehat{\mathcal{L}}_{\mathrm{SS}}^{\phi,\ell}(s_{\mathbf{w}}) = \frac{1}{n} \sum_{i=1}^{n} \left[\min_{\mu_i} \frac{\tau}{m} \sum_{j=1}^{m} \phi\left(\frac{\ell(\Delta_{\mathbf{w}}(\mathbf{x}_i, \mathbf{y}_i, \mathbf{y}'_j)) - \mu_i}{\tau}\right) + \mu_i\right]. \tag{52}$$

### E.1.3. STATISTICAL CONSISTENCY

We will also study the statistical consistency of $\mathcal{L}^{\phi,\ell}$. We denote $\mathcal{L}_{*}^{\phi,\ell} = \inf_s \mathcal{L}^{\phi,\ell}(s)$, where the infimum is taken over all measurable scoring functions $s : \mathcal{X} \times \mathcal{Y} \to \mathbb{R}$. Recall that $\mathcal{E}^* = \sup_s \mathcal{E}(s)$ over all measurable scoring functions $s : \mathcal{X} \times \mathcal{Y} \to \mathbb{R}$.

**Definition E.3.** The contrastive loss $\mathcal{L}^{\phi,\ell}(s)$ is said to be statistically consistent with respect to $\mathcal{E}(s)$ if, for any sequence of scoring functions $\{s_n\}$, the following holds over all joint distributions on $\mathcal{X} \times \mathcal{Y}$.

$$\text{If } \mathcal{L}^{\phi,\ell}(s_n) \to \mathcal{L}_{*}^{\phi,\ell} \quad \text{then} \quad \mathcal{E}(s_n) \to \mathcal{E}^*.$$

## E.2. Useful lemmas about OCE

Now we introduce an interesting property of the empirical OCE w.r.t. $\ell_\infty$-norm.

**Lemma E.4.** *For any* $\mathbf{u} = (u_1, \cdots, u_m), \bar{\mathbf{u}} = (\bar{u}_1, \cdots, \bar{u}_m)$, *we define*

$$q(\mathbf{u}) = \min_{\mu \in \mathbb{R}} \frac{\tau}{m} \sum_{j=1}^{m} \phi\left(\frac{u_j - \mu}{\tau}\right) + \mu, \quad q(\bar{\mathbf{u}}) = \min_{\mu \in \mathbb{R}} \frac{1}{m} \sum_{j=1}^{m} \phi\left(\frac{\bar{u}_j - \mu}{\tau}\right) + \mu$$

*then there holds*

$$|q(\mathbf{u}) - q(\bar{\mathbf{u}})| \le \|\mathbf{u} - \bar{\mathbf{u}}\|_\infty = \max_j |u_j - \bar{u}_j|.$$

As a special case, we take $\phi(t) = \exp(t) - 1$ and derive that $\tau \log[\sum_{j=1}^{m} \exp(u_j/\tau)/m]$ is $1$-$\ell_\infty$-Lipschitz. This property plays an important role in controlling the covering numbers.

*Proof of Lemma E.4.* We assume that

$$\mu_1 \in \arg\min_{\mu \in \mathbb{R}} \frac{\tau}{m} \sum_{j=1}^{j} \phi\left(\frac{u_j - \mu}{\tau}\right) + \mu, \mu_2 \in \arg\min_{\mu \in \mathbb{R}} \frac{\tau}{m} \sum_{j=1}^{m} \phi\left(\frac{\bar{u}_j - \mu}{\tau}\right) + \mu.$$

By the optimality, we have

$$0 \in -\frac{\tau}{m} \sum_{j=1}^{m} \partial\phi\left(\frac{u_j - \mu}{\tau}\right) + 1, 0 \in -\frac{1}{m} \sum_{j=1}^{m} \partial\phi\left(\frac{\bar{u}_j - \mu}{\tau}\right) + 1. \tag{53}$$

Without loss of generality, we suppose $q(\mathbf{u}) \geq q(\bar{\mathbf{u}})$, then there holds

$$|q(\mathbf{u}) - q(\bar{\mathbf{u}})| = q(\mathbf{u}) - q(\bar{\mathbf{u}})$$

$$= \frac{\tau}{m} \sum_{j=1}^{m} \phi\left(\frac{u_j - \mu_1}{\tau}\right) + \mu_1 - \frac{\tau}{m} \sum_{j=1}^{m} \phi\left(\frac{\bar{u}_j - \mu_2}{\tau}\right) - \mu_2$$

$$= \frac{\tau}{m} \sum_{j=1}^{m} \left(\phi\left(\frac{u_j - \mu_1}{\tau}\right) - \phi\left(\frac{\bar{u}_j - \mu_2}{\tau}\right)\right) + \mu_1 - \mu_2$$

$$\leq \frac{\tau}{m} \sum_{j=1}^{m} \partial\phi\left(\frac{u_j - \mu_1}{\tau}\right)(u_j - \mu_1 - \bar{u}_j + \mu_2) + \mu_1 - \mu_2$$

$$= \frac{\tau}{m} \sum_{j=1}^{m} \partial\phi\left(\frac{u_j - \mu_1}{\tau}\right)(u_j - \bar{u}_j) + (\mu_1 - \mu_2)\left(1 - \frac{\tau}{m} \sum_{j=1}^{m} \partial\phi\left(\frac{u_j - \mu_1}{\tau}\right)\right),$$

where the inequality is due to the convexity of $\phi$. Since $\phi$ is nondecreasing, we have $\partial\phi(t) \geq 0$. Combining Eq. (53) yields

$$|q(\mathbf{u}) - q(\bar{\mathbf{u}})| = q(\mathbf{u}) - q(\bar{\mathbf{u}}) \leq \frac{\tau}{m} \sum_{j=1}^{m} \partial\phi\left(\frac{u_j - \mu_1}{\tau}\right)|u_j - \bar{u}_j|$$

$$\leq \left(\frac{\tau}{m} \sum_{j=1}^{m} \partial\phi\left(\frac{u_j - \mu_1}{\tau}\right)\right) \max_j |u_j - \bar{u}_j| = \max_j |u_j - \bar{u}_j| = \|\mathbf{u} - \bar{\mathbf{u}}\|_\infty.$$

The proof is completed. □

The following lemma indicates that we can constrain the parameter $\mu$ of OCE in a bounded domain.

**Lemma E.5.** *Suppose the function $f$ satisfies $f(\mathbf{z}) \in [a, b]$ for all $\mathbf{z}$. Then for any $\tau$, there holds*

$$\min_{\mu \in \mathbb{R}} \tau\mathbb{E}_{\mathbf{z}}\phi((f(\mathbf{z}) - \mu)/\tau) + \mu = \min_{\mu \in [a,b]} \tau\mathbb{E}_{\mathbf{z}}\phi((f(\mathbf{z}) - \mu)/\tau) + \mu,$$

$$\min_{\mu \in \mathbb{R}} \frac{1}{m} \sum_{j=1}^{m} \phi((f(\mathbf{z}_i) - \mu)/\tau) + \mu = \min_{\mu \in [a,b]} \frac{1}{m} \sum_{j=1}^{m} \phi((f(\mathbf{z}_i) - \mu)/\tau) + \mu.$$

*Proof.* We define

$$\lambda(\mu) = \tau\mathbb{E}_{\mathbf{z}}\phi((f(\mathbf{z}) - \mu)/\tau) + \mu.$$

To prove the first inequality, it suffices to prove that $\lambda(b + \epsilon) \geq \lambda(b), \lambda(a - \epsilon) \geq \lambda(a)$ for any $\epsilon > 0$.

To show that $\lambda(b + \epsilon) \geq \lambda(b)$ for $\epsilon > 0$. We define $X = f(\mathbf{z}) - b \in [a - b, 0]$. Then we have

$$\lambda(b + \epsilon) = \tau\mathbb{E}_{\mathbf{z}}\phi((f(\mathbf{z}) - b - \epsilon)/\tau) + b + \epsilon = \tau\mathbb{E}_{\mathbf{z}}\phi((X - \epsilon)/\tau) + b + \epsilon$$

$$\geq \tau\mathbb{E}_{\mathbf{z}}(\phi(X/\tau) - \epsilon/\tau) + b + \epsilon = \tau\mathbb{E}_{\mathbf{z}}\phi(X/\tau) + b = \lambda(b),$$

where the inequality is because $\phi$ is convex having 1 as a sub gradient at 0.

To show that $\lambda(a - \epsilon) \geq \lambda(a)$ for any $\epsilon > 0$. We define $Y = f(\mathbf{z}) - a \in [0, b - a]$. Then we have

$$\lambda(a - \epsilon) = \tau \mathbb{E}_{\mathbf{z}} \phi((f(\mathbf{z}) - a + \epsilon)/\tau) + a - \epsilon = \tau \mathbb{E}_{\mathbf{z}} \phi((Y + \epsilon)/\tau) + a - \epsilon$$
$$\geq \tau \mathbb{E}_{\mathbf{z}}(\phi(Y/\tau) + \epsilon/\tau) + a - \epsilon = \tau \mathbb{E}_{\mathbf{z}} \phi(Y/\tau) + a = \lambda(a),$$

where the inequality is because $\phi$ is convex having 1 as a sub gradient at 0. Therefore, the first part in the lemma has been proved. The second part follows similar arguments. $\square$

The following lemma shows that the expectation of empirical OCE risk can be upper bounded by the population OCE risk.

**Lemma E.6.** *Let $\phi$ be a disutility function, $\tau > 0$. Let $z$ be a random variable, $\{z_1, \cdots, z_m\}$ be $m$ copies of $z$. Then there holds*

$$\min_{\mu \in \mathbb{R}} \tau \mathbb{E}_z \phi((z - \mu)/\tau) + \mu \geq \mathbb{E}_{z_j} \min_{\mu \in \mathbb{R}} \frac{\tau}{m} \sum_{j=1}^{m} \phi((z_j - \mu)/\tau) + \mu.$$

*Proof.* Let $\mu_* \in \arg\min_{\mu \in \mathbb{R}} \tau \mathbb{E}_z \phi((z - \mu)/\tau) + \mu$, then

$$\mathbb{E}_{z_j} \min_{\mu \in \mathbb{R}} \frac{\tau}{m} \sum_{j=1}^{m} \phi((z_j - \mu)/\tau) + \mu \leq \mathbb{E}_{z_j} \frac{\tau}{m} \sum_{j=1}^{m} \phi((z_j - \mu_*)/\tau) + \mu_*$$
$$= \mathbb{E}_z \tau \phi((z - \mu_*)/\tau) + \mu_* = \min_{\mu \in \mathbb{R}} \tau \mathbb{E}_z \phi((z - \mu)/\tau) + \mu.$$

The proof is completed. $\square$

### E.3. Statistical consistency for $\mathcal{L}^{\phi, \ell}(s)$

In this subsection, we analyze the statistical consistency of a general class of contrastive losses $\mathcal{L}^{\phi, \ell}(s)$. To simplify the discussion and avoid measure-theoretic complications, we assume that both $\mathcal{X}, \mathcal{Y}$ are discrete spaces throughout Section E.3.

**Assumption E.7.** We assume that $\phi''(t) > 0$, $\ell$ is convex, differentiable, non-decreasing and $\ell'(0) > 0$.

*Remark* E.8. Similar assumptions for $\ell$ have been made in Gao & Zhou (2014). The condition for $\phi$ is used to analyze the optimal $\mu$ in OCE through implicit function theorem. The above assumption can be satisfied in many cases, for instance, $\phi(t) = \exp(t) - 1, \frac{1}{2}t^2 + t; \ell(t) = t, \exp(t)$.

**Theorem E.9.** *Let Assumption E.7 hold. Then $\mathcal{L}^{\phi, \ell}(s)$ is statistically consistent with $\mathcal{E}(s)$.*

*Remark* E.10. By taking $\phi(t) = \exp(t) - 1, \ell(t) = t$, the above theorem shows that standard contrastive loss $\mathcal{L}(s)$ is statistically consistent, aligning with the result in Theorem 3.1. However, the proof there depends on the explicit formulation of optimal scoring functions. While the analysis here does not require such knowledge. Furthermore, our theorem demonstrates that there exists a wide class of contrastive losses with different $\phi, \ell$ that performs well in downstream tasks, except for log-sum-exp based risk.

Compared with the pairwise loss studied in Gao & Zhou (2014), the risk $\mathcal{L}^{\phi, \ell}(s)$ admits a more complicated OCE structure, which contains a convex function $\phi$ and a minimization problem. The proof of the previous work relies on the monotonicity of $\ell'(s(\mathbf{x}, \mathbf{y}') - s(\mathbf{x}, \mathbf{y}))$ w.r.t. $\mathbf{y}$ for each $\mathbf{x}, \mathbf{y}'$. However, this property no longer holds in OCE. To address this issue, we show that the expectation of compositional derivative is decreasing.

**Lemma E.11.** *Let Assumption E.7 hold. $\psi(t) = \tau \phi(t/\tau)$. $h$ is an arbitrary function. We further define*

$$\mu(t) = \arg\min_\mu \mu + \mathbb{E}_{\mathbf{y} \sim p_{\mathbf{x}}^-} \psi(\ell(h(\mathbf{y}) - t) - \mu),$$
$$U(t) = \mathbb{E}_{\mathbf{y} \sim p_{\mathbf{x}}^-} \psi'(\ell(h(\mathbf{y}) - t) - \mu(t)) \ell'(h(\mathbf{y}) - t).$$

*Then $U(t)$ is a decreasing function of $t$.*

*Proof.* We first show that the optimal $\mu$ is unique for each $t$. Suppose

$$\mu_1, \mu_2 \in \arg\min_\mu \mu + \mathbb{E}_{\mathbf{y} \sim p_{\mathbf{x}}^-} \psi(\ell(h(\mathbf{y}) - t) - \mu), \quad \mu_1 > \mu_2.$$

Then for any $\mathbf{y}$, $\ell(h(\mathbf{y}) - t) - \mu_1 < \ell(h(\mathbf{y}) - t) - \mu_2$. Since $\psi$ is a strictly convex function, we have

$$\psi'(\ell(h(\mathbf{y}) - t) - \mu_1) - \psi'(\ell(h(\mathbf{y}) - t) - \mu_2) < 0.$$

By the optimality condition, we have $\mathbb{E}_{\mathbf{y} \sim p_{\mathbf{x}}^-} \psi'(\ell(h(\mathbf{y}) - t) - \mu_i) = 1, i = 1, 2$. Then there holds

$$
\begin{aligned}
0 =& \mathbb{E}_{\mathbf{y} \sim p_{\mathbf{x}}^-} \psi'(\ell(h(\mathbf{y}) - t) - \mu_1) - \mathbb{E}_{\mathbf{y} \sim p_{\mathbf{x}}^-} \psi'(\ell(h(\mathbf{y}) - t) - \mu_2) \\
=& \mathbb{E}_{\mathbf{y} \sim p_{\mathbf{x}}^-} (\psi'(\ell(h(\mathbf{y}) - t) - \mu_1) - \psi'(\ell(h(\mathbf{y}) - t) - \mu_2)) < 0.
\end{aligned}
$$

This contradiction implies that for each $t$, the optimal $\mu$ is unique, which we denote by $\mu(t)$. Hence,

$$\mathbb{E}_{\mathbf{y} \sim p_{\mathbf{x}}^-} \psi'(\ell(h(\mathbf{y}) - t) - \mu(t)) = 1 \tag{54}$$

for any $t$. For $t \geq s$, we prove that $U(t) \leq U(s)$.

$$
\begin{aligned}
& (t - s)U(t) \\
=& \mathbb{E}_{\mathbf{y} \sim p_{\mathbf{x}}^-} \psi'(\ell(h(\mathbf{y}) - t) - \mu(t))\ell'(h(\mathbf{y}) - t)(t - s) \\
\leq& \mathbb{E}_{\mathbf{y} \sim p_{\mathbf{x}}^-} \psi'(\ell(h(\mathbf{y}) - t) - \mu(t))(\ell(h(\mathbf{y}) - s) - \ell(h(\mathbf{y}) - t)) \\
=& \mathbb{E}_{\mathbf{y} \sim p_{\mathbf{x}}^-} \psi'(\ell(h(\mathbf{y}) - t) - \mu(t))(\ell(h(\mathbf{y}) - s) - \mu(s)) + \mu(s) \\
& - [\mathbb{E}_{\mathbf{y} \sim p_{\mathbf{x}}^-} \psi'(\ell(h(\mathbf{y}) - t) - \mu(t))(\ell(h(\mathbf{y}) - t) - \mu(t)) + \mu(t)] \\
=& \mathbb{E}_{\mathbf{y} \sim p_{\mathbf{x}}^-} \psi'(\ell(h(\mathbf{y}) - t) - \mu(t))[(\ell(h(\mathbf{y}) - s) - \mu(s)) - (\ell(h(\mathbf{y}) - t) - \mu(t))] \\
& + \mu(s) - \mu(t) \\
\leq& \mathbb{E}_{\mathbf{y} \sim p_{\mathbf{x}}^-} (\psi(\ell(h(\mathbf{y}) - s) - \mu(s)) - \psi(\ell(h(\mathbf{y}) - t) - \mu(t))) + \mu(s) - \mu(t),
\end{aligned}
$$

where the second equality is due to Eq. (54), two inequalities result from the convexity of $\ell, \psi$. Applying similar methods, we have

$$
\begin{aligned}
& (t - s)U(t) \\
\leq& \mathbb{E}_{\mathbf{y} \sim p_{\mathbf{x}}^-} (\psi(\ell(h(\mathbf{y}) - s) - \mu(s)) - \psi(\ell(h(\mathbf{y}) - t) - \mu(t))) + \mu(s) - \mu(t) \\
\leq& \mathbb{E}_{\mathbf{y} \sim p_{\mathbf{x}}^-} \psi'(\ell(h(\mathbf{y}) - s) - \mu(s))[(\ell(h(\mathbf{y}) - s) - \mu(s)) - (\ell(h(\mathbf{y}) - t) - \mu(t))] \\
& + \mu(s) - \mu(t) \\
=& \mathbb{E}_{\mathbf{y} \sim p_{\mathbf{x}}^-} \psi'(\ell(h(\mathbf{y}) - s) - \mu(s))(\ell(h(\mathbf{y}) - s) - \mu(s)) + \mu(s) \\
& - [\mathbb{E}_{\mathbf{y} \sim p_{\mathbf{x}}^-} \psi'(\ell(h(\mathbf{y}) - s) - \mu(s))(\ell(h(\mathbf{y}) - t) - \mu(t)) + \mu(t)] \\
=& \mathbb{E}_{\mathbf{y} \sim p_{\mathbf{x}}^-} \psi'(\ell(h(\mathbf{y}) - s) - \mu(s))(\ell(h(\mathbf{y}) - s) - \ell(h(\mathbf{y}) - t)) \\
\leq& \mathbb{E}_{\mathbf{y} \sim p_{\mathbf{x}}^-} \psi'(\ell(h(\mathbf{y}) - s) - \mu(s))\ell'(h(\mathbf{y}) - s)(t - s) = U(s)(t - s).
\end{aligned}
$$

Since $t \geq s$, we have $U(t) \leq U(s)$. The proof is completed. $\qquad \square$

For a function $h : \mathcal{Y} \to \mathbb{R}$, an anchor point $\mathbf{x}$, disutility function $\phi$ and pairwise loss $\ell$, we define

$$Q_{\mathbf{x}}^{\phi,\ell}(h) = \mathbb{E}_{\mathbf{y} \sim p_{\mathbf{x}}^+} \left[ \min_{\mu \in \mathbb{R}} \tau \mathbb{E}_{\mathbf{y}' \sim p_{\mathbf{x}}^-} \phi \left( \frac{\ell(h(\mathbf{y}') - h(\mathbf{y})) - \mu}{\tau} \right) + \mu \right]. \tag{55}$$

Then $\mathcal{L}^{\phi,\ell}(s) = \mathbb{E}_{\mathbf{x}} Q_{\mathbf{x}}^{\phi,\ell}(s(\mathbf{x}, \cdot))$. Therefore, $s_*$ is the minimum of $\mathcal{L}^{\phi,\ell}(s)$ if and only if $s_*(\mathbf{x}, \cdot)$ is the minimum of $Q_{\mathbf{x}}^{\phi,\ell}$ for all $\mathbf{x}$. Let $\mathcal{B}$ be the set of all scoring functions that maximize $\mathcal{E}(s)$. According to Lemma 3.3, we have

$$\mathcal{B} = \left\{ s : \left( \frac{p_{\mathbf{x}}^+(\mathbf{y})}{p_{\mathbf{x}}^-(\mathbf{y})} > \frac{p_{\mathbf{x}}^+(\mathbf{y}')}{p_{\mathbf{x}}^-(\mathbf{y}')} \right)(s(\mathbf{x}, \mathbf{y}) - s(\mathbf{x}, \mathbf{y}')) > 0, \quad \text{if } \frac{p_{\mathbf{x}}^+(\mathbf{y})}{p_{\mathbf{x}}^-(\mathbf{y})} \neq \frac{p_{\mathbf{x}}^+(\mathbf{y}')}{p_{\mathbf{x}}^-(\mathbf{y}')} \right\}.$$

To prove Theorem E.9, we present the following lemma.

**Lemma E.12.** *Let Assumption E.7 hold, then we have*

$$\inf_{s \notin \mathcal{B}} \mathcal{L}^{\phi,\ell}(s) > \inf_s \mathcal{L}^{\phi,\ell}(s).$$

*Proof of Lemma E.12.* We prove by contradiction. Note that $\inf_{s \notin \mathcal{B}} \mathcal{L}^{\phi,\ell}(s) \geq \inf_s \mathcal{L}^{\phi,\ell}(s)$. If the lemma does not hold true, we have $\inf_{s \notin \mathcal{B}} \mathcal{L}^{\phi,\ell}(s) = \inf_s \mathcal{L}^{\phi,\ell}(s)$. Then there exists an optimal scoring function $s_*$ such that $\mathcal{L}^{\phi,\ell}(s_*) = \mathcal{L}_*^{\phi,\ell}$, $s_* \notin \mathcal{B}$. Therefore, there exists $\mathbf{x} \in \mathcal{X}, \mathbf{y}_1, \mathbf{y}_2 \in \mathcal{Y}$ with

$$\frac{p_{\mathbf{x}_1}^+(\mathbf{y}_1)}{p_{\mathbf{x}}^-(\mathbf{y}_1)} - \frac{p_{\mathbf{x}}^+(\mathbf{y}_2)}{p_{\mathbf{x}}^-(\mathbf{y}_2)} > 0, s_*(\mathbf{x}, \mathbf{y}_1) - s_*(\mathbf{x}, \mathbf{y}_2) \leq 0.$$

According to Eq. (55), $h_*(\cdot) := s_*(\mathbf{x}, \cdot)$ is a minimum of $Q_{\mathbf{x}}^{\phi,\ell}$. We introduce a function $h_1$ s.t. $h_1(\mathbf{y}) \neq 0$ if $\mathbf{y} \neq \mathbf{y}_1$ and $h_1(\mathbf{y}_1) = 1$. Let $g(\beta) = Q_{\mathbf{x}_1}^{\phi,\ell}(h_* + \beta h_1)$ for $\beta \in \mathbb{R}$, then $g$ is convex. By the optimality of $h_*$, we have $g'(0) = 0$.

For simplicity, we denote $\psi = \tau\phi(\cdot/\tau)$, then $\psi'' > 0$ and

$$g(\beta) = \mathbb{E}_{\mathbf{y} \sim p_{\mathbf{x}}^+} \min_{\mu \in \mathbb{R}} \mathbb{E}_{\mathbf{y}' \sim p_{\mathbf{x}}^-} \psi(\ell(h_*(\mathbf{y}') - h_*(\mathbf{y}) + \beta(h_1(\mathbf{y}') - h_1(\mathbf{y}))) - \mu) + \mu.$$

For $\beta$ and $\mathbf{y}$, we define the optimal $\mu(\beta, h_*(\mathbf{y}))$ as

$$\mu(\beta, h_*(\mathbf{y})) = \arg\min_{\mu \in \mathbb{R}} \mathbb{E}_{\mathbf{y}' \sim p_{\mathbf{x}}^-} \psi(\ell(h_*(\mathbf{y}') - h_*(\mathbf{y}) + \beta(h_1(\mathbf{y}') - h_1(\mathbf{y}))) - \mu) + \mu.$$

It satisfies the following equality

$$\mathbb{E}_{\mathbf{y}' \sim p_{\mathbf{x}}^-} \psi'(\ell(h_*(\mathbf{y}') - h_*(\mathbf{y}) + \beta(h_1(\mathbf{y}') - h_1(\mathbf{y}))) - \mu(\beta, h_*(\mathbf{y}))) - 1 = 0. \tag{56}$$

When $\beta = 0$, there holds $\mu(0, h_*(\mathbf{y})) = \mu(h_*(\mathbf{y}))$ by the definition of $\mu(h_*(\mathbf{y}))$ in Lemma E.11. Since $\psi'' > 0$, implicit function theorem guarantees the existence of $\frac{\partial \mu(\beta, h_*(\mathbf{y}))}{\partial \beta}$. Therefore,

$$g'(0)$$
$$=\mathbb{E}_{\mathbf{y} \sim p_{\mathbf{x}}^+}\left[\mathbb{E}_{\mathbf{y}' \sim p_{\mathbf{x}}^-} \psi'(\ell(h_*(\mathbf{y}') - h_*(\mathbf{y})) - \mu(0, h_*(\mathbf{y})))\left(\ell'(h_*(\mathbf{y}') - h_*(\mathbf{y}))(h_1(\mathbf{y}') - h_1(\mathbf{y})) - \frac{\partial \mu(\beta, h_*(\mathbf{y}))}{\partial \beta}\bigg|_{\beta=0}\right)\right.$$
$$\left.+\frac{\partial \mu(\beta, h_*(\mathbf{y}))}{\partial \beta}\bigg|_{\beta=0}\right]$$
$$=\mathbb{E}_{\mathbf{y} \sim p_{\mathbf{x}}^+}\left[\mathbb{E}_{\mathbf{y}' \sim p_{\mathbf{x}}^-} \psi'(\ell(h_*(\mathbf{y}') - h_*(\mathbf{y})) - \mu(h_*(\mathbf{y})))\ell'(h_*(\mathbf{y}') - h_*(\mathbf{y}))(h_1(\mathbf{y}') - h_1(\mathbf{y}))\right.$$
$$\left.-\frac{\partial \mu(\beta, h_*(\mathbf{y}))}{\partial \beta}\bigg|_{\beta=0}(\mathbb{E}_{\mathbf{y}' \sim p_{\mathbf{x}}^-} \psi'(\ell(h_*(\mathbf{y}') - h_*(\mathbf{y})) - \mu(0, h_*(\mathbf{y}))) - 1)\right]$$
$$=\mathbb{E}_{\mathbf{y} \sim p_{\mathbf{x}}^+}\mathbb{E}_{\mathbf{y}' \sim p_{\mathbf{x}}^-} \psi'(\ell(h_*(\mathbf{y}') - h_*(\mathbf{y})) - \mu(h_*(\mathbf{y})))\ell'(h_*(\mathbf{y}') - h_*(\mathbf{y}))(h_1(\mathbf{y}') - h_1(\mathbf{y})),$$

where the last equality is due to Eq. (56) with $\beta = 0$. By noting $g'(0) = 0$ and the definition of $h_1$, we further have

$$\int_{\mathcal{Y}/\mathbf{y}_1} p_{\mathbf{x}}^+(\mathbf{y})\frac{p_{\mathbf{x}}^-(\mathbf{y}_1)}{p_{\mathbf{x}}^+(\mathbf{y}_1)}\psi'(\ell(h_*(\mathbf{y}_1) - h_*(\mathbf{y})) - \mu(h_*(\mathbf{y})))\ell'(h_*(\mathbf{y}_1) - h_*(\mathbf{y}))$$
$$-p_{\mathbf{x}}^-(\mathbf{y})\psi'(\ell(h_*(\mathbf{y}) - h_*(\mathbf{y}_1)) - \mu(h_*(\mathbf{y}_1)))\ell'(h_*(\mathbf{y}) - h_*(\mathbf{y}_1))d\mathbf{y} = 0, \tag{57}$$

In a similar way, we get

$$\int_{\mathcal{Y}/\mathbf{y}_2} p_{\mathbf{x}}^+(\mathbf{y})\frac{p_{\mathbf{x}}^-(\mathbf{y}_2)}{p_{\mathbf{x}}^+(\mathbf{y}_2)}\psi'(\ell(h_*(\mathbf{y}_2) - h_*(\mathbf{y})) - \mu(h_*(\mathbf{y})))\ell'(h_*(\mathbf{y}_2) - h_*(\mathbf{y}))$$
$$-p_{\mathbf{x}}^-(\mathbf{y})\psi'(\ell(h_*(\mathbf{y}) - h_*(\mathbf{y}_2)) - \mu(h_*(\mathbf{y}_2)))\ell'(h_*(\mathbf{y}) - h_*(\mathbf{y}_2))d\mathbf{y} = 0. \tag{58}$$

Combining Eqs. (57), (58) yields

$$A + B + C + D = 0, \tag{59}$$

where we define

$$
\begin{aligned}
A = \int_{\mathcal{Y}/\{\mathbf{y}_1, \mathbf{y}_2\}} p_{\mathbf{x}}^+(\mathbf{y}) \Bigg[ & \frac{p_{\mathbf{x}}^-(\mathbf{y}_1)}{p_{\mathbf{x}}^+(\mathbf{y}_1)} \psi'(\ell(h_*(\mathbf{y}_1) - h_*(\mathbf{y})) - \mu(h_*(\mathbf{y}))\ell'(h_*(\mathbf{y}_1) - h_*(\mathbf{y})) \\
& - \frac{p_{\mathbf{x}}^-(\mathbf{y}_2)}{p_{\mathbf{x}}^+(\mathbf{y}_2)} \psi'(\ell(h_*(\mathbf{y}_2) - h_*(\mathbf{y})) - \mu(h_*(\mathbf{y}))\ell'(h_*(\mathbf{y}_2) - h_*(\mathbf{y})) \Bigg] d\mathbf{y},
\end{aligned}
$$

$$
\begin{aligned}
B = \int_{\mathcal{Y}} p_{\mathbf{x}}^-(\mathbf{y})[&(\psi'(\ell(h_*(\mathbf{y}) - h_*(\mathbf{y}_2)) - \mu(h_*(\mathbf{y}_2))\ell'(h_*(\mathbf{y}) - h_*(\mathbf{y}_2)) \\
& - \psi'(\ell(h_*(\mathbf{y}) - h_*(\mathbf{y}_1)) - \mu(h_*(\mathbf{y}_1))\ell'(h_*(\mathbf{y}) - h_*(\mathbf{y}_1)))]d\mathbf{y},
\end{aligned}
$$

$$
\begin{aligned}
C = & p_{\mathbf{x}}^-(\mathbf{y}_1)\psi'(\ell(0) - \mu(h_*(\mathbf{y}_1))\ell'(0) \\
& - p_{\mathbf{x}}^+(\mathbf{y}_1)\frac{p_{\mathbf{x}}^-(\mathbf{y}_2)}{p_{\mathbf{x}}^+(\mathbf{y}_2)}\psi'(\ell(h_*(\mathbf{y}_2) - h_*(\mathbf{y}_1)) - \mu(h_*(\mathbf{y}_1))\ell'(h_*(\mathbf{y}_2) - h_*(\mathbf{y}_1))),
\end{aligned}
$$

$$
\begin{aligned}
D = & (p_{\mathbf{x}}^+(\mathbf{y}_2)\frac{p_{\mathbf{x}}^-(\mathbf{y}_1)}{p_{\mathbf{x}}^+(\mathbf{y}_1)}\psi'(\ell(h_*(\mathbf{y}_1) - h_*(\mathbf{y}_2)) - \mu(h_*(\mathbf{y}_2))\ell'(h_*(\mathbf{y}_1) - h_*(\mathbf{y}_2)) \\
& - p_{\mathbf{x}}^-(\mathbf{y}_2)\psi'(\ell(0) - \mu(h_*(\mathbf{y}_2))\ell'(0)).
\end{aligned}
$$

Now we bound each term respectively. Since $\frac{p_{\mathbf{x}}^-(\mathbf{y}_1)}{p_{\mathbf{x}}^+(\mathbf{y}_1)} < \frac{p_{\mathbf{x}}^-(\mathbf{y}_2)}{p_{\mathbf{x}}^+(\mathbf{y}_2)}, h_*(\mathbf{y}_1) \le h_*(\mathbf{y}_2)$, by the convexity of $\psi, \ell$, we have

$$
\begin{aligned}
\psi'(\ell(h_*(\mathbf{y}_1) - h_*(\mathbf{y})) - \mu(h_*(\mathbf{y})) &\le \psi'(\ell(h_*(\mathbf{y}_2) - h_*(\mathbf{y})) - \mu(h_*(\mathbf{y})), \\
\ell'(h_*(\mathbf{y}_1) - h_*(\mathbf{y})) &\le \ell'(h_*(\mathbf{y}_2) - h_*(\mathbf{y})), \quad \forall \mathbf{y} \in \mathcal{Y}/\{\mathbf{y}_1, \mathbf{y}_2\}.
\end{aligned}
$$

Therefore, $A \le 0$.

Applying Lemma E.11 yields

$$B = U(h_*(\mathbf{y}_2)) - U(h_*(\mathbf{y}_1)) \le 0.$$

Now we control $C$ and $D$. Note that $0 < \ell'(0) \le \ell'(h_*(\mathbf{y}_2) - h_*(\mathbf{y}_1)), \ell(0) \le \ell(h_*(\mathbf{y}_2) - h_*(\mathbf{y}_1))$. Hence, there holds

$$\psi'(\ell(0) - \mu(h_*(\mathbf{y}_1))\ell'(0) \le \psi'(\ell(h_*(\mathbf{y}_2) - h_*(\mathbf{y}_1)) - \mu(h_*(\mathbf{y}_1))\ell'(h_*(\mathbf{y}_2) - h_*(\mathbf{y}_1)).$$

As a result,

$$C \le p_{\mathbf{x}}^+(\mathbf{y}_1)\frac{p_{\mathbf{x}}^-(\mathbf{y}_2)}{p_{\mathbf{x}}^+(\mathbf{y}_2)}\psi'(\ell(h_*(\mathbf{y}_2) - h_*(\mathbf{y}_1)) - \mu(h_*(\mathbf{y}_1))\ell'(h_*(\mathbf{y}_2) - h_*(\mathbf{y}_1)) < 0.$$

In a similar manner, we can prove that $D < 0$.

Combining above results together, we have

$$A + B + C + D < 0,$$

which is contrary to Eq. (59). Therefore, the lemma holds true. $\qquad \square$

*Proof of Theorem E.9.* By Lemma E.12, we set $\delta = \inf_{s \notin \mathcal{B}} \mathcal{L}^{\phi, \ell}(s) - \inf_s \mathcal{L}^{\phi, \ell}(s) > 0$. Suppose $\{s_n\}$ be a sequence such that $\mathcal{L}^{\phi, \ell}(s_n) \to \mathcal{L}_*^{\phi, \ell}$, then there exists $N > 0$ such that

$$\mathcal{L}^{\phi, \ell}(s_n)) - \mathcal{L}_*^{\phi, \ell} < \delta/2 < \inf_{s \notin \mathcal{B}} \mathcal{L}^{\phi, \ell}(s) - \mathcal{L}_*^{\phi, \ell}, \quad n \ge N.$$

Therefore, for $n \ge N$, we have $h_n \in \mathcal{B}$. It means that $R(h_n) = R^*$. Hence, $\mathcal{L}^{\phi, \ell}(s)$ is statistically consistent with $\mathcal{E}(s)$. $\quad \square$

Now we derive the calibration-style inequality, relating the excess risks of CRL and downstream task. We make the following assumptions.

**Assumption E.13.** We assume the pairwise loss $\ell$ is linear: $\ell(t) = t$. Furthermore, suppose that $|s(\mathbf{x}, \mathbf{y})| \le B$ and $\phi''(t) \in [\gamma_1, \gamma_2]$ over $[-2B, 2B]$, where $\gamma_2 > \gamma_1 > 0$. For the optimal scoring function $s_* \in \arg\min_s \mathcal{L}^{\phi,\ell}(s)$, we assume that $|s_*(\mathbf{x}, \mathbf{y})| \le B$. Additionally, for any $\mathbf{x}, \mathbf{y}, \mathbf{y}'$, there exists a convex function $\kappa$ with $\kappa(0) = 0$ such that

$$(s_*(\mathbf{x}, \mathbf{y}) - s_*(\mathbf{x}, \mathbf{y}'))^2 \ge \kappa\left(\frac{1}{2}\left|\frac{p_\mathbf{x}^+(\mathbf{y})}{p_\mathbf{x}^-(\mathbf{y})} - \frac{p_\mathbf{x}^+(\mathbf{y}')}{p_\mathbf{x}^-(\mathbf{y}')}\right|\right).$$

**Theorem E.14.** *Let Assumption E.13 hold. Then we have*

$$\mathcal{L}(s) - \min_s \mathcal{L}(s) \gtrsim \kappa(\max_s \mathcal{E}(s) - \mathcal{E}(s)).$$

*Remark* E.15. For log-sum-exp contrastive loss $\mathcal{L}(s)$, the optimal scoring functions are captured in Lemma 3.2. According to Eq. (32), Assumption E.13 holds $\kappa(t) = 4\tau^2 t^2/\alpha^2$. We could get similar results in Theorem 3.4.

*Proof of Theorem E.14.* We denote $\psi(\cdot) = \tau\phi(\cdot/\tau)$ for simplicity. We define the norm $\|\cdot\|$ by $\|s\| = \sqrt{\mathbb{E}_\mathbf{x}\mathbb{E}_{\mathbf{y}\sim p_\mathbf{x}^-} s(\mathbf{x}, \mathbf{y})^2}$. Let

$$g_1(\mathbf{x}, \mathbf{y}) = \frac{s(\mathbf{x}, \mathbf{y}) - s_*(\mathbf{x}, \mathbf{y})}{\|s - s_*\|}. \tag{60}$$

We define $s_r = s_* + rg_1, r \in [0, \|h - h_*\|]$, then $s_0 = s_*, s_{\|h - h_*\|} = s, \sup_{\mathbf{x},\mathbf{y}} |s_r(\mathbf{x}, \mathbf{y})| \le B$. We further define

$$G(r) = \mathcal{L}^{\phi,\ell}(s_r) = \mathbb{E}_\mathbf{x}\mathbb{E}_{\mathbf{y}\sim p_\mathbf{x}^+} \min_{\mu\in\mathbb{R}} \mathbb{E}_{\mathbf{y}'\sim p_\mathbf{x}^-} \psi(s_r(\mathbf{x}, \mathbf{y}') - s_r(\mathbf{x}, \mathbf{y}) - \mu) + \mu.$$

We define the optimal $\mu$ for each triple $(r, \mathbf{x}, \mathbf{y})$ as

$$\mu(r, \mathbf{x}, \mathbf{y}) = \arg\min_{\mu\in\mathbb{R}} \mathbb{E}_{\mathbf{y}'\sim p_\mathbf{x}^-} \psi(s_r(\mathbf{x}, \mathbf{y}') - s_r(\mathbf{x}, \mathbf{y}) - \mu) + \mu.$$

By Lemma E.5 and $s_r(\mathbf{x}, \mathbf{y}') - s_r(\mathbf{x}, \mathbf{y}) \in [-2B, 2B]$, we can suppose that $\mu(r, \mathbf{x}, \mathbf{y}) \in [-2B, 2B]$. Then we have

$$\mathbb{E}_{\mathbf{y}'\sim p_\mathbf{x}^-} \psi'(s_r(\mathbf{x}, \mathbf{y}') - s_r(\mathbf{x}, \mathbf{y}) - \mu(r, \mathbf{x}, \mathbf{y})) = 1. \tag{61}$$

Therefore,

$$\begin{aligned}
G'(r) &= \mathbb{E}_\mathbf{x}\mathbb{E}_{\mathbf{y}\sim p_\mathbf{x}^+}\mathbb{E}_{\mathbf{y}'\sim p_\mathbf{x}^-} \psi'(s_r(\mathbf{x}, \mathbf{y}') - s_r(\mathbf{x}, \mathbf{y}) - \mu(r, \mathbf{x}, \mathbf{y}))[(g_1(\mathbf{x}, \mathbf{y}') - g_1(\mathbf{x}, \mathbf{y})) - \frac{\partial\mu(r, \mathbf{x}, \mathbf{y})}{\partial r}] + \frac{\partial\mu(r, \mathbf{x}, \mathbf{y})}{\partial r} \\
&= \mathbb{E}_\mathbf{x}\mathbb{E}_{\mathbf{y}\sim p_\mathbf{x}^+}\mathbb{E}_{\mathbf{y}'\sim p_\mathbf{x}^-} \psi'(s_r(\mathbf{x}, \mathbf{y}') - s_r(\mathbf{x}, \mathbf{y}) - \mu(r, \mathbf{x}, \mathbf{y}))(g_1(\mathbf{x}, \mathbf{y}') - g_1(\mathbf{x}, \mathbf{y})) \\
&\quad + \mathbb{E}_\mathbf{x}\mathbb{E}_{\mathbf{y}\sim p_\mathbf{x}^+} \frac{\partial\mu(r, \mathbf{x}, \mathbf{y})}{\partial r}(\mathbb{E}_{\mathbf{y}'\sim p_\mathbf{x}^-} \psi'(s_r(\mathbf{x}, \mathbf{y}') - s_r(\mathbf{x}, \mathbf{y}) - \mu(r, \mathbf{x}, \mathbf{y})) - 1) \\
&= \mathbb{E}_\mathbf{x}\mathbb{E}_{\mathbf{y}\sim p_\mathbf{x}^+}\mathbb{E}_{\mathbf{y}'\sim p_\mathbf{x}^-} \psi'(s_r(\mathbf{x}, \mathbf{y}') - s_r(\mathbf{x}, \mathbf{y}) - \mu(r, \mathbf{x}, \mathbf{y}))(g_1(\mathbf{x}, \mathbf{y}') - g_1(\mathbf{x}, \mathbf{y})).
\end{aligned}$$

By the optimality of $s_*$, $G$ has a minimum 0, implying that $G'(0) = 0$. We further have

$$\begin{aligned}
&G''(r) \\
=&\mathbb{E}_\mathbf{x}\mathbb{E}_{\mathbf{y}\sim p_\mathbf{x}^+}\mathbb{E}_{\mathbf{y}'\sim p_\mathbf{x}^-} \psi''(s_r(\mathbf{x}, \mathbf{y}') - s_r(\mathbf{x}, \mathbf{y}) - \mu(r, \mathbf{x}, \mathbf{y}))[(g_1(\mathbf{x}, \mathbf{y}') - g_1(\mathbf{x}, \mathbf{y})) - \frac{\partial\mu(r, \mathbf{x}, \mathbf{y})}{\partial r}](g_1(\mathbf{x}, \mathbf{y}') - g_1(\mathbf{x}, \mathbf{y})).
\end{aligned}$$

Taking derivative to Eq. (61) w.r.t. $r$, we know that

$$\mathbb{E}_{\mathbf{y}'\sim p_\mathbf{x}^-} \psi''(s_r(\mathbf{x}, \mathbf{y}') - s_r(\mathbf{x}, \mathbf{y}) - \mu(r, \mathbf{x}, \mathbf{y}))(g_1(\mathbf{x}, \mathbf{y}') - g_1(\mathbf{x}, \mathbf{y}) - \frac{\partial\mu(r, \mathbf{x}, \mathbf{y})}{\partial r}) = 0. \tag{62}$$

We define the density function for each $(\mathbf{x}, \mathbf{y}, r)$:

$$q_{r,\mathbf{x},\mathbf{y}}(\mathbf{y}') = \frac{p_\mathbf{x}^-(\mathbf{y}')\psi''(s_r(\mathbf{x}, \mathbf{y}') - s_r(\mathbf{x}, \mathbf{y}) - \mu(r, \mathbf{x}, \mathbf{y}))}{\mathbb{E}_{\mathbf{y}'\sim p_\mathbf{x}^-} \psi''(s_r(\mathbf{x}, \mathbf{y}') - s_r(\mathbf{x}, \mathbf{y}) - \mu(r, \mathbf{x}, \mathbf{y}))}.$$

Then by Eq. (62), we have

$$\frac{\partial \mu(r, \mathbf{x}, \mathbf{y})}{\partial r} = \frac{\mathbb{E}_{\mathbf{y}' \sim p_{\mathbf{x}}^-} \psi''(s_r(\mathbf{x}, \mathbf{y}') - s_r(\mathbf{x}, \mathbf{y}) - \mu(r, \mathbf{x}, \mathbf{y}))(g_1(\mathbf{x}, \mathbf{y}') - g_1(\mathbf{x}, \mathbf{y}))}{\mathbb{E}_{\mathbf{y}' \sim p_{\mathbf{x}}^-} \psi''(s_r(\mathbf{x}, \mathbf{y}') - s_r(\mathbf{x}, \mathbf{y}) - \mu(r, \mathbf{x}, \mathbf{y}))}$$

$$= \int q_{r, \mathbf{x}, \mathbf{y}}(\mathbf{y}')(g_1(\mathbf{x}, \mathbf{y}') - g_1(\mathbf{x}, \mathbf{y}))d\mathbf{y}' = \mathbb{E}_{\mathbf{y}' \sim q_{r, \mathbf{x}, \mathbf{y}}}(g_1(\mathbf{x}, \mathbf{y}') - g_1(\mathbf{x}, \mathbf{y})).$$

Consequently,

$$G''(r)$$

$$= \mathbb{E}_{\mathbf{x}} \mathbb{E}_{\mathbf{y} \sim p_{\mathbf{x}}^+} \mathbb{E}_{\mathbf{y}' \sim p_{\mathbf{x}}^-} \psi''(s_r(\mathbf{x}, \mathbf{y}') - s_r(\mathbf{x}, \mathbf{y}) - \mu(r, \mathbf{x}, \mathbf{y}))[(g_1(\mathbf{x}, \mathbf{y}') - g_1(\mathbf{x}, \mathbf{y})) - \frac{\partial \mu(r, \mathbf{x}, \mathbf{y})}{\partial r}](g_1(\mathbf{x}, \mathbf{y}') - g_1(\mathbf{x}, \mathbf{y}))$$

$$= \mathbb{E}_{\mathbf{x}} \mathbb{E}_{\mathbf{y} \sim p_{\mathbf{x}}^+} \mathbb{E}_{\mathbf{y}' \sim p_{\mathbf{x}}^-} \psi''(s_r(\mathbf{x}, \mathbf{y}') - s_r(\mathbf{x}, \mathbf{y}) - \mu(r, \mathbf{x}, \mathbf{y}))[(g_1(\mathbf{x}, \mathbf{y}') - g_1(\mathbf{x}, \mathbf{y})) - \frac{\partial \mu(r, \mathbf{x}, \mathbf{y})}{\partial r}](g_1(\mathbf{x}, \mathbf{y}') - g_1(\mathbf{x}, \mathbf{y}))$$

$$- \mathbb{E}_{\mathbf{x}} \mathbb{E}_{\mathbf{y} \sim p_{\mathbf{x}}^+} \mathbb{E}_{\mathbf{y}' \sim p_{\mathbf{x}}^-} \psi''(s_r(\mathbf{x}, \mathbf{y}') - s_r(\mathbf{x}, \mathbf{y}) - \mu(r, \mathbf{x}, \mathbf{y}))(g_1(\mathbf{x}, \mathbf{y}') - g_1(\mathbf{x}, \mathbf{y}) - \frac{\partial \mu(r, \mathbf{x}, \mathbf{y})}{\partial r}) \frac{\partial \mu(r, \mathbf{x}, \mathbf{y})}{\partial r}$$

$$= \mathbb{E}_{\mathbf{x}} \mathbb{E}_{\mathbf{y} \sim p_{\mathbf{x}}^+} \mathbb{E}_{\mathbf{y}' \sim p_{\mathbf{x}}^-} \psi''(s_r(\mathbf{x}, \mathbf{y}') - s_r(\mathbf{x}, \mathbf{y}) - \mu(r, \mathbf{x}, \mathbf{y}))[(g_1(\mathbf{x}, \mathbf{y}') - g_1(\mathbf{x}, \mathbf{y})) - \frac{\partial \mu(r, \mathbf{x}, \mathbf{y})}{\partial r}]^2, \tag{63}$$

where the first equality is due to Eq. (62). By the definition of $q_{r, \mathbf{x}, \mathbf{y}}$, we have

$$\mathbb{E}_{\mathbf{y}' \sim p_{\mathbf{x}}^-} \psi''(s_r(\mathbf{x}, \mathbf{y}') - s_r(\mathbf{x}, \mathbf{y}) - \mu(r, \mathbf{x}, \mathbf{y}))[(g_1(\mathbf{x}, \mathbf{y}') - g_1(\mathbf{x}, \mathbf{y})) - \frac{\partial \mu(r, \mathbf{x}, \mathbf{y})}{\partial r}]^2$$

$$= (\mathbb{E}_{\mathbf{y}' \sim p_{\mathbf{x}}^-} \psi''(s_r(\mathbf{x}, \mathbf{y}') - s_r(\mathbf{x}, \mathbf{y}) - \mu(r, \mathbf{x}, \mathbf{y})))$$

$$\times \mathbb{E}_{\mathbf{y}' \sim p_{\mathbf{x}}^-} \frac{\psi''(s_r(\mathbf{x}, \mathbf{y}') - s_r(\mathbf{x}, \mathbf{y}) - \mu(r, \mathbf{x}, \mathbf{y}))}{\mathbb{E}_{\mathbf{y}' \sim p_{\mathbf{x}}^-} \psi''(s_r(\mathbf{x}, \mathbf{y}') - s_r(\mathbf{x}, \mathbf{y}) - \mu(r, \mathbf{x}, \mathbf{y}))} [(g_1(\mathbf{x}, \mathbf{y}') - g_1(\mathbf{x}, \mathbf{y})) - \frac{\partial \mu(r, \mathbf{x}, \mathbf{y})}{\partial r}]^2$$

$$= (\mathbb{E}_{\mathbf{y}' \sim p_{\mathbf{x}}^-} \psi''(s_r(\mathbf{x}, \mathbf{y}') - s_r(\mathbf{x}, \mathbf{y}) - \mu(r, \mathbf{x}, \mathbf{y}))) \mathbb{E}_{\mathbf{y}' \sim q_{r, \mathbf{x}, \mathbf{y}}}[(g_1(\mathbf{x}, \mathbf{y}') - g_1(\mathbf{x}, \mathbf{y})) - \frac{\partial \mu(r, \mathbf{x}, \mathbf{y})}{\partial r}]^2$$

$$= \mathbb{E}_{\mathbf{y}' \sim p_{\mathbf{x}}^-} \psi''(s_r(\mathbf{x}, \mathbf{y}') - s_r(\mathbf{x}, \mathbf{y}) - \mu(r, \mathbf{x}, \mathbf{y})) \mathbb{E}_{\mathbf{y}' \sim q_{r, \mathbf{x}, \mathbf{y}}}[(g_1(\mathbf{x}, \mathbf{y}') - g_1(\mathbf{x}, \mathbf{y})) - \mathbb{E}_{\mathbf{y}' \sim q_{r, \mathbf{x}, \mathbf{y}}}(g(\mathbf{x}, \mathbf{y}') - g(\mathbf{x}, \mathbf{y}))]^2$$

$$\geq \frac{\gamma_1}{\tau} \text{Var}_{\mathbf{y}' \sim q_{r, \mathbf{x}, \mathbf{y}}}[(g_1(\mathbf{x}, \mathbf{y}') - g_1(\mathbf{x}, \mathbf{y}))], \tag{64}$$

where the last inequality results from $\psi''(t) = \phi''(t/\tau)/\tau \geq \gamma_1/\tau$. According to Lemma 3.2, we can choose $s_*$ such that for all $\mathbf{x}$, $\mathbb{E}_{\mathbf{y} \sim p_{\mathbf{x}}^-} s(\mathbf{x}, \mathbf{y}) = \mathbb{E}_{\mathbf{y} \sim p_{\mathbf{x}}^-} s_*(\mathbf{x}, \mathbf{y})$. From the definition of $q_{r, \mathbf{x}, \mathbf{y}}$, we know that $q_{r, \mathbf{x}, \mathbf{y}}(\mathbf{y}')/p_{\mathbf{x}}^-(\mathbf{y}') \geq \gamma_1/\gamma_2$ for all $\mathbf{y}' \in \mathcal{Y}$. Hence, for fixed $r, \mathbf{x}, \mathbf{y}$,

$$\text{Var}_{\mathbf{y}' \sim q_{r, \mathbf{x}, \mathbf{y}}}[(g_1(\mathbf{x}, \mathbf{y}') - g_1(\mathbf{x}, \mathbf{y}))] = \frac{1}{2} \mathbb{E}_{\mathbf{y}', \mathbf{y}'' \sim q_{r, \mathbf{x}, \mathbf{y}}}(g_1(\mathbf{x}, \mathbf{y}') - g_1(\mathbf{x}, \mathbf{y}''))^2$$

$$= \frac{1}{2} \int \int q_{r, \mathbf{x}, \mathbf{y}}(\mathbf{y}') q_{r, \mathbf{x}, \mathbf{y}''}(\mathbf{y}'')(g_1(\mathbf{x}, \mathbf{y}') - g_1(\mathbf{x}, \mathbf{y}''))^2 d\mathbf{y}' d\mathbf{y}''$$

$$\geq \frac{\gamma_1^2}{2\gamma_2^2} \int \int p_{\mathbf{x}}^-(\mathbf{y}') p_{\mathbf{x}}^-(\mathbf{y}'')(g_1(\mathbf{x}, \mathbf{y}') - g_1(\mathbf{x}, \mathbf{y}''))^2 d\mathbf{y}' d\mathbf{y}''$$

$$= \frac{\gamma_1^2}{2\gamma_2^2} \mathbb{E}_{\mathbf{y}', \mathbf{y}'' \sim p_{\mathbf{x}}^-}(g_1(\mathbf{x}, \mathbf{y}') - g_1(\mathbf{x}, \mathbf{y}))^2$$

$$= \frac{\gamma_1^2}{2\gamma_2^2} \mathbb{E}_{\mathbf{y}', \mathbf{y}'' \sim p_{\mathbf{x}}^-}(g_1(\mathbf{x}, \mathbf{y}')^2 + g_1(\mathbf{x}, \mathbf{y}'')^2) - \frac{\gamma_1^2}{\gamma_2^2} \mathbb{E}_{\mathbf{y}' \sim p_{\mathbf{x}}^-} g_1(\mathbf{x}, \mathbf{y}') \mathbb{E}_{\mathbf{y}'' \sim p_{\mathbf{x}}^-} g_1(\mathbf{x}, \mathbf{y}'') = \frac{\gamma_1^2}{\gamma_2^2} \mathbb{E}_{\mathbf{y}' \sim p_{\mathbf{x}}^-} g_1(\mathbf{x}, \mathbf{y}')^2, \tag{65}$$

where the first inequality is due to $\text{Var}[X] = \frac{1}{2} \mathbb{E}_{X, X'}(X - X')^2$, the last inequality is by

$$\mathbb{E}_{\mathbf{y}' \sim p_{\mathbf{x}}^-} g_1(\mathbf{x}, \mathbf{y}) = \frac{\mathbb{E}_{\mathbf{y}' \sim p_{\mathbf{x}}^-} s(\mathbf{x}, \mathbf{y}') - \mathbb{E}_{\mathbf{y}' \sim p_{\mathbf{x}}^-} s_*(\mathbf{x}, \mathbf{y}')}{\|s - s_*\|_2} = 0.$$

Combining Eqs. (63), (64), (65) together, we have

$$G''(r) \geq \frac{\gamma_1^3}{\gamma_2^2 \tau} \mathbb{E}_{\mathbf{x}} \mathbb{E}_{\mathbf{y} \sim p_{\mathbf{x}}^+} \mathbb{E}_{\mathbf{y}' \sim p_{\mathbf{x}}^-} g_1(\mathbf{x}, \mathbf{y}')^2 = \frac{\gamma_1^3}{\gamma_2^2 \tau} \mathbb{E}_{\mathbf{x}} \mathbb{E}_{\mathbf{y}' \sim p_{\mathbf{x}}^-} g_1(\mathbf{x}, \mathbf{y}')^2$$
$$= \frac{\gamma_1^3}{\gamma_2^2 \tau} \mathbb{E}_{\mathbf{x}} \mathbb{E}_{\mathbf{y}' \sim p_{\mathbf{x}}^-} \frac{(s(\mathbf{x}, \mathbf{y}') - s_*(\mathbf{x}, \mathbf{y}'))^2}{\|s - s_*\|_2^2} = \frac{\gamma_1^3}{\gamma_2^2 \tau}.$$

Applying Taylor's expansion yields

$$\mathcal{L}^{\phi, \ell}(s) - \mathcal{L}^{\phi, \ell}(s_*) = G(\|s - s_*\|) - G(0) = G'(0)\|s - s_*\| + \frac{1}{2}G''(\xi)\|s - s_*\|^2, \quad \xi \in [0, \|s - s_*\|]$$
$$\geq \frac{\gamma_1^3}{2\gamma_2^2 \tau} \mathbb{E}_{\mathbf{x}} \mathbb{E}_{\mathbf{y} \sim p_{\mathbf{x}}^-} (s(\mathbf{x}, \mathbf{y}) - s_*(\mathbf{x}, \mathbf{y}))^2.$$

Then we make the following decomposition that introduces the pair $\mathbf{y}, \mathbf{y}'$:

$$2\mathbb{E}_{\mathbf{y} \sim p_{\mathbf{x}}^-} |s(\mathbf{x}, \mathbf{y}) - s_*(\mathbf{x}, \mathbf{y})|^2$$
$$= \mathbb{E}_{\mathbf{y} \sim p_{\mathbf{x}}^-} |s(\mathbf{x}, \mathbf{y}) - s_*(\mathbf{x}, \mathbf{y})|^2 + \mathbb{E}_{\mathbf{y}' \sim p_{\mathbf{x}}^-} |s(\mathbf{x}, \mathbf{y}') - s_*(\mathbf{x}, \mathbf{y}')|^2$$
$$= \mathbb{E}_{\mathbf{y} \sim p_{\mathbf{x}}^-, \mathbf{y}' \sim p_{\mathbf{x}}^-} (|s(\mathbf{x}, \mathbf{y}) - s_*(\mathbf{x}, \mathbf{y})|^2 + |s(\mathbf{x}, \mathbf{y}') - s_*(\mathbf{x}, \mathbf{y}')|^2). \tag{66}$$

We will show that for any $\mathbf{x}, \mathbf{y}, \mathbf{y}'$,

$$|s(\mathbf{x}, \mathbf{y}) - s_*(\mathbf{x}, \mathbf{y})|^2 + |s(\mathbf{x}, \mathbf{y}') - s_*(\mathbf{x}, \mathbf{y}')|^2$$
$$\geq \frac{1}{2} \mathbb{I}\left[\left(\frac{p_{\mathbf{x}}^+(\mathbf{y})}{p_{\mathbf{x}}^-(\mathbf{y})} - \frac{p_{\mathbf{x}}^+(\mathbf{y}')}{p_{\mathbf{x}}^-(\mathbf{y}')}\right)(s(\mathbf{x}, \mathbf{y}) - s(\mathbf{x}, \mathbf{y}')) \leq 0\right] \kappa\left(\frac{1}{2}\left|\frac{p_{\mathbf{x}}^+(\mathbf{y})}{p_{\mathbf{x}}^-(\mathbf{y})} - \frac{p_{\mathbf{x}}^+(\mathbf{y}')}{p_{\mathbf{x}}^-(\mathbf{y}')}\right|\right). \tag{67}$$

It is trivial when RHS is equal to $0$. We consider the case

$$(\frac{p_{\mathbf{x}}^+(\mathbf{y})}{p_{\mathbf{x}}^-(\mathbf{y})} - \frac{p_{\mathbf{x}}^+(\mathbf{y}')}{p_{\mathbf{x}}^-(\mathbf{y}')})(s(\mathbf{x}, \mathbf{y}) - s(\mathbf{x}, \mathbf{y}')) \leq 0, \quad \frac{p_{\mathbf{x}}^+(\mathbf{y})}{p_{\mathbf{x}}^-(\mathbf{y})} - \frac{p_{\mathbf{x}}^+(\mathbf{y}')}{p_{\mathbf{x}}^-(\mathbf{y}')} \neq 0.$$

Without loss of generality, suppose that

$$\frac{p_{\mathbf{x}}^+(\mathbf{y})}{p_{\mathbf{x}}^-(\mathbf{y})} > \frac{p_{\mathbf{x}}^+(\mathbf{y}')}{p_{\mathbf{x}}^-(\mathbf{y}')}, \quad s(\mathbf{x}, \mathbf{y}) - s(\mathbf{x}, \mathbf{y}') \leq 0.$$

Therefore,

$$|s(\mathbf{x}, \mathbf{y}) - s_*(\mathbf{x}, \mathbf{y})|^2 + |s(\mathbf{x}, \mathbf{y}') - s_*(\mathbf{x}, \mathbf{y}')|^2$$
$$\geq \frac{1}{2}|s(\mathbf{x}, \mathbf{y}) - s_*(\mathbf{x}, \mathbf{y}) - s(\mathbf{x}, \mathbf{y}') + s_*(\mathbf{x}, \mathbf{y}')|^2$$
$$= \frac{1}{2}|s(\mathbf{x}, \mathbf{y}') - s(\mathbf{x}, \mathbf{y}) + s_*(\mathbf{x}, \mathbf{y}) - s_*(\mathbf{x}, \mathbf{y}')|^2$$
$$= \frac{1}{2}(s(\mathbf{x}, \mathbf{y}') - s(\mathbf{x}, \mathbf{y}) + s_*(\mathbf{x}, \mathbf{y}) - s_*(\mathbf{x}, \mathbf{y}'))$$
$$\geq \frac{1}{2}(s_*(\mathbf{x}, \mathbf{y}) - s_*(\mathbf{x}, \mathbf{y}'))^2 \geq \frac{1}{2}\kappa\left(\frac{1}{2}\left|\frac{p_{\mathbf{x}}^+(\mathbf{y})}{p_{\mathbf{x}}^-(\mathbf{y})} - \frac{p_{\mathbf{x}}^+(\mathbf{y}')}{p_{\mathbf{x}}^-(\mathbf{y}')}\right|\right).$$

As a result, Eq. (67) holds true.

According to Eq. (26),

$$\mathcal{E}^* - \mathcal{E}(s) \leq \frac{1}{2} \mathbb{E}_{\mathbf{x}} \mathbb{E}_{\mathbf{y} \sim p_{\mathbf{x}}^-, \mathbf{y}' \sim p_{\mathbf{x}}^-} \mathbb{I}[(\frac{p_{\mathbf{x}}^+(\mathbf{y})}{p_{\mathbf{x}}^-(\mathbf{y})} - \frac{p_{\mathbf{x}}^+(\mathbf{y}')}{p_{\mathbf{x}}^-(\mathbf{y}')})(s(\mathbf{x}, \mathbf{y}) - s(\mathbf{x}, \mathbf{y}')) \leq 0]|\frac{p_{\mathbf{x}}^+(\mathbf{y})}{p_{\mathbf{x}}^-(\mathbf{y})} - \frac{p_{\mathbf{x}}^+(\mathbf{y}')}{p_{\mathbf{x}}^-(\mathbf{y}')}|.$$

Therefore,

$$
\begin{aligned}
\mathcal{L}^{\phi,\ell}(s) - \mathcal{L}^{\phi,\ell}(s_*) &\geq \frac{\gamma_1^3}{2\gamma_2^2\tau}\mathbb{E}_{\mathbf{x}}\mathbb{E}_{\mathbf{y}\sim p_{\mathbf{x}}^-}(s(\mathbf{x},\mathbf{y}) - s_*(\mathbf{x},\mathbf{y}))^2 \\
&\geq \frac{\gamma_1^3}{8\gamma_2^2\tau}\mathbb{E}_{\mathbf{x}}\mathbb{E}_{\mathbf{y},\mathbf{y}'\sim\mathbf{p}_{\mathbf{x}}^-}\left[\mathbb{I}[(\frac{p_{\mathbf{x}}^+(\mathbf{y})}{p_{\mathbf{x}}^-(\mathbf{y})} - \frac{p_{\mathbf{x}}^+(\mathbf{y}')}{p_{\mathbf{x}}^-(\mathbf{y}')})(s(\mathbf{x},\mathbf{y}) - s(\mathbf{x},\mathbf{y}')) \leq 0]\kappa(\frac{1}{2}|\frac{p_{\mathbf{x}}^+(\mathbf{y})}{p_{\mathbf{x}}^-(\mathbf{y})} - \frac{p_{\mathbf{x}}^+(\mathbf{y}')}{p_{\mathbf{x}}^-(\mathbf{y}')}|)\right] \\
&= \frac{\gamma_1^3}{8\gamma_2^2\tau}\mathbb{E}_{\mathbf{x}}\mathbb{E}_{\mathbf{y},\mathbf{y}'\sim\mathbf{p}_{\mathbf{x}}^-}\left[\kappa(\frac{1}{2}|\frac{p_{\mathbf{x}}^+(\mathbf{y})}{p_{\mathbf{x}}^-(\mathbf{y})} - \frac{p_{\mathbf{x}}^+(\mathbf{y}')}{p_{\mathbf{x}}^-(\mathbf{y}')}|\mathbb{I}[(\frac{p_{\mathbf{x}}^+(\mathbf{y})}{p_{\mathbf{x}}^-(\mathbf{y})} - \frac{p_{\mathbf{x}}^+(\mathbf{y}')}{p_{\mathbf{x}}^-(\mathbf{y}')})(s(\mathbf{x},\mathbf{y}) - s(\mathbf{x},\mathbf{y}')) \leq 0])\right] \\
&\geq \frac{\gamma_1^3}{8\gamma_2^2\tau}\kappa(\frac{1}{2}\mathbb{E}_{\mathbf{x}}\mathbb{E}_{\mathbf{y},\mathbf{y}'\sim\mathbf{p}_{\mathbf{x}}^-}\mathbb{I}[(\frac{p_{\mathbf{x}}^+(\mathbf{y})}{p_{\mathbf{x}}^-(\mathbf{y})} - \frac{p_{\mathbf{x}}^+(\mathbf{y}')}{p_{\mathbf{x}}^-(\mathbf{y}')})(s(\mathbf{x},\mathbf{y}) - s(\mathbf{x},\mathbf{y}')) \leq 0]|\frac{p_{\mathbf{x}}^+(\mathbf{y})}{p_{\mathbf{x}}^-(\mathbf{y})} - \frac{p_{\mathbf{x}}^+(\mathbf{y}')}{p_{\mathbf{x}}^-(\mathbf{y}')}|) \\
&= \frac{\gamma_1^3}{8\gamma_2^2\tau}\kappa(\mathcal{E}^* - \mathcal{E}(s)),
\end{aligned}
$$

where the first equality results from $\kappa(0) = 0$, the last inequality is due to Jenson's inequality and the convexity of $\kappa$. The proof is completed. $\square$

### E.4. Generalization analysis of SCRL for general OCE and pairwise loss

In this subsection, we give generalization analysis of SCRL for general OCE and pairwise loss. For disutility function $\phi$ and pairwise loss $\ell$, recall that

$$
\begin{aligned}
\mathcal{L}_{\mathrm{S}}^{\phi,\ell}(s_{\mathbf{w}}) &= \mathbb{E}_{\mathbf{x}}\mathbb{E}_{\mathbf{y}}\left[\min_{\mu\in\mathbb{R}}\tau\mathbb{E}_{\mathbf{y}'}\phi\left(\frac{\ell(\Delta_{\mathbf{w}}(\mathbf{x},\mathbf{y},\mathbf{y}')) - \mu}{\tau}\right) + \mu\right], \\
\widehat{\mathcal{L}}_{\mathrm{S}}^{\phi,\ell}(s_{\mathbf{w}}) &= \frac{1}{n}\sum_{i=1}^{n}\left[\min_{\mu_i\in\mathbb{R}}\frac{\tau}{m}\sum_{j=1}^{m}\phi\left(\frac{\ell(\Delta_{\mathbf{w}}(\mathbf{x}_i,\mathbf{y}_i,\mathbf{y}'_{ij})) - \mu_i}{\tau}\right) + \mu_i\right].
\end{aligned}
$$

We aim to control the uniform convergence bound

$$
\sup_{\mathbf{w}\in\mathcal{W}}\left|\mathcal{L}_{\mathrm{S}}^{\phi,\ell}(s_{\mathbf{w}}) - \widehat{\mathcal{L}}_{\mathrm{S}}^{\phi,\ell}(s_{\mathbf{w}})\right|.
$$

We make similar error decomposition as Eq. (12).

$$
\left|\mathcal{L}_{\mathrm{S}}^{\phi,\ell}(s_{\mathbf{w}}) - \widehat{\mathcal{L}}_{\mathrm{S}}^{\phi,\ell}(s_{\mathbf{w}})\right| \leq \underbrace{\left|\mathcal{L}_{\mathrm{S}}^{\phi,\ell}(s_{\mathbf{w}}) - \mathbb{E}\left[\widehat{\mathcal{L}}_{\mathrm{S}}^{\phi,\ell}(s_{\mathbf{w}})\right]\right|}_{\text{inner error}} + \underbrace{\left|\mathbb{E}\left[\widehat{\mathcal{L}}_{\mathrm{S}}^{\phi,\ell}(s_{\mathbf{w}})\right] - \widehat{\mathcal{L}}_{\mathrm{S}}^{\phi,\ell}(s_{\mathbf{w}})\right|}_{\text{outer error}}. \tag{68}
$$

We will estimate inner error and outer error respectively.

To start, we impose mild assumptions on the pairwise loss function $\ell$ and the disutility function $\phi$.

**Assumption E.16.** Assume that $\ell$ is $G$-Lipschitz continuous over the interval $[-2B, 2B]$. Moreover, there exists a constant $M > 0$ such that $\ell(t) \in [-M, M]$ for all $t \in [-2B, 2B]$.

**Assumption E.17.** The disutility function $\phi$ is $\eta$-Lipschitz continuous over the interval $[-2M/\tau, 2M/\tau]$.

**Assumption E.18.** Suppose that $\phi$ is $\gamma$-strongly convex over the interval $[-2M/\tau, 2M/\tau]$, where $\gamma > 0$ denotes the strong convexity parameter.

*Remark* E.19. It is worth noting that the above assumptions do not require $\ell$ or $\phi$ to satisfy Lipschitz continuity or strong convexity globally over $\mathbb{R}$—a condition that is often too restrictive for practical functions in CRL. Instead, our analysis is confined to the bounded domain of $\ell(t)$, which lies within $[-2B, 2B]$ when Assumption 4.1 holds. This boundedness renders Assumption E.16 valid for most commonly used pairwise loss functions in practice.

By the same token, the Lipschitz continuity and strong convexity of $\phi$ only need to hold over the bounded interval $[-2M/\tau, 2M/\tau]$, which is induced by the boundedness of $\ell(t)$ (Assumption E.16). For instance, consider the exponential disutility function $\phi(t) = \exp(t) - 1$, which is neither Lipschitz continuous nor strongly convex over the entire real line. However, over the bounded interval $[-2M/\tau, 2M/\tau]$, $\phi$ is $\exp(2M/\tau)$-Lipschitz continuous and $\exp(-2M/\tau)$-strongly convex.

### E.4.1. OUTER ERROR

For $\mathbf{w} \in \mathcal{W}$, we introduce $g_{\mathbf{w}} : \mathcal{X} \times \mathcal{Y}^{m+1} \to \mathbb{R}$ as follows:

$$g_{\mathbf{w}}(\mathbf{x}, \mathbf{y}, \mathbf{y}'_1, \cdots, \mathbf{y}'_m) = \min_{\mu \in \mathbb{R}} \frac{\tau}{m} \sum_{j=1}^{m} \phi\left( \frac{\ell(\Delta_{\mathbf{w}}(\mathbf{x}, \mathbf{y}, \mathbf{y}'_j)) - \mu}{\tau} \right) + \mu.$$

Then we have

$$
\begin{aligned}
\mathbb{E}\left[ \widehat{\mathcal{L}}_{\mathrm{S}}^{\phi, \ell}(\mathbf{w}) \right] - \widehat{\mathcal{L}}_{\mathrm{S}}^{\phi, \ell}(\mathbf{w}) =& \mathbb{E}_{\mathbf{x}, \mathbf{y}, \mathbf{y}'_1, \cdots, \mathbf{y}'_m} \left[ \frac{1}{n} \sum_{i=1}^{n} \left( \min_{\mu_i \in \mathbb{R}} \frac{\tau}{m} \sum_{j=1}^{m} \phi\left( \frac{\ell(\Delta_{\mathbf{w}}(\mathbf{x}, \mathbf{y}, \mathbf{y}'_j)) - \mu_i}{\tau} \right) + \mu_i \right) \right] \\
&- \frac{1}{n} \sum_{i=1}^{n} \left( \min_{\mu_i \in \mathbb{R}} \frac{\tau}{m} \sum_{j=1}^{m} \phi\left( \frac{\ell(\Delta_{\mathbf{w}}(\mathbf{x}_i, \mathbf{y}_i, \mathbf{y}'_{ij})) - \mu_i}{\tau} \right) + \mu_i \right) \\
=& \mathbb{E}_{\mathbf{x}, \mathbf{y}, \mathbf{y}'_1, \cdots, \mathbf{y}'_m} \left[ g_{\mathbf{w}}(\mathbf{x}, \mathbf{y}, \mathbf{y}'_1, \cdots, \mathbf{y}'_m) \right] - \frac{1}{n} \sum_{i=1}^{n} g_{\mathbf{w}}(\mathbf{x}_i, \mathbf{y}_i, \mathbf{y}'_{i1}, \cdots, \mathbf{y}'_{im}).
\end{aligned}
$$

By Proposition A.1, we can estimate it through Rademacher complexity $\mathfrak{R}_S(\mathcal{G})$, where

$$\mathcal{G} = \left\{ (\mathbf{x}, \mathbf{y}, \mathbf{y}'_1, \cdots, \mathbf{y}'_m) \mapsto g_{\mathbf{w}}(\mathbf{x}, \mathbf{y}, \mathbf{y}'_1, \cdots, \mathbf{y}'_m) : \mathbf{w} \in \mathcal{W} \right\}.$$

and recall that

$$S = \{ (\mathbf{x}_1, \mathbf{y}_1, \mathbf{y}'_{11}, \cdots, \mathbf{y}'_{1m}), \cdots, (\mathbf{x}_n, \mathbf{y}_n, \mathbf{y}'_{n1}, \cdots, \mathbf{y}'_{nm}) \}.$$

According to Proposition A.3, $\mathfrak{R}_S(\mathcal{G})$ can be controlled by $\mathcal{N}(\mathcal{G}, \epsilon, L_2(S))$. Let $S_1, \mathcal{H}$ be defined in Eqs. (19), (20). Then we have

**Lemma E.20.** *Let Assumptions 4.1, E.16 hold. Then we have*

$$\mathcal{N}(\mathcal{G}, \epsilon, L_2(S)) \le \mathcal{N}(\mathcal{H}, \epsilon/(G), L_{\infty}(S_1)).$$

*Proof.* Let $g_{\mathbf{w}}, g_{\mathbf{w}'}$ be two functions in $\mathcal{G}$. For $i \in [n]$, we define

$$\mathbf{u}_i = ((\ell(\Delta_{\mathbf{w}}(\mathbf{x}_i, \mathbf{y}_i, \mathbf{y}'_{i1})), \cdots, \ell(\Delta_{\mathbf{w}}(\mathbf{x}_i, \mathbf{y}_i, \mathbf{y}'_{im}))), \mathbf{u}'_i = (\ell(\Delta_{\mathbf{w}'}(\mathbf{x}_i, \mathbf{y}_i, \mathbf{y}'_{i1})), \cdots, \ell(\Delta_{\mathbf{w}'}(\mathbf{x}_i, \mathbf{y}_i, \mathbf{y}'_{im}))).$$

According to Lemma E.4, there holds

$$
\begin{aligned}
\| g_{\mathbf{w}} - g_{\mathbf{w}'} \|_{L_2(S)} =& \sqrt{\frac{1}{n} \sum_{i=1}^{n} (g_{\mathbf{w}}(\mathbf{x}_i, \mathbf{y}_i, \mathbf{y}'_{i1}, \cdots, \mathbf{y}'_{im}) - g_{\mathbf{w}'}(\mathbf{x}_i, \mathbf{y}_i, \mathbf{y}'_{i1}, \cdots, \mathbf{y}'_{im}))^2} \\
\le& \max_i |g_{\mathbf{w}}(\mathbf{x}_i, \mathbf{y}_i, \mathbf{y}'_{i1}, \cdots, \mathbf{y}'_{im}) - g_{\mathbf{w}'}(\mathbf{x}_i, \mathbf{y}_i, \mathbf{y}'_{i1}, \cdots, \mathbf{y}'_{im})| \\
=& \max_i |q(\mathbf{u}_i) - q(\bar{\mathbf{u}}_i)| \le \max_i \| \mathbf{u}_i - \bar{\mathbf{u}}_i \|_{\infty} \\
=& \max_i \max_j |\ell(\Delta_{\mathbf{w}}(\mathbf{x}_i, \mathbf{y}_i, \mathbf{y}'_{ij})) - \ell(\Delta_{\mathbf{w}'}(\mathbf{x}_i, \mathbf{y}_i, \mathbf{y}'_{ij}))| \\
\le& G \max_i \max_j |\Delta_{\mathbf{w}}(\mathbf{x}_i, \mathbf{y}_i, \mathbf{y}'_{ij}) - \Delta_{\mathbf{w}'}(\mathbf{x}_i, \mathbf{y}_i, \mathbf{y}'_{ij})| \\
\le& G \max_{(\hat{\mathbf{x}}, \hat{\mathbf{y}}, \hat{\mathbf{y}}') \in S_1} |\Delta_{\mathbf{w}}(\hat{\mathbf{x}}, \hat{\mathbf{y}}, \hat{\mathbf{y}}') - \Delta_{\mathbf{w}'}(\hat{\mathbf{x}}, \hat{\mathbf{y}}, \hat{\mathbf{y}}')| = G \| \Delta_{\mathbf{w}} - \Delta_{\mathbf{w}'} \|_{L_{\infty}(S_1)}.
\end{aligned}
$$

Therefore,

$$\mathcal{N}(\mathcal{G}, \epsilon, L_2(S)) \le \mathcal{N}(\mathcal{H}, \epsilon/(G), L_{\infty}(S_1)).$$

The proof is completed. $\qquad\square$

The outer error for general $\phi, \ell$ is stated as follows.

**Lemma E.21.** *Suppose that Assumptions 4.1, E.16, E.17 hold. Then with probability at least $1 - \delta$, there holds*

$$\sup_{\mathbf{w}\in\mathcal{W}} \left| \mathbb{E}\left[\widehat{\mathcal{L}}_{\mathrm{S}}^{\phi,\ell}(s_{\mathbf{w}})\right] - \widehat{\mathcal{L}}_{\mathrm{S}}^{\phi,\ell}(s_{\mathbf{w}}) \right| = \widetilde{O}\left( \frac{(GB+M)(\sqrt{\log(1/\delta)} + \sqrt{\sum_{l=1}^{L} d_l d_{l-1}})}{\sqrt{n}} \right).$$

*Proof.* We first give a uniform bound for $g_{\mathbf{w}}(\mathbf{x}, \mathbf{y}, \mathbf{y}_1', \cdots, \mathbf{y}_m')$ for $\mathbf{w} \in \mathcal{W}, \mathbf{x} \in \mathcal{X}, \mathbf{y}, \mathbf{y}_j' \in \mathcal{Y}$. Let

$$\mathbf{u} = (\ell(\Delta_{\mathbf{w}}(\mathbf{x}, \mathbf{y}, \mathbf{y}_1')), \cdots, \ell(\Delta_{\mathbf{w}}(\mathbf{x}, \mathbf{y}, \mathbf{y}_1'))), \mathbf{u}' = (\ell(0), \cdots, \ell(0)) \in \mathbb{R}^m.$$

Let function $q(\cdot)$ be defined in Lemma E.4. We have

$$q(\mathbf{u}') = \min_{\mu \in \mathbb{R}} \tau \phi((\ell(0) - \mu)/\tau) + \mu \geq \ell(0) - \mu + \mu = \ell(0)$$

and the minimum can be obtained at $\mu = \ell(0)$. Therefore, $q(\mathbf{u}') = \ell(0)$. Applying Lemma E.4 to $\mathbf{u}, \mathbf{u}'$ yields

$$
\begin{aligned}
|g_{\mathbf{w}}(\mathbf{x}, \mathbf{y}, \mathbf{y}_1', \cdots, \mathbf{y}_m')| = |q(\mathbf{u})| \leq & |q(\mathbf{u}) - q(\mathbf{u}')| + |q(\mathbf{u}')| \leq \|\mathbf{u} - \mathbf{u}'\|_\infty + |\ell(0)| \\
\leq & \sup_{\mathbf{x}\in\mathcal{X},\mathbf{y},\mathbf{y}'\in\mathcal{Y}} |\ell(\Delta_{\mathbf{w}}(\mathbf{x}, \mathbf{y}, \mathbf{y}'))| + |\ell(0)| \\
\leq & \sup_{\mathbf{x}\in\mathcal{X},\mathbf{y},\mathbf{y}'\in\mathcal{Y}} |\ell(\Delta_{\mathbf{w}}(\mathbf{x}, \mathbf{y}, \mathbf{y}')) - \ell(0)| + 2|\ell(0)| \\
\leq & G \sup_{\mathbf{x}\in\mathcal{X},\mathbf{y},\mathbf{y}'\in\mathcal{Y}} |\Delta_{\mathbf{w}}(\mathbf{x}, \mathbf{y}, \mathbf{y}')| + 2|\ell(0)| \leq 2GB + 2M,
\end{aligned}
$$

where we have used the $G$-Lipschitzness of $\ell$ in the last inequality. Then

$$B_{\mathcal{G}} := \sup_{\mathbf{w}} \|g_{\mathbf{w}}\|_{L_2(S)} \leq \max_i \sup_{\mathbf{w}} |g_{\mathbf{w}}(\mathbf{x}_i, \mathbf{y}_i, \mathbf{y}_{i1}', \cdots, \mathbf{y}_{im}')| \leq 2GB + 2M.$$

According to Proposition A.3 and Lemma A.5, there holds

$$
\begin{aligned}
\mathfrak{R}_S(\mathcal{G}) \leq & \inf_{\alpha>0} 4\alpha + \frac{12}{\sqrt{n}} \int_{\alpha}^{B_{\mathcal{G}}} \sqrt{\log \mathcal{N}(\mathcal{G}, \epsilon, L_2(S))} d\epsilon \leq \frac{4}{n} + \frac{12}{\sqrt{n}} \int_{\frac{1}{n}}^{2GB+2M} \sqrt{\log \mathcal{N}(\mathcal{G}, \epsilon, L_2(S))} d\epsilon \\
\leq & \frac{4}{n} + \frac{12}{\sqrt{n}} \int_{\frac{1}{n}}^{2GB+2M} \sqrt{\log \mathcal{N}(\mathcal{H}, \epsilon/G, L_\infty(S_1))} d\epsilon \leq \frac{4}{n} + \frac{12}{\sqrt{n}} \int_{\frac{1}{n}}^{2GB+2M} \sqrt{\sum_{l=1}^{L} d_l d_{l-1} \log\left(1 + \frac{8L\Pi_{l=1}^{L} s_l^2}{\epsilon}\right)} d\epsilon \\
\leq & \frac{4}{n} + \frac{12}{\sqrt{n}} \sqrt{\sum_{l=1}^{L} d_l d_{l-1} \log(1 + 8nL\Pi_{l=1}^{L} s_l^2)} (2GB + 2M - 1/n) = \widetilde{O}\left( \frac{(GB+M)\sqrt{\sum_{l=1}^{L} d_l d_{l-1}}}{n} \right).
\end{aligned}
$$

Using Proposition A.1, with probability at least $1 - \delta$, we have

$$
\begin{aligned}
& \sup_{\mathbf{w}\in\mathcal{W}} \left[ \mathbb{E}\left[\widehat{\mathcal{L}}_{\mathrm{S}}^{\phi,\ell}(s_{\mathbf{w}})\right] - \widehat{\mathcal{L}}_{\mathrm{S}}^{\phi,\ell}(s_{\mathbf{w}}) \right] \\
= & \sup_{\mathbf{w}\in\mathcal{W}} \left[ \mathbb{E}_{\mathbf{x},\mathbf{y},\mathbf{y}_1',\cdots,\mathbf{y}_m'}[g_{\mathbf{w}}(\mathbf{x}, \mathbf{y}, \mathbf{y}_1', \cdots, \mathbf{y}_m')] - \frac{1}{n} \sum_{i=1}^{n} g_{\mathbf{w}}(\mathbf{x}_i, \mathbf{y}_i, \mathbf{y}_{i1}', \cdots, \mathbf{y}_{im}') \right] \\
\leq & 2\mathfrak{R}_S(\mathcal{G}) + 12(GB+M)\sqrt{\frac{\log(2/\delta)}{2n}} = \widetilde{O}\left( \frac{(GB+M)(\sqrt{\log(1/\delta)} + \sqrt{\sum_{l=1}^{L} d_l d_{l-1}})}{\sqrt{n}} \right).
\end{aligned}
$$

The same upper bound also holds for the other side. The proof is completed. $\square$

E.4.2. INNER ERROR

In this part, we will estimate the inner error $|\mathcal{L}_S^{\phi,\ell}(s_\mathbf{w}) - \mathbb{E}[\widehat{\mathcal{L}}_S^{\phi,\ell}(s_\mathbf{w})]|$. We apply algorithmic stability methods to handle with strongly convex $\phi$, and uniform convergence theory to study general $\phi$.

Note that

$$
\begin{aligned}
&\mathcal{L}_S^{\phi,\ell}(s_\mathbf{w}) - \mathbb{E}\widehat{\mathcal{L}}_S^{\phi,\ell}(s_\mathbf{w}) \\
=&\mathbb{E}_{\mathbf{x},\mathbf{y}}\left\{ \left[ \min_{\mu\in\mathbb{R}} \mathbb{E}_{\mathbf{y}'} \tau\phi\left(\frac{\ell(\Delta_\mathbf{w}(\mathbf{x},\mathbf{y},\mathbf{y}')) - \mu}{\tau}\right) + \mu \right] - \mathbb{E}_{\mathbf{y}'_j}\left[ \min_{\mu\in\mathbb{R}} \frac{1}{m}\sum_{j=1}^m \tau\phi\left(\frac{\ell(\Delta_\mathbf{w}(\mathbf{x},\mathbf{y},\mathbf{y}'_j)) - \mu}{\tau}\right) + \mu \right] \right\}
\end{aligned}
\tag{69}
$$

According to Lemma E.6, for any $\mathbf{w},\mathbf{x},\mathbf{y}$, there holds

$$
\left[ \min_{\mu\in\mathbb{R}} \mathbb{E}_{\mathbf{y}'}\tau\phi\left(\frac{\ell(\Delta_\mathbf{w}(\mathbf{x},\mathbf{y},\mathbf{y}')) - \mu}{\tau}\right) + \mu \right] - \mathbb{E}_{\mathbf{y}'_j}\left[ \min_{\mu\in\mathbb{R}} \frac{1}{m}\sum_{j=1}^m \tau\phi\left(\frac{\ell(\Delta_\mathbf{w}(\mathbf{x},\mathbf{y},\mathbf{y}'_j)) - \mu}{\tau}\right) + \mu \right] \geq 0.
$$

Hence,

$$
\mathcal{L}_S^{\phi,\ell}(s_\mathbf{w}) - \mathbb{E}\widehat{\mathcal{L}}_S^{\phi,\ell}(s_\mathbf{w}) \geq 0.
\tag{70}
$$

**General $\phi$**  The inner error is similar to generalization error of risk-averse learning in Lee et al. (2020). However, they only considered one random variable, while we need to deal with triplet $(\mathbf{x},\mathbf{y},\mathbf{y}')$ and a pairwise loss $\ell$, which requires a more refined analysis.

For $\mathbf{x},\mathbf{y}$, according to Eq. (37), we have

$$
\mathcal{G}_{\mathbf{x},\mathbf{y}} := \{\mathbf{y}' \to \Delta_\mathbf{w}(\mathbf{x},\mathbf{y},\mathbf{y}'), \mathbf{w}\in\mathcal{W}\}, \quad S' = \{\mathbf{y}'_1,\cdots,\mathbf{y}'_m\}.
$$

The inner error for general disutility function $\phi$ is stated as follows.

**Lemma E.22.** *Let Assumptions 4.1, E.16, E.17 hold. Then there holds*

$$
\sup_{\mathbf{w}\in\mathcal{W}}\left| \mathcal{L}_S^{\phi,\ell}(s_\mathbf{w}) - \mathbb{E}\left[\widehat{\mathcal{L}}_S^{\phi,\ell}(s_\mathbf{w})\right]\right| = \widetilde{O}\left(\frac{\eta(GB\sqrt{\sum_{l=1}^L d_l d_{l-1}} + M)}{\sqrt{m}}\right).
$$

*Proof.* For $\mathbf{w},\mathbf{x},\mathbf{y}$, we define

$$
\hat\mu_{\mathbf{w},\mathbf{x},\mathbf{y}}(S') \in \arg\min_{\mu\in\mathbb{R}} \frac{1}{m}\sum_{j=1}^m \tau\phi\left(\frac{\ell(\Delta_\mathbf{w}(\mathbf{x},\mathbf{y},\mathbf{y}'_j)) - \mu}{\tau}\right) + \mu.
\tag{71}
$$

By Lemma E.5 and $\ell(\Delta_\mathbf{w}(\mathbf{x},\mathbf{y},\mathbf{y}'_j)) \in [-M,M]$, we can assume that $\hat\mu_{\mathbf{w},\mathbf{x},\mathbf{y}}(S') \in [-M,M]$. We further have

$$
\min_{\mu\in\mathbb{R}} \mathbb{E}_{\mathbf{y}'}\tau\phi\left(\frac{\ell(\Delta_\mathbf{w}(\mathbf{x},\mathbf{y},\mathbf{y}')) - \mu}{\tau}\right) + \mu \leq \mathbb{E}_{\mathbf{y}'}\tau\phi\left(\frac{\ell(\Delta_\mathbf{w}(\mathbf{x},\mathbf{y},\mathbf{y}')) - \hat\mu_{\mathbf{w},\mathbf{x},\mathbf{y}}(S')}{\tau}\right) + \hat\mu_{\mathbf{w},\mathbf{x},\mathbf{y}}(S').
\tag{72}
$$

Taking expectation over $S'$ yields

$$
\min_{\mu\in\mathbb{R}} \mathbb{E}_{\mathbf{y}'}\tau\phi\left(\frac{\ell(\Delta_\mathbf{w}(\mathbf{x},\mathbf{y},\mathbf{y}')) - \mu}{\tau}\right) + \mu \leq \mathbb{E}_{S'}\mathbb{E}_{\mathbf{y}'}\tau\phi\left(\frac{\ell(\Delta_\mathbf{w}(\mathbf{x},\mathbf{y},\mathbf{y}')) - \hat\mu_{\mathbf{w},\mathbf{x},\mathbf{y}}(S')}{\tau}\right) + \hat\mu_{\mathbf{w},\mathbf{x},\mathbf{y}}(S').
$$

Let

$$
\mathcal{H}_{\mathbf{x},\mathbf{y}} := \{\mathbf{y}' \mapsto \ell(\Delta_\mathbf{w}(\mathbf{x},\mathbf{y},\mathbf{y}')) - \mu : \mathbf{w}\in\mathcal{W}, \mu\in[-M,M]\}.
$$

Then we have

$$
\sup_{\mathbf{w}\in\mathcal{W}} \min_{\mu\in\mathbb{R}} \mathbb{E}_{\mathbf{y}'} \tau\phi\left(\frac{\ell(\Delta_{\mathbf{w}}(\mathbf{x},\mathbf{y},\mathbf{y}')) - \mu}{\tau}\right) + \mu - \mathbb{E}_{\{\mathbf{y}'_j\}}\left[\min_{\mu\in\mathbb{R}} \frac{1}{m}\sum_{j=1}^{m}\tau\phi\left(\frac{\ell(\Delta_{\mathbf{w}}(\mathbf{x},\mathbf{y},\mathbf{y}'_j)) - \mu}{\tau}\right) + \mu\right]
$$

$$
\leq \sup_{\mathbf{w}\in\mathcal{W}} \mathbb{E}_{S'}\mathbb{E}_{\mathbf{y}'} \tau\phi\left(\frac{\ell(\Delta_{\mathbf{w}}(\mathbf{x},\mathbf{y},\mathbf{y}')) - \hat{\mu}_{\mathbf{w},\mathbf{x},\mathbf{y}}(S')}{\tau}\right) + \hat{\mu}_{\mathbf{w},\mathbf{x},\mathbf{y}}(S')
$$

$$
- \left[\frac{1}{m}\sum_{j=1}^{m}\tau\phi\left(\frac{\ell(\Delta_{\mathbf{w}}(\mathbf{x},\mathbf{y},\mathbf{y}'_j)) - \hat{\mu}_{\mathbf{w},\mathbf{x},\mathbf{y}}(S')}{\tau}\right) + \hat{\mu}_{\mathbf{w},\mathbf{x},\mathbf{y}}(S')\right] \tag{73}
$$

$$
\leq \mathbb{E}_{S'} \sup_{\mathbf{w}\in\mathcal{W}} \mathbb{E}_{\mathbf{y}'} \tau\phi\left(\frac{\ell(\Delta_{\mathbf{w}}(\mathbf{x},\mathbf{y},\mathbf{y}')) - \hat{\mu}_{\mathbf{w},\mathbf{x},\mathbf{y}}(S')}{\tau}\right) - \left[\frac{1}{m}\sum_{j=1}^{m}\tau\phi\left(\frac{\ell(\Delta_{\mathbf{w}}(\mathbf{x},\mathbf{y},\mathbf{y}'_j)) - \hat{\mu}_{\mathbf{w},\mathbf{x},\mathbf{y}}(S')}{\tau}\right)\right]
$$

$$
\leq \mathbb{E}_{S'} \sup_{\mathbf{w}\in\mathcal{W},\mu\in[-M,M]} \mathbb{E}_{\mathbf{y}'} \tau\phi\left(\frac{\ell(\Delta_{\mathbf{w}}(\mathbf{x},\mathbf{y},\mathbf{y}')) - \mu}{\tau}\right) - \left[\frac{1}{m}\sum_{j=1}^{m}\tau\phi\left(\frac{\ell(\Delta_{\mathbf{w}}(\mathbf{x},\mathbf{y},\mathbf{y}'_j)) - \mu}{\tau}\right)\right]
$$

$$
\leq 2\eta\mathbb{E}_{S'}\mathfrak{R}_{S'}(\mathcal{H}_{\mathbf{x},\mathbf{y}}),
$$

where the second inequality results from the convexity of sup and Jenson's inequality, the last inequality is due to Proposition A.1, A.2. Now we derive an upper bound of $\mathfrak{R}_{S'}(\mathcal{H}_{\mathbf{x},\mathbf{y}})$. Note that

$$
\mathfrak{R}_{S'}(\mathcal{H}_{\mathbf{x},\mathbf{y}}) = \mathbb{E}_{\epsilon} \sup_{\mathbf{w}\in\mathcal{W},\mu\in[-M,M]} \frac{1}{m}\sum_{j=1}^{m}\epsilon_j(\ell(\Delta_{\mathbf{w}}(\mathbf{x},\mathbf{y},\mathbf{y}'_j)) - \mu)
$$

$$
\leq \mathbb{E}_{\epsilon} \sup_{\mathbf{w}\in\mathcal{W}} \frac{1}{m}\sum_{j=1}^{m}\epsilon_j\ell(\Delta_{\mathbf{w}}(\mathbf{x},\mathbf{y},\mathbf{y}'_j)) + \mathbb{E}_{\epsilon} \sup_{\mu\in[-M,M]} \frac{1}{m}\sum_{j=1}^{m}\epsilon_j\mu. \tag{74}
$$

We will control two terms respectively.

$$
\mathbb{E}_{\epsilon} \sup_{\mu\in[-M,M]} \frac{1}{m}\sum_{j=1}^{m}\epsilon_j\mu \leq M\mathbb{E}_{\epsilon}\left|\frac{1}{m}\sum_{j=1}^{m}\epsilon_j\right| \leq M\sqrt{\mathbb{E}_{\epsilon}\left|\frac{1}{m}\sum_{j=1}^{m}\epsilon_j\right|^2} = \frac{M}{\sqrt{m}}. \tag{75}
$$

Note that $\ell$ is $G$-Lipschitz. We apply Lemma A.2, Eq. (39) and derive

$$
\mathbb{E}_{\epsilon} \sup_{\mathbf{w}} \frac{1}{m}\sum_{j=1}^{m}\epsilon_j\ell(\Delta_{\mathbf{w}}(\mathbf{x},\mathbf{y},\mathbf{y}'_j)) \leq G\mathbb{E}_{\epsilon} \sup_{\mathbf{w}} \frac{1}{m}\sum_{j=1}^{m}\epsilon_j\Delta_{\mathbf{w}}(\mathbf{x},\mathbf{y},\mathbf{y}'_j) \leq \widetilde{O}\left(\frac{GB\sqrt{\sum_{l=1}^{L}d_ld_{l-1}}}{\sqrt{m}}\right). \tag{76}
$$

Plugging Eqs. (75), (76) into Eq. (74) yields

$$
\mathfrak{R}_{S'}(\mathcal{H}_{\mathbf{x},\mathbf{y}}) \leq \mathbb{E}_{\epsilon} \sup_{\mathbf{w}} \frac{1}{m}\sum_{j=1}^{m}\epsilon_j\ell(\Delta_{\mathbf{w}}(\mathbf{x},\mathbf{y},\mathbf{y}'_j)) + \mathbb{E}_{\epsilon} \sup_{\mu\in[-M,M]} \frac{1}{m}\sum_{j=1}^{m}\epsilon_j\mu \leq \widetilde{O}\left(\frac{GB\sqrt{\sum_{l=1}^{L}d_ld_{l-1}} + M}{\sqrt{m}}\right). \tag{77}
$$

Combining Eq. (73) and Eq. (77), we know that for all $\mathbf{x}, \mathbf{y}$, there holds

$$
\sup_{\mathbf{w}\in\mathcal{W}} \min_{\mu\in\mathbb{R}} \mathbb{E}_{\mathbf{y}'} \tau\phi\left(\frac{\ell(\Delta_{\mathbf{w}}(\mathbf{x},\mathbf{y},\mathbf{y}')) - \mu}{\tau}\right) + \mu - \mathbb{E}_{\{\mathbf{y}'_j\}}\left[\min_{\mu\in\mathbb{R}} \frac{1}{m}\sum_{j=1}^{m}\tau\phi\left(\frac{\ell(\Delta_{\mathbf{w}}(\mathbf{x},\mathbf{y},\mathbf{y}'_j)) - \mu}{\tau}\right) + \mu\right]
$$

$$
\leq \widetilde{O}\left(\frac{\eta(GB\sqrt{\sum_{l=1}^{L}d_ld_{l-1}} + M)}{\sqrt{m}}\right). \tag{78}
$$

By the convexity of $\sup$ and Jenson's inequality, we have

$$\sup_{\mathbf{w}\in\mathcal{W}} \mathcal{L}_S^{\phi,\ell}(s_{\mathbf{w}}) - \mathbb{E}\widehat{\mathcal{L}}_S^{\phi,\ell}(s_{\mathbf{w}})$$

$$= \sup_{\mathbf{w}\in\mathcal{W}} \mathbb{E}_{\mathbf{x},\mathbf{y}} \left\{ \left[ \min_{\mu\in\mathbb{R}} \mathbb{E}_{\mathbf{y}'} \tau\phi\left(\frac{\ell(\Delta_{\mathbf{w}}(\mathbf{x},\mathbf{y},\mathbf{y}')) - \mu}{\tau}\right) + \mu \right] - \mathbb{E}_{\mathbf{y}'_j} \left[ \min_{\mu\in\mathbb{R}} \frac{1}{m}\sum_{j=1}^m \tau\phi\left(\frac{\ell(\Delta_{\mathbf{w}}(\mathbf{x},\mathbf{y},\mathbf{y}'_j)) - \mu}{\tau}\right) + \mu \right] \right\}$$

$$\leq \mathbb{E}_{\mathbf{x},\mathbf{y}} \sup_{\mathbf{w}\in\mathcal{W}} \left\{ \min_{\mu\in\mathbb{R}} \mathbb{E}_{\mathbf{y}'} \tau\phi\left(\frac{\ell(\Delta_{\mathbf{w}}(\mathbf{x},\mathbf{y},\mathbf{y}')) - \mu}{\tau}\right) + \mu - \mathbb{E}_{\{\mathbf{y}'_j\}} \left[ \min_{\mu\in\mathbb{R}} \frac{1}{m}\sum_{j=1}^m \tau\phi\left(\frac{\ell(\Delta_{\mathbf{w}}(\mathbf{x},\mathbf{y},\mathbf{y}'_j)) - \mu}{\tau}\right) + \mu \right] \right\}$$

$$\leq \widetilde{O}\left(\frac{\eta(GB\sqrt{\sum_{l=1}^L d_l d_{l-1}} + M)}{\sqrt{m}}\right),$$

where the last inequality is due to Eq. (78). The proof is completed by noting Eq. (70). $\qquad\square$

**Strongly convex $\phi$**  The OCE risk contains an inner minimization problem. For strongly convex $\phi$, we could obtain a fine-grained analysis through algorithmic stability, similar to Lemma 4.3.

**Lemma E.23.** *Let Assumptions 4.1, E.16, E.17, E.18 hold. Then we have*

$$\sup_{\mathbf{w}} \mathcal{L}_S^{\phi,\ell}(\mathbf{w}) - \mathbb{E}\widehat{\mathcal{L}}_S^{\phi,\ell}(\mathbf{w}) \leq \frac{2\eta\tau}{\gamma m}.$$

*Proof.* We denote $\Psi_{\mathbf{w},\mathbf{x},\mathbf{y}}(\mu;\mathbf{y}') = \tau\phi\left(\frac{\ell(\Delta_{\mathbf{w}}(\mathbf{x},\mathbf{y},\mathbf{y}')) - \mu}{\tau}\right) + \mu, \mu \in [-M,M]$. Then $\Psi_{\mathbf{w},\mathbf{x},\mathbf{y}}(\mu;\mathbf{y}')$ is $\gamma/\tau$-strongly convex w.r.t. $\mu$. We also have

$$\frac{\partial\Psi_{\mathbf{w},\mathbf{x},\mathbf{y}}(\mu;\mathbf{y}')}{\partial\mu} = 1 - \partial\phi\left(\frac{\ell(\Delta_{\mathbf{w}}(\mathbf{x},\mathbf{y},\mathbf{y}')) - \mu}{\tau}\right) \in [1-\eta, 1].$$

By the property of $\phi$ in Definition E.1, we have $\eta \geq 1$. Therefore, for any $\mu_1, \mu_2 \in [-M,M]$,

$$|\Psi_{\mathbf{w},\mathbf{x},\mathbf{y}}(\mu_1;\mathbf{y}') - \Psi_{\mathbf{w},\mathbf{x},\mathbf{y}}(\mu_2;\mathbf{y}')| \leq \eta|\mu_1 - \mu_2|.$$

Let $\hat{\mu}(S')$ be defined in Eq. (71). Then for any $S'$,

$$\min_{\mu\in[-M,M]} \mathbb{E}_{\mathbf{y}'}\Psi_{\mathbf{w},\mathbf{x},\mathbf{y}}(\mu;\mathbf{y}') \leq \mathbb{E}_{\mathbf{y}'}\Psi_{\mathbf{w},\mathbf{x},\mathbf{y}}(\hat{\mu}(\{S'\});\mathbf{y}'),$$

therefore,

$$\min_{\mu\in[-M,M]} \mathbb{E}_{\mathbf{y}'}\Psi_{\mathbf{w},\mathbf{x},\mathbf{y}}(\mu;\mathbf{y}') \leq \mathbb{E}_{S'}\mathbb{E}_{\mathbf{y}'}\Psi_{\mathbf{w},\mathbf{x},\mathbf{y}}(\bar{\mu}(S');\mathbf{y}'). \tag{79}$$

By the $\gamma/\tau$-strong convexity of $\Psi_{\mathbf{w},\mathbf{x},\mathbf{y}}(\mu;\mathbf{y}')$, we can define a convex function

$$\hat{\Psi}_{\mathbf{w},\mathbf{x},\mathbf{y}}(\mu;\mathbf{y}') := \Psi_{\mathbf{w},\mathbf{x},\mathbf{y}}(\mu;\mathbf{y}') - \frac{\gamma}{2\tau}\mu^2.$$

Then

$$\hat{\mu}_{\mathbf{w},\mathbf{x},\mathbf{y}}(S') = \operatorname*{arg\,min}_{\mu\in[-M,M]} \frac{1}{m}\sum_{j=1}^m \hat{\Psi}_{\mathbf{w},\mathbf{x},\mathbf{y}}(\mu;\mathbf{y}'_j) + \frac{\gamma}{2\tau}\mu^2,$$

which is the regularized empirical risk minimization algorithm. According to Theorem 12, 22 in Bousquet & Elisseeff (2002), we have

$$\mathbb{E}_{S'}\left[\mathbb{E}_{\mathbf{y}'}\Psi_{\mathbf{w},\mathbf{x},\mathbf{y}}(\hat{\mu}_{\mathbf{w},\mathbf{x},\mathbf{y}}(S');\mathbf{y}') - \frac{1}{m}\sum_{j=1}^m \Psi_{\mathbf{w},\mathbf{x},\mathbf{y}}(\hat{\mu}_{\mathbf{w},\mathbf{x},\mathbf{y}}(S');\mathbf{y}'_j)\right] \leq \frac{2\eta\tau}{\gamma m}$$

holds uniformly for $\mathbf{x}, \mathbf{y}, \mathbf{w}$. Combined with Eq. (79), for any $\mathbf{x}, \mathbf{y}, \mathbf{w}$, there holds

$$\min_{\mu \in [M,M]} \mathbb{E}_{\mathbf{y}'} \Psi_{\mathbf{w},\mathbf{x},\mathbf{y}}(\mu; \mathbf{y}') - \mathbb{E}_{S'} \frac{1}{m} \sum_{j=1}^{m} \Psi_{\mathbf{w},\mathbf{x},\mathbf{y}}(\bar{\mu}_{\mathbf{w},\mathbf{x},\mathbf{y}}(S'); \mathbf{y}'_j)$$

$$\leq \mathbb{E}_{\{\mathbf{y}'_j\}} \left[ \mathbb{E}_{\mathbf{y}} \Psi_{\mathbf{w},\mathbf{x},\mathbf{y}}(\hat{\mu}_{\mathbf{w},\mathbf{x},\mathbf{y}}(S'); \mathbf{y}') - \frac{1}{m} \sum_{j=1}^{m} \Psi_{\mathbf{w},\mathbf{x},\mathbf{y}}(\hat{\mu}_{\mathbf{w},\mathbf{x},\mathbf{y}}(S'); \mathbf{y}'_j) \right] \leq \frac{2\eta\tau}{\gamma m}.$$

As a result,

$$\sup_{\mathbf{w}} \left[ \mathcal{L}^{\phi,\ell}(\mathbf{w}) - \mathbb{E}\widehat{\mathcal{L}}_S^{\phi,\ell}(\mathbf{w}) \right] = \sup_{\mathbf{w}} \mathbb{E}_{\mathbf{x},\mathbf{y}} \left[ \min_{\mu \in [-M,M]} \mathbb{E}_{\mathbf{y}'} \Psi_{\mathbf{w},\mathbf{x},\mathbf{y}}(\mu; \mathbf{y}') - \mathbb{E}_{\mathbf{y}'_j} \min_{\mu \in [-M,M]} \frac{1}{m} \sum_{j=1}^{m} \Psi_{\mathbf{w},\mathbf{x},\mathbf{y}}(\mu; \mathbf{y}'_j) \right] \leq \frac{2\eta\tau}{\gamma m}.$$

The proof is completed. $\qquad \square$

Combining Lemma E.21, E.22, E.23 with Eq. (68) establishes the generalization error for $\mathcal{L}_S^{\phi,\ell}(\mathbf{w})$.

**Theorem E.24.** *Let Assumptions 4.1, E.16, E.17 hold, with probability at least $1 - \delta$, we have*

$$\sup_{\mathbf{w} \in \mathcal{W}} \left| \mathcal{L}_S^{\phi,\ell}(s_{\mathbf{w}}) - \widehat{\mathcal{L}}_S^{\phi,\ell}(s_{\mathbf{w}}) \right| = \widetilde{O}\left( \eta(GB + M)(\sqrt{\log(1/\delta)} + \sqrt{\sum_{l=1}^{L} d_l d_{l-1}})(\frac{1}{\sqrt{n}} + \frac{1}{\sqrt{m}}) \right).$$

*If Assumption E.18 holds, we obtain a tighter bound*

$$\sup_{\mathbf{w} \in \mathcal{W}} \mathcal{L}_S^{\phi,\ell}(s_{\mathbf{w}}) - \hat{\mathcal{L}}_S^{\phi,\ell}(s_{\mathbf{w}}) = \widetilde{O}\left( \frac{(GB + M)(\sqrt{\log(1/\delta)} + \sum_{l=1}^{L} \sqrt{d_l d_{l-1}})}{\sqrt{n}} + \frac{2\eta\tau}{\gamma m} \right).$$

The above theorem indicates that strongly convex disutility function $\phi$ admits a better generalization performance. A special case is $\phi(t) = \exp(t) - 1$, corresponding to standard contrastive loss $\mathcal{L}$. This shows the advantage of the log-sum-exp structure.

### E.5. Generalization analysis of SSCRL for general OCE and pairwise loss

In this subsection, we turn to the generalization analysis of SSCRL for general OCE and pairwise loss Recall that

$$L_{SS}^{\phi,\ell}(s_{\mathbf{w}}) = \mathbb{E}_{\mathbf{x},\mathbf{y}} \left[ \min_{\mu \in \mathbb{R}} \mathbb{E}_{\mathbf{y}'} \tau\phi\left( \frac{\ell(\Delta_{\mathbf{w}}(\mathbf{x}, \mathbf{y}, \mathbf{y}')) - \mu}{\tau} \right) + \mu \right],$$

$$\widehat{\mathcal{L}}_{SS}^{\phi,\ell}(s_{\mathbf{w}}) = \frac{1}{n} \left[ \sum_{i=1}^{n} \min_{\mu_i} \frac{\tau}{m} \sum_{j=1}^{m} \phi\left( \frac{\ell(\Delta_{\mathbf{w}}(\mathbf{x}_i, \mathbf{y}_i, \mathbf{y}'_j)) - \mu_i}{\tau} \right) + \mu_i \right].$$

We aim to bound the uniform convergence

$$\sup_{\mathbf{w} \in \mathcal{W}} \left| \mathcal{L}_{SS}^{\phi,\ell}(s_{\mathbf{w}}) - \widehat{\mathcal{L}}_{SS}^{\phi,\ell}(s_{\mathbf{w}}) \right|.$$

Similar to the analysis in Section D, we make the following error decomposition:

$$|\mathcal{L}_{SS}^{\phi,\ell}(s_{\mathbf{w}}) - \widehat{\mathcal{L}}_{SS}^{\phi,\ell}(s_{\mathbf{w}})|$$

$$\leq \underbrace{\left| \mathbb{E}_{\mathbf{x},\mathbf{y}} H_{\mathbf{w}}(\mathbf{x}, \mathbf{y}) - \frac{1}{n} \sum_{i=1}^{n} H_{\mathbf{w}}(\mathbf{x}_i, \mathbf{y}_i) \right|}_{\text{outer error}} + \underbrace{\left| \frac{1}{n} \sum_{i=1}^{n} \left( H_{\mathbf{w}}(\mathbf{x}_i, \mathbf{y}_i) - \widehat{H}_{\mathbf{w}}(\mathbf{x}_i, \mathbf{y}_i) \right) \right|}_{\text{inner error}}, \qquad (80)$$

where we define

$$H_{\mathbf{w}}(\mathbf{x}, \mathbf{y}) = \min_{\mu \in \mathbb{R}} \mathbb{E}_{\mathbf{y}'} \tau \phi \left( \frac{\ell(\Delta_{\mathbf{w}}(\mathbf{x}, \mathbf{y}, \mathbf{y}')) - \mu}{\tau} \right) + \mu,$$

$$\widehat{H}_{\mathbf{w}}(\mathbf{x}, \mathbf{y}) = \min_{\mu \in \mathbb{R}} \frac{\tau}{m} \sum_{j=1}^{m} \phi \left( \frac{\ell(\Delta_{\mathbf{w}}(\mathbf{x}, \mathbf{y}, \mathbf{y}_j')) - \mu}{\tau} \right) + \mu.$$

We will analyze inner error and outer error respectively.

### E.5.1. ESTIMATION OF THE OUTER ERROR

Recall that $S_4 = \{(\mathbf{x}_1, \mathbf{y}_1), \cdots, (\mathbf{x}_n, \mathbf{y}_n)\}$, we define the function class

$$\mathcal{H}_{\mathbf{w}} = \{(\mathbf{x}, \mathbf{y}) \mapsto H_{\mathbf{w}}(\mathbf{x}, \mathbf{y}), \mathbf{w} \in \mathcal{W}\}.$$

We establish the following bound for the integral of the covering number $\mathcal{N}(\mathcal{H}_{\mathbf{w}}, \epsilon, L_2(S_4))$.

**Lemma E.25.** *Let Assumption 4.1 hold, we have*

$$\int_{1/n}^{\eta M} \sqrt{\log \mathcal{N}(\mathcal{H}_{\mathbf{w}}, \epsilon, L_2(S_4))} d\epsilon \leq \widetilde{O}\left( \eta M \sqrt{\sum_{l=1}^{L} d_l d_{l-1}} \right).$$

*Proof.* we define

$$\mu_1 \in \underset{\mu \in [-M, M]}{\arg\min} \ \mathbb{E}_{\mathbf{y}'} \tau \phi \left( \frac{\ell(\Delta_{\mathbf{w}}(\mathbf{x}, \mathbf{y}, \mathbf{y}')) - \mu}{\tau} \right) + \mu, \mu_2 \in \underset{\mu \in [-M, M]}{\arg\min} \ \mathbb{E}_{\mathbf{y}'} \tau \phi \left( \frac{\ell(\Delta_{\mathbf{w}'}(\mathbf{x}, \mathbf{y}, \mathbf{y}')) - \mu}{\tau} \right) + \mu.$$

We suppose that $H_{\mathbf{w}}(\mathbf{x}, \mathbf{y}) - H_{\mathbf{w}'}(\mathbf{x}, \mathbf{y}) \geq 0$, we have

$$|H_{\mathbf{w}}(\mathbf{x}, \mathbf{y}) - H_{\mathbf{w}'}(\mathbf{x}, \mathbf{y})| = H_{\mathbf{w}}(\mathbf{x}, \mathbf{y}) - H_{\mathbf{w}'}(\mathbf{x}, \mathbf{y})$$

$$= \mathbb{E}_{\mathbf{y}'} \tau \phi \left( \frac{\ell(\Delta_{\mathbf{w}}(\mathbf{x}, \mathbf{y}, \mathbf{y}')) - \mu_1}{\tau} \right) + \mu_1 - \mathbb{E}_{\mathbf{y}'} \tau \phi \left( \frac{\ell(\Delta_{\mathbf{w}'}(\mathbf{x}, \mathbf{y}, \mathbf{y}')) - \mu_2}{\tau} \right) - \mu_2$$

$$\leq \mathbb{E}_{\mathbf{y}'} \tau \phi \left( \frac{\ell(\Delta_{\mathbf{w}}(\mathbf{x}, \mathbf{y}, \mathbf{y}')) - \mu_2}{\tau} \right) - \mathbb{E}_{\mathbf{y}'} \tau \phi \left( \frac{\ell(\Delta_{\mathbf{w}'}(\mathbf{x}, \mathbf{y}, \mathbf{y}')) - \mu_2}{\tau} \right)$$

$$\leq \mathbb{E}_{\mathbf{y}'} \tau \left| \phi \left( \frac{\ell(\Delta_{\mathbf{w}}(\mathbf{x}, \mathbf{y}, \mathbf{y}')) - \mu_2}{\tau} \right) - \phi \left( \frac{\ell(\Delta_{\mathbf{w}'}(\mathbf{x}, \mathbf{y}, \mathbf{y}')) - \mu_2}{\tau} \right) \right|$$

$$\leq \mathbb{E}_{\mathbf{y}'} \eta |\ell(\Delta_{\mathbf{w}}(\mathbf{x}, \mathbf{y}, \mathbf{y}')) - \ell(\Delta_{\mathbf{w}'}(\mathbf{x}, \mathbf{y}, \mathbf{y}'))|$$

$$\leq \mathbb{E}_{\mathbf{y}'} \eta G |\Delta_{\mathbf{w}}(\mathbf{x}, \mathbf{y}, \mathbf{y}') - \Delta_{\mathbf{w}'}(\mathbf{x}, \mathbf{y}, \mathbf{y}')|,$$

where the first inequality is by the definition of $\mu_1$, the second and third inequalities use the Lipschitzness of $\phi, \ell$, respectively.

We choose a cover $\mathcal{C}_\varepsilon$ of $\mathcal{W}$ such that Eq. (17) holds. By Eq. 21, we know that

$$|H_{\mathbf{w}}(\mathbf{x}, \mathbf{y}) - H_{\mathbf{w}'}(\mathbf{x}, \mathbf{y})| \leq \eta G \mathbb{E}_{\mathbf{y}'} |\Delta_{\mathbf{w}}(\mathbf{x}, \mathbf{y}, \mathbf{y}') - \Delta_{\mathbf{w}'}(\mathbf{x}, \mathbf{y}, \mathbf{y}')|$$

$$\leq 4\eta G \Pi_{l=1}^{L} s_l \mathbb{E}_{\mathbf{y}'} \max_{\mathbf{v} \in \{\mathbf{x}, \mathbf{y}, \mathbf{y}'\}} \|f_{\mathbf{w}}(\mathbf{v}) - f_{\mathbf{w}'}(\mathbf{v})\|_2$$

$$\leq 4\eta G \Pi_{l=1}^{L} s_l \max_{\mathbf{v} \in \mathcal{X}, \mathcal{Y}} \|f_{\mathbf{w}}(\mathbf{v}) - f_{\mathbf{w}'}(\mathbf{v})\|_2 \leq 4\eta G \Pi_{l=1}^{L} s_l \varepsilon.$$

As a result,

$$\|h_{\mathbf{w}} - h_{\mathbf{w}'}\|_{L_2(S_4)} \leq 4\eta G \Pi_{l=1}^{L} s_l \varepsilon$$

For $\epsilon > 0$, choosing

$$\varepsilon = \frac{\epsilon}{4\eta G \Pi_{l=1}^{L} s_l}.$$

Then we have

$$\log \mathcal{N}(\mathcal{H}_{\mathbf{w}}, \epsilon, L_2(S_4)) \le \log |\mathcal{C}_{\varepsilon}| \le \sum_{l=1}^{L} d_l d_{l-1} \log\left(1 + \frac{2L\Pi_{l=1}^{L} s_l}{\varepsilon}\right) = \sum_{l=1}^{L} d_l d_{l-1} \log\left(1 + \frac{8\eta GL\Pi_{l=1}^{L} s_l^2}{\epsilon}\right).$$

It implies that

$$\int_{1/n}^{\eta M} \sqrt{\log \mathcal{N}(\mathcal{H}_{\mathbf{w}}, \epsilon, L_2(S_4))} d\epsilon \le \int_{1/n}^{\eta M} \sqrt{\sum_{l=1}^{L} d_l d_{l-1} \log\left(1 + \frac{8\eta GL\Pi_{l=1}^{L} s_l^2}{\epsilon}\right)} d\epsilon$$

$$\le \sqrt{\sum_{l=1}^{L} d_l d_{l-1} \log(1 + 8\eta GnL\Pi_{l=1}^{L} s_l^2)(\eta M - 1/n)}$$

$$\le \eta M \log(1 + 8\eta GnL\Pi_{l=1}^{L} s_l^2) \sqrt{\sum_{l=1}^{L} d_l d_{l-1}}.$$

The proof is completed. $\qquad\square$

The outer error can be estimated as follows

**Lemma E.26.** *Let Assumption 4.1, then with probability at least $1 - \delta$, there holds*

$$\sup_{\mathbf{w} \in \mathcal{W}} \left| \mathbb{E}_{\mathbf{x},\mathbf{y}} H_{\mathbf{w}}(\mathbf{x}, \mathbf{y}) - \frac{1}{n}\sum_{i=1}^{n} H_{\mathbf{w}}(\mathbf{x}_i, \mathbf{y}_i) \right| \le \widetilde{O}\left(\frac{\eta M(\sqrt{\sum_{l=1}^{L} d_l d_{l-1}} + \sqrt{\log(1/\delta)})}{\sqrt{n}}\right)$$

*Proof.* Since $\phi(0) = 0, \phi(x) \ge x$, we have

$$H_{\mathbf{w}}(\mathbf{x}, \mathbf{y}) = \min_{\mu \in \mathbb{R}} \mathbb{E}_{\mathbf{y}'} \tau \phi\left(\frac{\ell(\Delta_{\mathbf{w}}(\mathbf{x}, \mathbf{y}, \mathbf{y}')) - \mu}{\tau}\right) + \mu \le \mathbb{E}_{\mathbf{y}'} \tau \phi\left(\frac{\ell(\Delta_{\mathbf{w}}(\mathbf{x}, \mathbf{y}, \mathbf{y}')) - 0}{\tau}\right) + 0$$

$$\le \mathbb{E}_{\mathbf{y}'} \tau \left| \phi\left(\frac{\ell(\Delta_{\mathbf{w}}(\mathbf{x}, \mathbf{y}, \mathbf{y}'))}{\tau}\right) - 0 \right| \le \mathbb{E}_{\mathbf{y}'} \eta |\ell(\Delta_{\mathbf{w}}(\mathbf{x}, \mathbf{y}, \mathbf{y}'))| \le \eta M.$$

and

$$H_{\mathbf{w}}(\mathbf{x}, \mathbf{y}) = \min_{\mu \in \mathbb{R}} \mathbb{E}_{\mathbf{y}'} \tau \phi\left(\frac{\ell(\Delta_{\mathbf{w}}(\mathbf{x}, \mathbf{y}, \mathbf{y}')) - \mu}{\tau}\right) + \mu \ge \mathbb{E}_{\mathbf{y}'} \ell(\Delta_{\mathbf{w}}(\mathbf{x}, \mathbf{y}, \mathbf{y}')) \ge -M.$$

By noting that $\eta \ge 1$, we have $|H_{\mathbf{w}}(\mathbf{x}, \mathbf{y})| \le \eta M$. For the similar reason, there also holds $\widehat{H}_{\mathbf{w}}(\mathbf{x}, \mathbf{y})| \le \eta M$ Then by Proposition A.3 and Lemma E.25, we have

$$\mathfrak{R}_{S_4}(\mathcal{H}_{\mathbf{w}}) \le \inf_{\alpha > 0} 4\alpha + \frac{12}{\sqrt{n}} \int_{\alpha}^{\eta M} \sqrt{\log \mathcal{N}(\mathcal{H}_{\mathbf{w}}, \epsilon, L_2(S_4))} d\epsilon \le \frac{4}{n} + \frac{12}{\sqrt{n}} \int_{1/n}^{\eta M} \sqrt{\log \mathcal{N}(\mathcal{H}_{\mathbf{w}}, \epsilon, L_2(S_4))} d\epsilon$$

$$= \widetilde{O}\left(\frac{\eta M \sqrt{\sum_{l=1}^{L} d_l d_{l-1}}}{\sqrt{n}}\right).$$

Combined with Proposition A.1, we have with probability at least $1 - \delta$,

$$\sup_{\mathbf{w} \in \mathcal{W}} \mathbb{E}_{\mathbf{x},\mathbf{y}} H_{\mathbf{w}}(\mathbf{x}, \mathbf{y}) - \frac{1}{n}\sum_{i=1}^{n} H_{\mathbf{w}}(\mathbf{x}_i, \mathbf{y}_i) \le 2\mathfrak{R}_{S_4}(\mathcal{H}_{\mathbf{w}}) + 6\eta M \sqrt{\frac{\log(1/\delta)}{n}} \tag{81}$$

$$\le \widetilde{O}\left(\frac{\eta M(\sqrt{\sum_{l=1}^{L} d_l d_{l-1}} + \sqrt{\log(1/\delta)})}{\sqrt{n})}\right). \tag{82}$$

Similar inequality holds for another direction. The proof is completed. $\qquad\square$

E.5.2. ESTIMATION OF THE INNER ERROR

**Lemma E.27.** *Let Assumption 4.1 hold, then with probability at least* $1 - \delta$*, we have*

$$\sup_{\mathbf{w} \in \mathcal{W}} \left| \frac{1}{n} \sum_{i=1}^{n} (H_{\mathbf{w}}(\mathbf{x}_i, \mathbf{y}_i) - \widehat{H}_{\mathbf{w}}(\mathbf{x}_i, \mathbf{y}_i)) \right| = \widetilde{O}\left( \frac{\eta(GB\sqrt{\sum_{l=1}^{L} d_l d_{l-1}} + M\sqrt{\log(1/\delta)})}{\sqrt{m}} \right).$$

*Proof.* Recall that

$$\mathcal{G}_{\mathbf{x},\mathbf{y}} = \{\mathbf{y}' \to \Delta_{\mathbf{w}}(\mathbf{x}, \mathbf{y}, \mathbf{y}'), \mathbf{w} \in \mathcal{W}\}, \quad S' = \{\mathbf{y}'_1, \cdots, \mathbf{y}'_m\},$$
$$\mathcal{H}_{\mathbf{x},\mathbf{y}} = \{\mathbf{y}' \mapsto \ell(\Delta_{\mathbf{w}}(\mathbf{x}, \mathbf{y}, \mathbf{y}')) - \mu : \mathbf{w} \in \mathcal{W}, \mu \in [-M, M]\}.$$

By Proposition A.1 and Eqs. (73), (77), for any fixed $(\mathbf{x}, \mathbf{y})$, with probability at least $1 - \delta$ over the sampling of $\mathbf{y}'_1, \cdots, \mathbf{y}'_m$, there holds

$$H_{\mathbf{w}}(\mathbf{x}, \mathbf{y}) - \widehat{H}_{\mathbf{w}}(\mathbf{x}, \mathbf{y}) \leq 2\eta \mathfrak{R}_{S'}(\mathcal{H}_{\mathbf{x},\mathbf{y}}) + 12\eta M \sqrt{\frac{\log(1/\delta)}{m}}$$
$$= \widetilde{O}\left( \frac{\eta(GB\sqrt{\sum_{l=1}^{L} d_l d_{l-1}} + M\sqrt{\log(2/\delta)})}{\sqrt{2m}} \right)$$

By the independence of $S'$ and $\mathbf{x}_i, \mathbf{y}_i$, with probability at least $1 - \delta/n$, the following holds for all $i \in [n]$:

$$\sup_{\mathbf{w} \in \mathcal{W}} H_{\mathbf{w}}(\mathbf{x}_i, \mathbf{y}_i) - \widehat{H}_{\mathbf{w}}(\mathbf{x}_i, \mathbf{y}_i) = \widetilde{O}\left( \frac{\eta(GB\sqrt{\sum_{l=1}^{L} d_l d_{l-1}} + M\sqrt{\log(1/\delta)})}{\sqrt{m}} \right).$$

Taking summation yields

$$\sup_{\mathbf{w} \in \mathcal{W}} \frac{1}{n} \sum_{i=1}^{n} (H_{\mathbf{w}}(\mathbf{x}_i, \mathbf{y}_i) - \widehat{H}_{\mathbf{w}}(\mathbf{x}_i, \mathbf{y}_i)) = \widetilde{O}\left( \frac{\eta(GB\sqrt{\sum_{l=1}^{L} d_l d_{l-1}} + M\sqrt{\log(n/\delta)})}{\sqrt{m}} \right).$$

The other side of the inequality holds for the same reason. The proof of the lemma is completed.

$\square$

The generalization bound is stated in the following theorem.

**Theorem E.28.** *Suppose Assumption 4.1, E.16, E.17 hold. Then with probability at least* $1 - \delta$*, there holds*

$$\sup_{\mathbf{w} \in \mathcal{W}} \left| \mathcal{L}_{\mathrm{SS}}^{\phi,\ell}(s_{\mathbf{w}}) - \widehat{\mathcal{L}}_{\mathrm{SS}}^{\phi,\ell}(s_{\mathbf{w}}) \right| \leq \widetilde{O}\left( \eta(GB + M)\left( \sqrt{\sum_{l=1}^{L} d_l d_{l-1}} + \sqrt{\log(n/\delta)}\right)\left(\frac{1}{\sqrt{m}} + \frac{1}{\sqrt{n}}\right) \right)$$

*Proof.* Plugging Lemma E.26, E.27 into the error decomposition Eq. (80), with probability at least $1 - \delta$, there holds

$$\sup_{\mathbf{w} \in \mathcal{W}} |\mathcal{L}_{\mathrm{SS}}(s_{\mathbf{w}}) - \widehat{\mathcal{L}}_{\mathrm{SS}}(s_{\mathbf{w}})| \leq \left| \mathbb{E}_{\mathbf{x},\mathbf{y}} h_{\mathbf{w}}(\mathbf{x}, \mathbf{y}) - \frac{1}{n} \sum_{i=1}^{n} h_{\mathbf{w}}(\mathbf{x}_i, \mathbf{y}_i) \right| + \left| \frac{1}{n} \sum_{i=1}^{n} (h_{\mathbf{w}}(\mathbf{x}_i, \mathbf{y}_i) - \hat{h}_{\mathbf{w}}(\mathbf{x}_i, \mathbf{y}_i)) \right|$$
$$\leq \widetilde{O}\left( \frac{\eta M(\sqrt{\sum_{l=1}^{L} d_l d_{l-1}} + \sqrt{\log(1/\delta)})}{\sqrt{n}} + \frac{\eta(GB\sqrt{\sum_{l=1}^{L} d_l d_{l-1}} + M\sqrt{\log(n/\delta)})}{\sqrt{m}} \right)$$
$$\leq \widetilde{O}\left( \eta(GB + M)\left( \sqrt{\sum_{l=1}^{L} d_l d_{l-1}} + \sqrt{\log(n/\delta)}\right)\left(\frac{1}{\sqrt{m}} + \frac{1}{\sqrt{n}}\right) \right).$$

The proof is completed.

$\square$

