# OpenReview forum: "Statistical Consistency and Generalization of Contrastive Representation Learning"
_ICML.cc/2026/Conference — ICML 2026 regular_

### Official Review · Reviewer_mXAj · 2026-03-09

**Soundness:** 3
**Presentation:** 2
**Significance:** 3
**Originality:** 3
**Overall Recommendation:** 3
**Confidence:** 3

**Summary:**

The paper studies the statistical consistency of contrastive learning objectives such as InfoNCE. Specifically, it asks whether minimizing the contrastive loss guarantees improved performance on a downstream retrieval task. The authors develop a theoretical framework linking the contrastive objective to a retrieval ranking objective and derive generalization bounds depending on the number of samples 𝑛
and the number of negatives 𝑚.

**Compliance With Llm Reviewing Policy:**

Affirmed.

**Final Justification:**

Im a bit torn about this paper, the topic is important, and the derivations has some interesting insights - something all reviewers may agree on. Having said this, I think it is highly difficult to follow the paper mathematically. Moreover, I deeply do not agree with the fact that pure learning theory do not have to be accompanied with an empirical evidences. For example, the amount of negatives is very relevant. In general in order to develop the mathematics we have a decent amount of assumptions, some are stated and some are hidden. Showing it in the "real life", is strengthen the assumptions validity, to make sure anything isn't hidden somewhere beneath. All reviewers including me agree that the work is valuable, my main concern is regarding the clarity and fluently of the manuscript, so - Im keeping my current score and let the AC decide in this case.

**Key Questions For Authors:**

* Is the stylish I is indicator function, in Eq. 2? Worth to mention formally. Moreover, why do you need to measure specifically the state where the inner product of the positive is equal to the negative? Isnt it extremely rare for a high-dimensional vectors? (Or am I missing something here?)

* You defined the L function on a single score (in Eq. 1), and it is clear. Then you used it in Eq. 3 for a sequence of scores, what does it means? it is not defined anywhere..

* In the problem definition you set: "Denote I be the indicator function such that an event E holds true if I[E]=0.". What? I think that indicator should count positives, namely I(true) = 1. Very confusing.


In general, theoretical understanding of the validity of CRL, especially when it drives applicative insights is extremely important. While the theoretical direction is interesting, the limited empirical validation and clarity issues make it difficult to fully assess the practical impact of the results, especially when the experiments conducted on regimes which are far away from those made in practice.

**Limitations:**

The empirical evidences is too shallow to validate the theoretical claims, moreover the working point (in terms of $\frac{m}{n}$) is way off from the trivial ones as currently in CLIP for example. Moreover the paper focuses on validating his claim with no discussion on what can be done based on this new understanding (there is a short discussion on the conclusion section, but mainly on additional theoretical analysis, rather than applicative gains)

**Strengths And Weaknesses:**

Strengths:

* Theoretical analysis of contrastive learning, particularly regarding statistical consistency, remains relatively limited compared to its widespread empirical success. Namely stating how the upstream general training is related to a downstream tasks.

* Analyze CRL differently, i.e, with respect to a retrieval task, is interesting.

Weaknesses:

* The introduction is difficult to follow because it mixes motivation, definitions, and technical details. The main research question only becomes clear after careful reading. A clearer early high-level explanation of the problem setting and the main contributions (shortly) would improve readability. It is also duplicated with section 2. (what is the difference between Eq. 1 and E1. 8)?

* The empirical evaluation is limited to CLIP-style vision–language retrieval. It would strengthen the paper to include additional contrastive frameworks such as SimCLR, where both views come from the same modality. Moreover, the results average MSCOCO and Flickr performance. Reporting them separately would provide a clearer picture of model behavior across datasets. Additionally,  the theoretical analysis focuses on regimes where the number of negatives 𝑚 is comparable to the number of samples 𝑛. However, in practical systems such as CLIP, the number of negatives per batch is extremely small relative to the dataset size. This raises concerns whether the analyzed regime reflects real training conditions.

Minor Weaknesses:

* I guess that the minus in equations 1 and 2 is coming from the negative group of x, however at first glance it seems like the whole expression is with minus, I would try to find a better notation to eliminate this possible confusion.

* Assumption 3.4 appears to hold trivially for any finite dataset. Since real-world datasets are finite, it is unclear what meaningful restriction this assumption imposes.

* The notations become heavy in the middle of the paper, which make it very hard to follow, I suggest to move the heavy proofs to the supp, and leave the main paper for the theorems and definitions leaving enough space for a more intuitive explanations, maybe through examples or figures (The only figure in the paper is the experimental graphs), you could think of a more intuitive ways to convey the idea through illustrations IMO.

* The paper could benefit from discussing the relation of CRL to Gaussianity as emerged for example from [1,2] lately. It seems related to me.

refs:

[1] InfoNCE Induces Gaussian Distribution. Betser et al. Arxiv.

[2] LeJEPA:Provable and Scalable Self-Supervised Learning Without the Heuristics. Balestriero & LeCunn. Arxiv.

---

> ### Author Rebuttal · Authors · 2026-03-31
>
> Thanks for acknowledging the strengths of our work. Our point-to-point response to your comments follows next.
> >**Q1.** A clearer early high-level explanation of the problem setting and the main contributions (shortly) would improve readability. The notations become heavy in the middle of the paper.
>
> **A:** We thank the reviewer for these constructive suggestions. We will present our work more clearyly in the revision.
> >**Q2.** What is the difference between Eq. 1 and Eq. 8?
>
> **A:** Eq. (8) is an instantiation of Eq. (1) when $p_x^+,p_x^-$ are defined in Eq. (7).
>
> >**Q3.** The results average MSCOCO and Flickr performance.
>
> **A:** We plot the curves of different datasets in this [anonymous link](https://github.com/icml2026consistency/icml2026consistency/blob/main/mscoco_flickr_separate.pdf) (commit e04d673), where we observe the same trend as in the paper.
>
> >**Q4.** The theoretical analysis focuses on regimes where $m$ is comparable to $n$. In practical systems such as CLIP, the number of negatives per batch is extremely small relative to the dataset size.
>
> **A:** There is no contradiction:
> - Our theory studies how the number of negatives in the definition of empirical CRL objective impacts the generalization;
> - The use of a small number of negative data in practice is for optimization purpose. Please note that for a CRL objective such as Eqn. (6) or (10) with a large $m$, we can use advanced optimization algorithms (e.g., FastCLIP [3]) that only uses a small mini-batch yet enjoy convergence guarantee.
> - For approaches that are based on mini-batch InfoNCE losses, existing stuides such as SimCLR have already shown that increasing the number of negatives leads to  better performance.
>
> >**Q5.** The empirical evidences is insufficient. Moreover the paper focuses on validating his claim with no discussion on what can be done based on this new understanding.
>
> **A:** We would like to clarify that **this is a paper on learning thoery.** While we believe exploring more experiments is an interesting direction, we would like to emphasize that this is **not an emprical-oriented paper**. We have intensive efforts on the statistical consistency in Sec. 3 and an improved generalization analysis in Sec. 4, which are both under-explored in the existing literature.
>
> In fact, our empirical study has brought a new insights on how the optimal number of negatives $m^*=n^{0.94}$ depends on the data size $n$. This could inform the design of new optimization algortihms in practice. For example, we do not need to use all data to define the negative set as in the FastCLIP paper, which could improve the convergence.
>
>
> >**Q6.** Assumption 3.4 is unclear.
>
> **A:**  We will **remove Assumption 3.4 entirely** using a refined analysis in the revision. The key improvement is replacing Eq. (29) with
> $$L(s)-L(s_*)\ge\mathbb{E}\_x\frac{\tau}{2}(\mathbb{E}\_{y\sim p_x^-}|\exp(s_*(x,y)/\tau))-\exp(s(x,y)/\tau)|)^2.$$ We will update the manuscript accordingly.
>
> >**Q8.** The paper could benefit from discussing the relation of CRL to Gaussianity.
>
> **A:** Thank you for highlighting this important recent line of work. We view our results and Gaussianity as complementary: our theory provides the **statistical guarantee** that optimal CRL leads to optimal retrieval performance, while Gaussianity reveals the **latent structure** that enables such good generalization [1]. We will add a discussion on this connection in the revised version.
>
> >**Q9.** Is the stylish I is indicator function, in Eq. 2? "Denote I be the indicator function such that an event E holds true if I[E]=0."
>
> **A:** We apologize for this error. The symbol $\mathbb{I}(\cdot)$ denotes the standard indicator function, and the correct definition is $\mathbb{I}[E]=1$ when event $E$ holds. We will fix this in the revision.
> >**Q 10.** Why do you need to measure specifically the state where the inner product of the positive is equal to the negative? Isnt it extremely rare for a high-dimensional vectors?
>
> **A:** You are correct that in high-dimensional feature spaces, the event $s(x, y) = s(x, y')$ is extremely rare. We include the $\frac{1}{2}$ term for mathematical completeness and rigor, consistent with **standard formulations in AUC and pairwise learning** (e.g., [2]).
> >**Q11.** You defined the L function on a single score (in Eq. 1), and it is clear. Then you used it in Eq. 3 for a sequence of scores, what does it means?
>
> **A:** $L(s_n)$ follows the **same definition** as Eq. (1):
> $$L(s_n)=\mathbb{E}\_x\mathbb{E}_{y\sim p_x^+}\tau\log\mathbb{E}\_{y'\sim p_x^-}\exp(\frac{s_n(x,y')-s_n(x,y)}{\tau}) .$$ We will clarify this notation explicitly in the revised manuscript.
>
> [1] Bester et al. InfoNCE induces Gaussian distribution.
>
> [2] Gao & Zhou On the consistency of auc pairwise optimization.
>
> [3] Wei et al. FastCLIP: A Suite of Optimization Techniques to Accelerate CLIP Training with Limited Resources.
>
> We hope that our responses to your comments are satisfactory. Thank you again!

---

> > ### Author Rebuttal · Reviewer_mXAj · 2026-04-01
> >
> > Im a bit torn about this paper, the topic is important, and the derivations has some interesting insights - something all reviewers may agree on. Having said this, I think it is highly difficult to follow the paper mathematically. Moreover, I deeply do not agree with the fact that pure learning theory do not have to be accompanied with an empirical evidences. For example, the amount of negatives is very relevant. In general in order to develop the mathematics we have a decent amount of assumptions, some are stated and some are hidden. Showing it in the "real life", is strengthen the assumptions validity, to make sure anything isn't hidden somewhere beneath. All reviewers including me agree that the work is valuable, my main concern is regarding the clarity and fluently of the manuscript, so - Im leaving my current score and let the AC decide in this case.

---

> > > ### Author Response · Authors · 2026-04-01
> > >
> > > We thank the Reviewer for the thoughtful follow-up and for acknowledging that our work addresses an important topic with interesting insights. We take the concerns regarding mathematical clarity and empirical grounding seriously and make our every effort to make the math more accessibly to the audience by adding notation tables and providing comments of practical
> > > implication of our theoretical results as well as writing down the constants explicitly in the revised version.
> > >
> > > We completely agree that theory should ground itself in "real-life" observations. As we responded, we have removed the bounded assumption (assumption 3.4) entirely, and provided additional experiments on MSCOCO and Flickr (provided in the anonymous link, commit e04d673) demonstrate that our findings hold across diverse, large-scale datasets, bridging the gap between rigorous learning theory and practical CRL systems.
> > >
> > > we hope that this is satisfactory. if you get any further questions/comments, feel free to let us know. We sincerely hope the score will reflect our true contribution and the value of our novel work.
> > >
> > > regards
> > > Authors

---

### Official Review · Reviewer_zeRB · 2026-03-11

**Soundness:** 3
**Presentation:** 3
**Significance:** 3
**Originality:** 3
**Overall Recommendation:** 5
**Confidence:** 3

**Summary:**

The paper studies the statistical consistency of contrastive learning by proving the correlation between the contrastive risk in Eqn. (1) with the probability that positive sample is ranked above negative sample given an anchor sample, i.e., Eqn. (2). The main contributions include the consistency result in Theorem 3.1. and generalization bounds for deep neural networks in supervised and self-supervised contrastive learning regimes.

**Compliance With Llm Reviewing Policy:**

Affirmed.

**Final Justification:**

The authors have resolved my concerns satisfactorily.

**Key Questions For Authors:**

Q1: I am not familiar with the line of results that connect contrastive risk to downstream tasks. However, I would like to understand the difference between the authors' provided result and results in the line of surrogate gaps between supervised and contrastive risk such as [Arora'19, Theorem 4.5]. Does these types of results already imply the consistency property given in the authors' paper?

Q2: To me the excess risk bounds (Sec. 4) and the consistency result seems a bit independent of each other in the manuscript. Is there any place in the manuscript that specifies that the SCRL and SSCRL risks are close to the [Wang&Isola'20] risk?

Q3: In some previous analysis in CRL (e.g., [Arora'19], [Lei'23], [Hieu&Ledent'25]), a multi-class contrastive risk definition (with a logistic loss) is used:
$$
L(f) := \mathbb{E}\_{c\sim\rho}\mathbb{E}\_{x,x^+\sim D\_c}\mathbb{E}\_{x_{1:m}^-\sim D\_c'}\log\left(1+\frac{1}{\tau}\sum_{k=1}^m e^{\Delta_{\bf w}(x,x^+,x_k^-)}\right).
$$

Here, $\rho$ is a categorical distribution over $\\{1,\dots,C\\}$ and $D\_c$ refers to the class-conditional distribution corresponding to class $c$ where $D\_c'$ is the class-conditional negative distribution. Notice that here the anchor, positive and negatives in each contrastive learning tuple are independent (given the class $c$), making this risk differs from [Wang&Iosla]. My question is: In multi-class settings, can the consistency result be proven for the above risk, i.e., does minimizing the above risk result in maximizing the multi-class version of the "positive-outranks-negative" probability? How different will the analysis be?

**Note**: I am going to temporarily score this paper a 4 (weak accept) because I need to better understand the manuscript. However, I am generally fond of the result in this paper and will very likely increase my evaluation to 5 (accept) provided that the authors address my questions adequately.

---

**References**:

[Khosla et al.'20] Supervised Contrastive Learning, NeurIPS 2020.

[Hieu&Ledent'25] Generalization Analysis in Supervised Contrastive Representation Learning under Non-IID Settings, ICML 2025.

**Limitations:**

Yes, the authors adequately discussed limitations.

**Strengths And Weaknesses:**

**Soundness**: The submission is technically sound, all results are properly supported with proofs.

**Presentation**: I appreciate the authors' presentation of the manuscript. The main results are generally easy to follow. However, I have a slight difficulty scrolling back and forth to revise the definitions of supervised ans self-supervised risks and to check their differences from the [Wang&Isola'20] risk. Maybe the authors can consider putting these definitions in a table.

**Significance**: In my opinion, the most novel contribution is the consistency result in Theorem 3.1. Most of the existing results in the CRL literature often relate the contrastive risk with the downstream classification error but this work relates it to the probability that positive samples outrank negative ones in terms of similarity to the anchors.

**Originality**: The originality mostly lies in the consistency result. The generalization bounds, as far as I understand, use standard parameter-counting techniques in [Long&Sedghi'19] or [Graf'22] and the cover chaining arguments (such as [Lei'23]). The small nuance is that the empirical risk used in the manuscript are biased and incurred a cost of $O(1/m)$ and $O(1/\sqrt{m})$.

---
### Some comments
---
**C1** (minor): In the self-supervised regime, In Eqn. (7), the notation $p\_{\mathcal{Y}}(y)$ is defined to mean the distribution of $y$ conditioned on it being negative for $x$ (this distribution depends on $x$). Earlier (in the Notations paragraph at the beginning of seciton 2), the same notation is used to  mean the marginal distribution over $y$ (a distribution which doesn't depend on $x$). This is a possible notation clash that could affect the clarity of the results. From the comments at the end of Sec. 3 stating that the negative samples are shared across all positives, one can indirectly deduce that in equation (8), the distribution of $y'$ follows the distribution of the original notation ($y'$ is independent of $x$). This matches the philosophy of modelling the self-supervised case as this allows for "class-collision" in the negatives, a key distinguishing characteristic compared to the SCRL formulation in equation (5).

**C2**: The definition in Eqn. (7) says that the positive sample is drawn conditional on the anchor $x$ without data-augmentation. This implies that positives have to be dependent on the anchors via some different variable like labels. This makes the SSCRL settings in this manuscript different from many prior works that assume positives are generated through augmentation ([Wang'22], [Cui'25] or [HaoChen et al.'21], to cite a few). The indirect requirement for labels (through the ability to generate a sample $y$ known to be positive for $x$) makes me unsure if the authors can call Eqn. (8) a self-supervised risk. Please help me clarify this point briefly.

---

**References**:

[Arora'19] A theoretical analysis of unsupervised contrastive representation learning, ICML 2019.

[Long&Sedghi'19] Generalization bounds for deep convolutional neural networks, ICLR 2019.

[Graf'22] On Measuring Excess Capacity in Neural Networks, NeurIPS 2022.

[Lei'23] Generalization Analysis for Contrastive Representation Learning, ICML 2023.

[Wang'22]  Chaos is a ladder: A new theoretical understanding of contrastive learning via augmentation overlap. ICLR 2022.

[Cui'25] An augmentation-aware theory for self-supervised contrastive learning. ICML 2025.

[HaoChen et al.'21] Provable guarantees for self-supervised deep learning with spectral contrastive loss, NeurIPS 2021.

---

> ### Author Rebuttal · Authors · 2026-03-31
>
> Thanks for acknowledging the strengths of our work. Our point-to-point response to your comments follows next.
> >**Q1.** The concern for the notation $p_{\mathcal{Y}}(y)$.
>
> **A:** Thank you for your careful reading and constructive feedback. We will resolve the notation ambiguity in the revised version.
>
> In Eq. (1), $p_x^-$ denotes the distribution of negative samples conditioned on anchor $x$, which may depend on $x$ as in supervised CRL. In self-supervised CRL, by contrast, negatives are sampled from the marginal distribution $p_{\mathcal{Y}}(y)$, independent of $x$. This setup is standard in existing work [1,2] and naturally induces class collision, which distinguishes SSCRL from SCRL as you noted.
>
> >**Q2.** Eqn (7) says that positives have to be dependent on the anchors via some different variable like labels.
>
>  **A:** We respectfully disagree with you. Eqn (7) $p^+_x(y)=p(y|x)$ only means that the postive data $y$ is not independent of $x$. It captures the practice to define postive data, e.g., $y$ is an augmentation of $x$ in unimodal CRL, or $y$ is the corresponding text of an image $x$ in CLIP.
>
>
> >**Q3.** The SSCRL settings in this manuscript different from many prior works that assume positives are generated through augmentation.
>
> **A:** Our framework **naturally covers augmentation-based contrastive learning**. By setting $\mathcal{Y}=\mathcal{X}$, and defining the positive distribution as the augmentation conditional $p_x^+(\cdot)=\mathcal{A}(\cdot|x)$, our analysis directly applies to augmentation-based SSCRL frameworks.
>
>
> >**Q4.**  Do surrogate gaps  already imply the consistency property given in the authors' paper?
>
> **A:** Thank you for the excellent question. Our answer is NO and we will add discussion in the revised version. While previous works (e.g., [3]) provide foundational insights, **the statistical consistency of CRL addressed in our paper represents a distinct and significantly stronger theoretical guarantee**.
>
> - Previous results mainly focus on **surrogate gap**, which only bound downstream error in terms of contrastive loss, showing correlation but not optimality.
> - In this paper, we establish **Fisher consistency** and provide a **calibration-type inequality** that links CRL specifically with downstream retrieval. Unlike surrogate bounds, our result proves that an optimal scoring function in the CRL framework directly implies the Bayes optimal performance in retrieval. This is a strictly stronger guarantee not implied by surrogate bounds.
>
>
> >**Q5.**   The excess risk bounds (Sec. 4) and the consistency result seems a bit independent of each other in the manuscript.
>
>  **A:**  Although Sections 3 and 4 analyze different properties, they form a **unified learning theory for CRL**. The consistency implies that minimizing the CRL population risk $\mathcal{L}(s)$ actually minimizes the downstream retrieval error $\mathcal{E}(s)$. The generalization analysis qantifies how the empirical risk $\widehat{\mathcal{L}}(s)$ converges to the population risk $\mathcal{L}(s)$ as the number of samples ($n$) and negatives ($m$) increases. These results together characterize the full pipeline from empirical training to downstream performance.
>
>
> >**Q6.**  Is there any place in the manuscript that specifies that the SCRL and SSCRL risks are close to the [Wang&Isola'20] risk?
>
>  **A:**  We cite [4] for introducing Eq.(1). We also compare with the theoretical bound in [4] in Lines 323-326 (right).
>
> >**Q7.** The consistency of multi-class settings in [3].
>
> **A:** Thank you for the constructive suggstion. We will add discussions in the revision.
> - We believe that there are some differences. The population risk in [4] takes the expectation of Eq. (6), and thus is different from Eq. (1). **The Fisher consistency of InfoNCE loss has been established in [5], as mentioned by reviewer bB2V**. Exploring its connection with multi-classification would be an interesting future problem.
> - However, our framework can cover the multi-class setting if the population risk is defined similar to Eq. (1), i.e.,
>
> $$L(s)=\mathbb{E}\_{c\sim\rho}\mathbb{E}_{x,x^+\sim D_c}\log\mathbb{E}\_{x^-\sim D_c'}\exp(\Delta(x,x^+,x^-)).$$
>
> We can apply our consistency results by setting $\mathcal{Y}=\mathcal{X}$ and
> $$p_x^+(x')=D\_{c(x)}(x'),p_x^-(x')=D\_{c'(x)}(x'),$$
> where $c(x)$ is the label of $x$.
>
> [1] Oko et al. A statistical theory of contrastive pre-training and multimodal generative ai.
>
> [2] Uesaka et al. Weighted point set embedding for multimodal contrastive learning toward optimal similarity metric.
>
> [3] Arora et al. A theoretical analysis of unsupervised contrastive representaion learning.
>
> [4] Wang&Isola Understanding contrastive representation learning through alignment and uniformity on the hypersphere
>
> [5] Ryu et al. Contrastive predictive coding done right for mutual information estimation.
>
> We hope this can address your questions and raise the score. Thank you again!

---

> > ### Author Rebuttal · Reviewer_zeRB · 2026-04-03
> >
> > I thank the Authors for their responses. My biggest concern about the difference between the main results and surrogate bound results has been resolved. As such, I will update my score to an Accept. Thank you very much.
> >
> > On a side note, Authors might cite [Long&Sedghi'19] and [Graf'22] since the generalization analysis approach is probably inspired by their work.

---

> > > ### Author Response · Authors · 2026-04-08
> > >
> > > thank you, reviewer zeRB, for your constructive suggestions and updating the score. We will surely include the references  [Long&Sedghi'19] and [Graf'22] in the revised version and add related discussions.  thank you again.
> > >
> > > regards
> > > Authors

---

### Official Review · Reviewer_XbBf · 2026-03-12

**Soundness:** 3
**Presentation:** 3
**Significance:** 3
**Originality:** 3
**Overall Recommendation:** 5
**Confidence:** 3

**Summary:**

This paper introduces a statistical learning theory for contrastive representation learning. The paper approaches the study of statistical consistency of CRL using a ranking-based performance measure (Eq. (2)) and shows that as the scoring function (i.e., representation learning mechanism) converges to the optimal scoring function (lower contrastive loss) it also achieves the highest performance measure. Moreover the authors derive generalization bounds for CRL that decrease as the number of negative grows, which is consistent with the empirical observations (a fact that has not been explained by the relevant recent literature). Experiments involve training a CLIP model in samples of the DFN dataset and then evaluting its zero-shot performance on ImageNet (classifiation), MSCOCO and Flickr (retrieval) to explore the practical behaviour of a CLIP model with respect to the bounds.

**Compliance With Llm Reviewing Policy:**

Affirmed.

**Final Justification:**

The authors have addressed my main concerns in their rebuttal, by indicating that they will clarify issues related to the use of negative samples and the exponential terms that were hidden using the $\tilde{O}$ notation.

**Key Questions For Authors:**

I would be happy to read the authors' thoughts and comments on the issues raised in the "Weaknesses" section above. Even if some limitations (such as the effect of the exponential terms of the bound) cannot be directly addressed, these should be clearly and openly mentioned in the paper and not "hidden" behind an approximation. Also, a clear discussion on the limitations of the suggested bounds should be offered. My review score assumes that these issues are clearly stated and I would be happy to increase it if these issues are effectively addressed.

**Limitations:**

The paper should clearly identify and state how the proposed theoretical results and bounds deviate from the practical settings of InfoNCE, SimCRL and CLIP. They should also clearly identify the limits and conditions under which the proposed bounds are useful.

**Strengths And Weaknesses:**

### Strengths

The two main contributions of the paper, as claimed by the authors are indeed its main strengths:

 - Rigorous investigation of an important topic. As the paper indicates, CRL and especially SSCRL is very important for model training today, so the deeper investigation and understanding of the statistical learning properties of CRL is highly important.

 - Theoretical results that are consistent with observations with respect to the effect of the number of negative training samples on the model effectiveness.

 - The experiment provided by the paper seems to be consistent with theoretical observations, with the empirical dependence falling between the $\sqrt n$ and $n$ lines (for the "critical negative data size", i.e., the value of $m$ for which performance saturates).

### Weaknesse

1. One concern for this paper is its definitions of contrastive losses which are different from the empirical contrastive losses used in practice and how these may lead to differences in results. Eq. (1) is almost the same as Eq. (2) of the Wang & Isola 2020 paper (multiplied by $\tau$, which is OK), but this definition is the asymptotic convergence point of the contrastive loss as the number of negative samples goes to infinity. As a result, the derived SCRL loss of Eq. (6), is different from InfoNCE. This results from the fact that InfoNCE includes the positive pair in its own denominator. When the model starts performing well and the negative term is low, these two losses give different values (and gradients).

For SimCLR there are additional differences related to the non-independent sampling of negative examples (Eq. (7)), since the negative examples of one term are positive for another term. This discrepancy between theory and practice should be clearly stated.

Regarding CLIP, in Eq. (9) the positive and negative pairs are drawn independently, so with high probability the positive pair does not appear in the denominator, contrary to the CLIP loss which always includes the positive pair in the denominator. Also, the issue with the independent sampling of the negatives is also present here.

In addition to the above, in practice the number of positive and negative samples often directly depend on the batch size, so they are not independently varying hyperparameters.

The authors should explicitly clarify any simplifying assumptions and / or limitations such as the above.

2. Another question / concern is the use of $\tilde{O}$ vs $O$. For example, the result of theorem 4.6 leads to \tilde{O} which is simplified using $O$ in the main text. The exact meaning of \tilde{O} should be clarified. More specifically, how does the big $O$ notation absorb the extra terms?

3. Following from the previous remark, the term $\exp(4B/\tau)$ in the theorem is huge for CLIP and other large networks. For example if $L=12$ and $s_l = 2$ for all layers then $B = 2^{24}$ and $\exp({4B/\tau})$ is practically infinite. This invalidates the bound in practice, however this term is hidden from the main paper.

---

> ### Author Rebuttal · Authors · 2026-03-31
>
> Thanks for acknowledging the strengths of our work. Our point-to-point response to your comments follows next.
>
> >**Q1.**  Thank you for your comments. Eq. (1) is the asymptotic convergence point of the contrastive loss as the number of negative samples goes to infinity.
>
> **A:** The advantages of Eq. (1) as the population risk can be found **in our answer to Q1 of reviewer bB2V**.
>
> >**Q2.**  Eq. (6), is different from InfoNCE. This results from the fact that InfoNCE includes the positive pair in its own denominator.
>
> **A:** The positive pair in the denominator of InfoNCE is arguably not a good feature.  The **Decoupled contrastive learning (DCL)**[2] explicitly removes the positive pair in the denominator and has been showed to behave better than InfoNCE.  In addition, the difference between InfoNCE and Eq. (6) could be negligible when the number of negative examples $m$ is very large since we average over all negative samples. Furthermore, the formulation of Eq. (6) enjoys a pairwise nature, which is more natural for us to define and analyze the consistency.
>
> >**Q3.** The negative examples of one term are positive for another term. This discrepancy between theory and practice should be clearly stated.
>
> **A:** Thank you for your insightful comments. We will acknowledge such gap in the revised version. However, similar independent assumptions  have been adopted in existing theoretical work of CRL ([4,6]). How to make more realistic assumptions is generally an important problem in learning theory.
>
> >**Q4.** In practice the number of positive and negative samples often directly depend on the batch size.
>
> **A:**  We would like to point out a clear difference between our analyzed objective and the practical optimization algorithms.
>
> - We analyze the theoretical properties of empirical CRL defined on a large set of positive and negative samples and examine how their sizes affect the generalization error.
> - Our theory does not contradict to the practical optimization algorithms used to optimize Eqn. (6) and Eqn. (10) based on mini-batches. For example, [3] exploited advanced compositional optimization algorithm to explicilty optimize these objectives by using only mini-batches per-iteration.
>
> > **Q5.** The use of $\tilde{O}$ vs $O$. For example, the result of theorem 4.6 leads to \tilde{O} which is simplified using $O$ in the main text.  More specifically, how does the big $O$  notation absorb the extra terms?
>
> **A:** Thank you for your excellent point. The result  of theorem 4.6 should be \tilde{O}, which hides logarithmic terms. More specifically, the result of Theorem 4.6 should be
> $$
> 24\log(1+8\exp(4B/\tau)nL\Pi_{l=1}^Ls_l^2)(\frac{B\sqrt{\sum_{l=1}^Ld_ld_{l-1}}+B\sqrt{\log(1/\delta)}}{n}+\frac{\exp(4B/\tau)(B\sqrt{\sum_{l=1}^Ld_ld_{l-1}}+\tau\sqrt{n/\delta})}{m})
> $$
>
> >**Q6.** The dependence on  $\exp(4B/\tau)$.
>
> **A:** Many thanks for your comments. We agree that it is necessary to present the hidden constants in the bounds and state the limitations. We will add it in the revision.
> - The term  $\exp(4B/\tau)$ arises from standard Lipschitz and strong convexity arguments applied to the exponential function over bounded domains, and is often standard in the generalization theory (e.g., [5]). The exponential dependences also appear in previous theoretical analyses of CRL ([1,6]).
> - In practical implementations such as CLIP and SimCLR, the scoring function typically uses cosine similarity, which naturally bounds $B\in[−1,1]$. Meanwhile, the temperature $\tau$ is often tuned to scale appropriately with $B$, bringing $\exp(4B/\tau)$  into **a practically meaningful range**.
> - The primary focus of our work is to characterize **how $n$ and $m$ influence the scaling behavior of generalization**, rather than obtaining tight numerical constants. We believe it would be an interesting future problem to achieve tighter bounds and remove these exponential dependencies.
>
> [1] Wang & Isola Understanding contrastive representation learning through alighment and uniformity on the hypersphere.
>
> [2] Yeh et al. Decoupled contrastive learning.
>
> [3] Wei et al. FastCLIP: A Suite of Optimization Techniques to Accelerate CLIP Training with Limited Resources.
>
> [4] Arora et al. A theoretical analysis of unsupervised contrastive representaion learning.
>
> [5] Hardt et al. Train faster, generalize better: Stability of stochastic gradient descent.
>
> [6] Lei et al. Generalization Analysis for Contrastive Representation Learning.
>
> We hope that our responses have addressed your questions. Thank you again!

---

> > ### Author Rebuttal · Reviewer_XbBf · 2026-04-03
> >
> > Thank you to the authors for their response. The authors indicate that they will clarify issues related to the negative samples as well as the $\tilde{O}$ notation possibly hiding some potentially large terms. If these are done then my main concerns with the paper are resolved.

---

### Official Review · Reviewer_bB2V · 2026-03-16

**Soundness:** 3
**Presentation:** 4
**Significance:** 4
**Originality:** 3
**Overall Recommendation:** 4
**Confidence:** 5

**Summary:**

This paper analyzes a contrastive representation learning (CRL) loss. In contrast to the existing analyses of CRL, this paper establishes a nonvacuous generalization bound that decay in the number of negative examples $m$, which supports the common empirical findings on the benefit of large $m$. Another contribution is to study the contrastive learning under the AUC-type measure in Eq. (2), especially via the calibration inequality in Theorem 3.6. This provides a concrete operational meaning to CRL.

**Compliance With Llm Reviewing Policy:**

Affirmed.

**Final Justification:**

The authors' rebuttals have clarified my questions.
Overall, I believe that it is a nice contribution (given all the clarifications in the reviews/rebuttals). I will keep my positive score!

**Key Questions For Authors:**

On top of the questions above, some minor comments are:
- Is there a connection of OCE mentioned in the paper to the general scoring rule framework of (Ryu et al. 2025)?
- I cannot understand Remark 3.7. Can the authors elaborate?
- Define an acronym just once, rather than defining multiple times.
- Line 202 (right col): is "given" ...

**Limitations:**

- The paper studies one special instance of CRL. As alluded to in the conclusion, there may exist more general theory for generalizability of CRL losses.
- The generalization gap seems to be much tighter than the available bounds in the literature, it seems that the additive bound seems still suboptimal.

**Strengths And Weaknesses:**

## Strengths
Given the popularity of CRL, its theoretical understanding is of great importance. This paper provides the first nonvacuous generalization bound for a CRL loss. The connection to the AUC-type objective seems very natural. Despite that the analysis techniques are standard, the resulting guarantees seem novel to my understanding. (It is surprising that there have been only vacuous rates in terms of $m$!) It is also good to see an empirical validation after theoretical discussions, even though it is a rather synthetic experiment.

Overall, the paper is very carefully written, and the authors' use of math is concise and clear. In the camera ready version, if accepted, I would appreciate if the authors can expand further on the OCE losses.

## Weaknesses
Here I list some comments about the clarity of the paper.
- Why is Eq. (5) the population risk? Shouldn't the population risk be the expectation of the empirical risk in Eq. (6)? I see that Eq. (6) can be understood as a "biased estimator" of Eq. (5), but a clarification would be instructive. Also, it would be worth noting that Eq. (6) is what's actually used for InfoNCE / CPC / SimCLR.
- One crucial question is the following: I believe that the population risk of Eq. (6) for finite m, where all expectations are taken outside, should be also "Fisher consistent" in the same sense of Theorem 3.1 for any finite $m$. For example, (Theorem 3.5; Ryu et al., 2025) shows the Fisher consistency of the InfoNCE loss as a density ratio up to a multiplicative constant. Given this, the additive bound $O(1/m + \\sqrt{\\frac{\\log 1/\\delta}{n}})$ for the generalization gap seems loose, as the gap does not go away unless $m\to\infty$. If the authors can address this, it would be a complete picture.
- Also it seems that the analyses in this paper are mostly about the "contrastive learning", but not necessarily about "representation learning". Except Assumption 4.1, the parametric form of $s(x,y)$ has never been specified (please correct me if I am wrong.) Given this, is the title really fair to say that "... of contrastive representation learning" rather than "... of contrastive learning"? Representation learning is just a simple application of the general contrastive learning framework.

---
## Reference
- Ryu et al. "Contrastive Predictive Coding Done Right for Mutual Information Estimation." arXiv preprint arXiv:2510.25983 (2025).

---

> ### Author Rebuttal · Authors · 2026-03-31
>
> Thanks for acknowledging the strengths of our work. Our point-to-point response to your comments follows next.
> > **Q1.** Why is Eq. (5) the population risk?
>
> **A:** thank you for the question. It facitiliates both **optimization and generalization analyses**.
> - Employing the **full population of negative samples is well-motivated from an optimization perspective**. For example, [1] considers Global Contrastive Objective (GCL), which contrasts each positive pair with all negative pairs for an anchor point. The proposed SogCLR [1] to optimize the objective to achieve better performance than SimCLR that just uses negative data in a mini-batch.
> - Defining Eq. (1) as the population risk also **yields a better and meaningful generalization bound** than existing work. Most previous work takes $\mathbb{E}\widehat{L}(s)$ as the population risk，resulting in generalization bounds of the order $\frac{\log m}{\sqrt{n}}$ (e.g.,[2]), indicating that large $m$ would hurt the porformance. However, by taking $L(s)$ as the population risk, we derive the generalization error of the order $1/m$，consistent with empirical observations that more negatives improve generalization performance.
>
> > **Q2.** The population risk of Eq. (6) for finite m should be also "Fisher consistent" in the same sense of Theorem 3.1 for any finite $m$.  The additive bound $O(1/m+\sqrt{\frac{\log(1/\delta)}{n}})$
>  for the generalization gap seems loose.
>
> **A:** Thank you for your constructive suggstions. We will **cite the appealing work [3] which we were not aware of, and include discussions** in the revised version.
> - While Fisher consistency was established as a qualitative notion, here we derive **a quantitative calibraion-type inequality for CRL for the first time**, which is practically important since the ideal optimal scoring function is often unattainable in real training.
> - In practice, **CRL usually benefits from large $m$**. For example, Google's SimCLR paper uses $m=8192$, while OpenAI's CLIP uses $m=32768$. Therefore, we believe $1/m$ is a reasonal upper bound.
> - Furthermore, **the additive bound $O(1/m+1/\sqrt{n})$ implies the trade-off between $m$ and $n$** as discussed in lines 306-313 in our paper. This is corroborated by our empirical studies.
>
> > **Q3.** The analyses in this paper are mostly about the "contrastive learning", but not necessarily about "representation learning".
>
> **A:** Thank you for you insightful comments. We would like to clarify this from three points.
> - The representation learning aspect is captured through the approximation error term (Line 380): CRL learns a useful representation to the extent that the chosen scoring class can approximate the optimal scoring function in Lemma 3.2, namely the log-density ratio. A full characterization of this approximation property is beyond the scope of the current paper and remains an important direction for future work.
> - This submission **covers "representation learning" both theoretically and empirically**. Our theoretical results imply that  CRL learns transferable representations for downstream tasks, which is further validated in our experiments.
> - We view representation learning as a principal application of contrastive learning. On the other hand, extending our analysis to other contrastive learning settings and broader scenarios is a promising direction for future work.
>
> > **Q4.** Is there a connection of OCE mentioned in the paper to the general scoring rule framework of (Ryu et al. 2025)?
>
>
> **A:**  Thank you for excellent suggestion. We will include a detailed discussion on this connection in the revised manuscript.
>
> We do recognize meaningful connections between OCE and the scoring rule framework.
> For instance, [3] notes that scoring rules reduce to **f-divergences** in certain special cases, which are closely linked to OCE via standard Lagrange duality theory (see Proposition E.2 in our paper).
>
> > **Q5.** Can the authors elaborate  Remark 3.7.?
>
> **A:** We will elaborate more on Remark 3.7 in the revised version. In short, this remark highlights that our **general statistical consistency result for CRL naturally extends to both supervised (SCRL) and self-supervised (SSCRL) settings**.
> Our statistical consistency theorem is established for general positive and negative distributions $p_{x}^+,p_{x}^-$, so it  applies to the specialized distributions used in both SCRL and SSCRL,, which differ in how $p_{x}^+,p_{x}^-$ are specified.
>
> > **Q6.** Define an acronym just once. Line 202 (right col): is "given" ...
>
> **A:** We will revise them accordingly in the next version.
>
> [1] Yuan et al. Provable stochastic optimization for global contrastive learning: Small batch does not harm performance.
>
> [2] Lei et al. Generalization Analysis for Contrastive Representation Learning.
>
> [3] Ryu et al. Contrastive predictive coding done right for mutual information estimation.
>
> We hope that our responses have addressed your questions. Thank you again!

---

> > ### Author Rebuttal · Reviewer_bB2V · 2026-04-04
> >
> > I appreciate the authors for the detailed rebuttals. However, I still do not understand the responses in Q1.
> > In the response, the authors noted that `Most previous work takes $\mathbb{E}\widehat{L}(s)$ as the population risk，resulting in generalization bounds of the order $\frac{\log m}{\sqrt{n}}$ (e.g.,[2]), indicating that large $m$ would hurt the porformance.` However, in Eq. (12), the authors analyzed the generalization error $|{L}(s)-\widehat{L}(s)|$ by the inner error and outer error, and the outer error is exactly the generalization error if you treat $\mathbb{E}\widehat{L}(s)$ as the population risk, which is controlled as $O(1/\sqrt{n})$ by Lemma 4.4. Maybe I'm missing something, but isn't this all we need which even holds for any $m\ge 1$?
> >
> > In light of this, I don't understand the "trade-off" argument, which is `Moreover, our results uncover an explicit trade-off between m and n: when the number of anchor points n is fixed, increasing m improves generalization only up to a certain point, beyond which the benefit saturates.` If the benefit "saturates", then such phenomenon does not qualify as a "tradeoff".
> >
> > I would be happy to be clarified if I misunderstand anything!

---

> > > ### Author Response · Authors · 2026-04-04
> > >
> > > We thank the Reviewer for careful reading of our work and these thoughtful  comments/suggestions.
> > >
> > > **Q1.** Regarding the outer error $\widetilde{O}(1/\sqrt{n})$.
> > >
> > > **A**: Thank you for your constructive comment. you are right that the outer error is exactly the generalization error widley studied in the literature.  In our Lemma 4.4, we have improved the previous estimate of the outer error to $\mathcal{O}(1/\sqrt{n})$, which indeed no longer depends on $m$. This improvement is achieved by applying covering number arguments directly, rather than using Rademacher complexity as used in [1], which lead to $\mathcal O({\log m \over \sqrt{n}})$.
> > >
> > > However, the outer error bound $\mathcal{O}(1/\sqrt{n})$ itself only indicates that large $m$ would not harm the performance.
> > > specifically, the outer-error bound alone still does not fully align with empirical observations, where the generalisation performance typically improves as $m$ increases initially and then saturates once $m$ reaches a certain threshold.
> > >
> > > our statistical analysis framework involves both the inner error and outer error which leads  to the combined estimation of ${1\over m} + {1\over \sqrt{n}}$ as shown in theorem 4.5. This is consistent with the empirical observations that the generalisation performance typically improves as $m$ increases initially and then saturates once $m$ reaches a certain threshold (see more explanation about this  "trade-off" below).
> > > We will add more discussions in the revised version.
> > >
> > > >**Q2. The "trade-off" between $m$ and $n$ and the saturation phenomenon.**
> > >
> > > **A**: We thank the reviewer for the sharp observation.  We would like to clarify that the "trade-off" in our paper is in the sense of **relative contribution of each term to the total error**.
> > >
> > > Our theoretical bound $\widetilde{O}(1/m + 1/\sqrt{n})$ implies that the generalization error is bottlenecked by the larger term. When $m \gg \sqrt{n}$, the $1/\sqrt{n}$ term dominates, and further increasing $m$ yields diminishing returns, leading to the observed saturation.  We will include a detailed discussion on this error dominance and saturation property.
> > >
> > > We hope that this is satisfactory and hope that you will update your score to reflect our true contribution and the value of our novel work.  If you have any further questions, please feel free to let us know !
> > >
> > > [1] Lei et al. Generalization Analysis for Contrastive Representation Learning.

---

### Decision · Program_Chairs · 2026-04-30

**Decision:**

Accept (regular)

**Comment:**

This paper contributes to theory of contrastive learning from the two perspectives: (1) The calibration analysis to show the contrastive loss is consistent with an AUC-type target loss; (2) The estimation error of the finite-sample contrastive loss. The former is quite intuitive yet has not been formalized in the existing literature. The latter improves the existing estimation error bound from $1/\sqrt m$ to $1/m$, where $m$ is the number of negative samples. Thus, both contributions push forward contrastive learning theory significantly. All reviewers basically acknowledge the significance of the theoretical contributions; while one reviewer keeps to have concerns about the paper clarity and empirical evaluation. In AC's viewpoint, the authors addressed these issues adequately, and we can expect they will update the paper accordingly.